# MEMETRON: Memetic Response Optimizer for Reward-Guided Post-Decoding Optimization of Large Language Models

**Son The Nguyen**                                                                     *snguye65@uic.edu*
*Department of Information Decision Sciences*
*University of Illinois Chicago*

**Theja Tulabandhula**                                                                     *theja@uic.edu*
*Department of Information Decision Sciences*
*University of Illinois Chicago*

**Reviewed on OpenReview:** *https://openreview.net/forum?id=QRW8OGn3vb*

## Abstract

Modern large language models (LLMs) are commonly optimized using scalar reward signals defined over completed responses, applied both during training and at inference time. However, most such reward-guided post-decoding methods remain one-shot: they independently sample a set of responses, score each once, and select the best. Staying shallow and narrow leaves higher-reward responses unrealized, while scaling up to shallow and wide sampling exacerbates reward hacking, making downstream selection methods such as Best-of-$N$ and Self-consistency unreliable. We propose MEMETRON, an anytime memetic optimization framework that formulates reward-guided post-decoding optimization (RPDO) as discrete black-box optimization over completed responses. MEMETRON alternates between GEN-ETRON for population-based optimization and ANNETRON for annealing-based local refinement under a black-box scalar reward. On mathematical reasoning, MEMETRON increases Pass@$k$ correctness coverage and improves the selection reliability of Best-of-$N$ and Self-consistency; on instruction following, it improves LLM-judge preference. On AMC 12 2025 with Qwen3-8B (non-thinking) and Skywork-Reward-V2-Llama-3.1-8B-40M, for instance, MEMETRON raises Best-of-$N$ accuracy from 61.9% at its 16-sample initialization to 78.6% after one generation and 85.7% after five, at an average of 464 seconds and 12 batched LLM requests per generation. By contrast, one-shot Best-of-$N$ with 1024 samples reaches only 47.6%. On verifiable tasks, MEMETRON can incorporate ground-truth correctness via reward shaping. Comparing shaped and unshaped runs exposes extreme cases where the reward model and ground-truth correctness disagree, and the resulting contrastive pairs serve as training signal for reward model fine-tuning, rejection-sampling SFT warmups for RL-based training pipelines such as PPO and GRPO, and direct preference learning such as DPO.

## 1 Introduction

Large language models (LLMs) have demonstrated strong capabilities across a wide range of generation, reasoning, and decision-making tasks. However, achieving aligned behavior requires techniques beyond pretraining, which operate at two complementary stages. The first is *post-training*, where model parameters are updated via offline methods such as SFT (Radford et al., 2018) and DPO (Rafailov et al., 2023) on fixed datasets, or via online on-policy RL methods such as PPO (Schulman et al., 2017; Ouyang et al., 2022) and GRPO (Shao et al., 2024) on self-generated rollouts, typically initialized from an SFT checkpoint. The

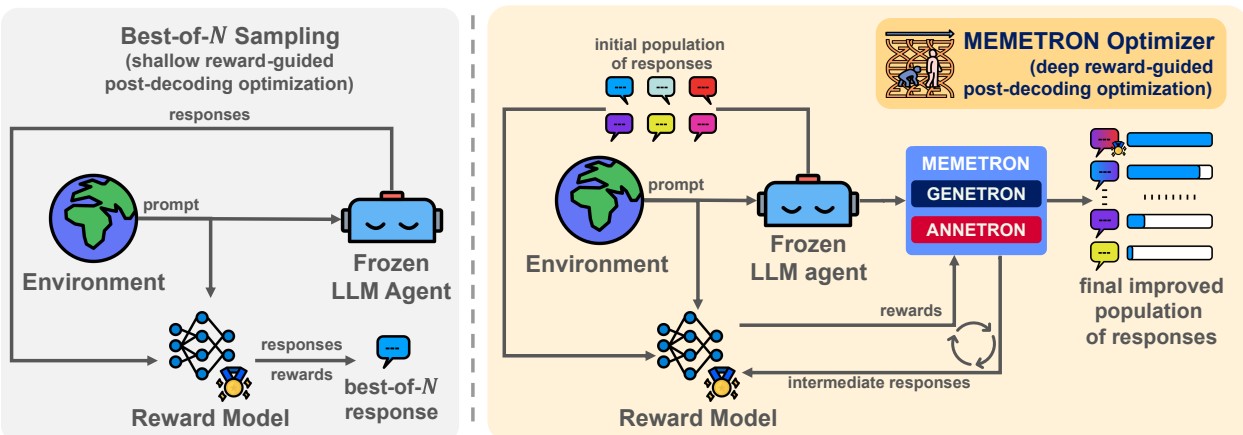

Figure 1: Comparison of shallow and deep Reward-Guided Post-Decoding Optimization (RPDO) under a scalar reward model (RM). *Shallow RPDO:* Best-of-$N$ independently samples candidates, scores each once, and selects the highest-reward response. *Deep RPDO:* MEMETRON performs structured search over the response space, combining population-based optimization via GENETRON and annealing-based local refinement via ANNETRON, and discovers higher-reward responses unreachable by shallow search.

second is *inference-time alignment*, where model parameters are fixed and outputs are guided via prompt engineering, guided decoding, or response engineering.

Offline post-training methods such as SFT with rejection sampling (Bai et al., 2022; Touvron et al., 2023; Yuan et al., 2023) and DPO with rejection sampling (Grattafiori et al., 2024), as well as inference-time response engineering methods such as Best-of-$N$ (Cobbe et al., 2021; Nakano et al., 2021), share a key pattern: the use of a reward signal, typically instantiated as a reward model (RM) that scores completed responses. Using the reward information, SFT with rejection sampling retains high-scoring candidates as fine-tuning examples, while DPO with rejection sampling builds preference pairs by contrasting high- and low-scoring samples. At inference time, Best-of-$N$ sampling selects the highest-scoring of $N$ responses for a query.

We call this family Reward-Guided Post-Decoding Optimization (RPDO), procedures that optimize completed responses under a scalar reward signal without updating model parameters. RPDO is reward-agnostic, applicable to any scalar signal including internal log-probabilities, programmatic verifiers, simulators, rubric-based scoring, human feedback, or composites of these sources, though in practice learned RMs are most common due to their scalability. Existing RPDO instantiations are typically one-shot: independently sampling a set of candidates, scoring each once, and selecting the best, analogous to random restarts in traditional optimization. This conservatism is often driven by concerns about reward hacking, favoring shallow selection over deeper optimization. However, shallow sampling does not fully address this concern and faces a dilemma: staying narrow leaves higher-reward responses unrealized, while scaling to wide sampling may exacerbate reward hacking, making downstream selection methods such as Best-of-$N$ and Self-consistency unreliable (Ichihara et al., 2025). This motivates going beyond independent candidate sampling by searching more deeply and structurally over the response space under a scalar reward objective.

Recent deeper RPDO methods have shown promising gains, but existing approaches are often specialized in both the reward signals and search strategies they employ. Mind Evolution (Lee et al., 2025) emphasizes population-based optimization via genetic algorithms and relies on task-specific programmatic verifiers to score candidates, limiting applicability to settings with reliable automatic evaluators. LLMRefine (Xu et al., 2024) targets open-ended tasks by iteratively refining responses with simulated annealing guided by a natural-language feedback model. The objective is then defined by scalarizing this feedback into a single score. This creates a two-stage pipeline of feedback generation followed by scalarization that may introduce noise and is difficult to calibrate across tasks. Consequently, existing methods are constrained either by requiring verifiable task-specific evaluators or by relying on bespoke feedback modeling and scoring pipelines. Moreover,

they typically prioritize either global optimization or local refinement rather than supporting both within a unified procedure. These limitations motivate a memetic RPDO approach that couples global optimization and local refinement under a shared scalar reward signal.

Building on these insights, we present MEMETRON, a memetic RPDO optimizer that formulates response optimization as discrete black-box optimization under scalar reward supervision. MEMETRON alternates between GENETRON for population-based optimization and ANNETRON for annealing-based local refinement within a unified procedure, using frozen LLMs as search operators. The framework requires only scalar reward access, making it compatible with any reward source. MEMETRON is also anytime: because the history buffer holds scored responses at every generation, the search can be halted after any generation, returning the best response found so far. This makes search depth an operating point rather than a fixed requirement. In this work, we instantiate MEMETRON with learned RMs, which removes dependence on task-specific programmatic verifiers, costly human evaluation, or task-specific feedback modeling and scoring design, enabling scalable evaluation across diverse tasks (see Figure 1).

Because the search space of RPDO consists of completed natural-language responses, MEMETRON uses off-the-shelf frozen LLMs as variation operators, implementing crossover and mutation for GENETRON and refinement and perturbation for ANNETRON. These operators are prompted to align with the RM objective without having access to raw reward scores, so that all operator actions are guided by semantic understanding rather than direct score feedback. Components that do not require language understanding, such as population management, annealing schedules, and bookkeeping, are handled by lightweight non-LLM modules driven by scalar reward signals.

We evaluate MEMETRON on instruction-following and mathematical reasoning tasks. MEMETRON reliably discovers higher-scoring responses according to the target RMs, and this translates to downstream gains. We distinguish oracle coverage, measured by Pass@$k$ with access to ground-truth labels, from *label-free selection*, where the final response must be chosen without ground-truth access, as in Best-of-$N$ and Self-consistency. On mathematical reasoning, MEMETRON increases Pass@$k$ correctness coverage and improves label-free selection reliability. For instance, on AMC 12 2025 with Qwen3-8B (non-thinking) (Yang et al., 2025) and Skywork-Reward-V2-Llama-3.1-8B-40M (Liu et al., 2025), each MEMETRON generation costs on average 464 seconds and 12 batched LLM requests per question; a single generation already raises best-of-$N$ accuracy from 61.9% at its 16-sample initialization to 78.6%, and five generations reach 85.7%; sampling 1024 responses instead lowers it to 47.6%. Full wall-clock and call counts for the AMC 12 experiment are reported in Appendix G. On instruction following, it improves LLM-judge preference. On verifiable tasks, we also show that MEMETRON can incorporate ground-truth correctness signals via correctness-conditioned reward shaping, favoring candidates that are both correct and highly scored by the RM. In doing so, comparing runs with and without shaping exposes extreme cases of RM-correctness misalignment, yielding contrastive response pairs that can serve as training signal for downstream pipelines; we discuss training-time, inference-time, and reward misalignment mitigation applications in Appendix H.

**Our main contributions are:**

1. **Problem formulation:** We formulate RPDO as a discrete black-box optimization problem over completed responses,

$$\max_{y\in\mathcal{Y}} r(x,y),$$

   unifying existing methods such as Best-of-$N$, SFT with rejection sampling, and DPO with rejection sampling as shallow, one-shot instantiations of this objective. The reward $r(x,y)$ is agnostic to its source and may be instantiated as a learned reward model, programmatic verifier, human feedback, or composite thereof. This framing exposes a gap: existing methods under-exploit the reward signal by treating candidates independently, motivating structured search as a principled alternative.

2. **Memetic optimization framework for RPDO:** We introduce MEMETRON, a memetic response optimizer that alternates between GENETRON for population-based optimization and ANNETRON for annealing-based local refinement under a black-box scalar reward signal. MEMETRON uses frozen LLMs as search operators prompted to align with the reward objective

without access to raw reward scores, encouraging operators to improve response quality semantically rather than exploit score artifacts.

3. **MEMETRON outperforms shallow sampling under label-free selection:** Across instruction-following and mathematical reasoning tasks, MEMETRON reliably optimizes the target RMs, discovering responses with higher RM scores. Downstream, MEMETRON increases Pass@$k$ correctness coverage and the reliability of Best-of-$N$ and Self-consistency on mathematical reasoning, and improves LLM-judge preference on instruction following. Scaling shallow sampling instead degrades Best-of-$N$ accuracy, since larger pools surface more high-reward incorrect responses.

4. **Reward misalignment diagnostics via correctness shaping:** On verifiable tasks where ground-truth correctness signals are available, MEMETRON can incorporate them via correctness-conditioned reward shaping, favoring candidates that are both correct and highly scored by the RM. This strengthens RM-based selection toward the Pass@$|\mathcal{H}|$ upper bound and, by comparing runs with and without shaping, exposes RM-correctness misalignment, yielding contrastive response pairs that can support reward model fine-tuning, rejection-sampling SFT warmups for RL-based training pipelines such as PPO and GRPO, and direct preference learning such as DPO.

## 2 Problem Formulation

We formulate RPDO as a discrete black-box optimization problem over completed LLM responses, without updating the model parameters, with the objective of identifying a high-quality response under a given scalar evaluator. Formally, given a prompt $x \in \mathcal{X}$, a set of frozen LLMs $\{\pi_m(y \mid x)\}_{m=1}^M$ (with $M \geq 1$), and a black-box scalar evaluator $r : \mathcal{X} \times \mathcal{Y} \to \mathbb{R}$ which assigns a scalar reward to each response $y \in \mathcal{Y}$, the goal is to find

$$y^\dagger \ = \ \arg\max_{y \in \mathcal{Y}} r(x, y),$$

where the response space $\mathcal{Y}$ consists of all sequences producible by the frozen models $\{\pi_m(y \mid x)\}_{m=1}^M$ under any decoding strategy. The reward function $r$ is agnostic to its source and may be instantiated as a model-derived score such as the sequence log-likelihood under a frozen model, a learned reward model, a programmatic verifier, a simulator, human feedback, rubric-based scoring, or a composite of these sources.

This setup presents several challenges:

- $\mathcal{Y}$ is discrete and finite but astronomically large, growing exponentially with the vocabulary size $|V|$ and the maximum length $\ell_{\max}$, on the order of $|V|^{\ell_{\max}}$,

- $r(x, y)$ is a black-box scalar reward signal with no gradient information, and

- the objective landscape is highly non-smooth and multimodal, where small changes in $y$ can cause abrupt jumps in $r$ and distinct high-reward regions may be disconnected.

Because the search space is discrete, non-differentiable, and highly multimodal, gradient-based methods and exact combinatorial search are ill-suited. We instead turn to memetic algorithms (MA), which combine population-based exploration via genetic algorithms (GA) with local refinement via simulated annealing (SA) and require only black-box queries to $r(x, y)$. Our goal is not to exhaustively search $\mathcal{Y}$ or guarantee global optimality, which is infeasible under realistic inference budgets. Instead, we aim to efficiently discover responses with higher reward than those obtained by standard decoding or shallow reranking.

**Reward Models** While RPDO is compatible with any black-box scalar evaluator, we primarily focus on learned RMs in this work, as they capture task-relevant aspects of response quality that are otherwise difficult to specify without hand-crafted reward functions. Unlike programmatic verifiers, which are limited to tasks with easily verifiable answers, learned RMs are broadly applicable across tasks. Once trained on human preference annotations, they efficiently approximate human judgment at inference scale without per-instance human evaluation. Furthermore, they do not require task-specific evaluation infrastructure such as manual

scalarization of natural language feedback. Learned RMs range from single-attribute scorers targeting a specific dimension to general preference models capturing multiple aspects of quality, and MEMETRON is compatible with both: for specific RMs, operators are prompted to align with the targeted dimension; for general RMs, the same RM can be reused across different objectives by adjusting the operator prompts alone, without retraining or replacing the RM.

Despite many advantages, RMs are inherently imperfect and may be partially misaligned with downstream goals, exhibiting failures in edge cases and susceptibility to reward hacking. That said, learning robust reward and preference models is a well-studied and active problem. Prior work has examined what makes RMs effective learning signals (Razin et al., 2025), how over-optimization and reward hacking can arise (Gao et al., 2023), and how such failures may be mitigated through constrained objectives (Moskovitz et al., 2023), demonstration-guided methods (Rita et al., 2024), or data-centric approaches (Nguyen et al., 2025; Northcutt et al., 2021). This work does not assume perfect reward alignment; rather, we assume access to a reasonably informative and robust RM, treating robust RM design as an orthogonal concern addressed by the literature above.

# 3 MEMETRON: Memetic Optimization for Reward-Guided Post-Decoding Response Optimization

We propose MEMETRON, a memetic RPDO optimizer that uses frozen LLMs as search operators prompted to align with the reward objective without access to raw reward scores, enabling them to propose and execute targeted edits and syntheses guided by semantic understanding of the objective. MEMETRON alternates between GENETRON for population-based optimization and ANNETRON for annealing-based local refinement, with lightweight non-LLM modules handling population management and annealing schedules via scalar reward signals.

At each generation $g$, MEMETRON maintains a population $G^{(g)} = \{y_i^{(g)}\}_{i=1}^N$ and a history buffer $\mathcal{H}$ containing all retained candidates from previous generations. The initial population is obtained by sampling from the base model,

$$y_i^{(0)} \sim \pi(y \mid x).$$

Each generation proceeds in three stages: global optimization, local refinement, and population selection. First, GENETRON applies **LLM-as-Crossover-Operator** and **LLM-as-Mutation-Operator** to the current population to generate offspring candidates. Second, each offspring is refined by AN-NETRON, which performs reward-guided local edits using **LLM-as-Refinement-Operator** and **LLM-as-Perturbation-Operator**; full details are given in Sections 3.1 and 3.2. Third, the best response found during refinement of each offspring is added to the history buffer, giving $N$ new entries, and the next population is selected as the top-$N$ responses by reward:

$$G^{(g+1)} = \arg \max_{\substack{S \subseteq \mathcal{H} \\ |S|=N}} \sum_{y \in S} r(x, y).$$

Intermediate candidates are scored but not retained, so $|\mathcal{H}|$ grows by $N$ per generation. This elitist selection strategy preserves the best solutions discovered so far while allowing exploration through stochastic generation operators. The search process terminates after a fixed number of generations or when improvement plateaus. Because $\mathcal{H}$ contains scored responses throughout the search, GENETRON, ANNETRON, and MEMETRON are *anytime* algorithms: they can be terminated after any generation or refinement step, returning the best response found so far,

$$\hat{y} = \arg \max_{y \in \mathcal{H}} r(x, y).$$

The number of generations is therefore a tunable budget rather than a fixed requirement.

## 3.1 GENETRON: Genetic Optimization for RPDO

We introduce GENETRON, a population-based evolutionary search procedure for RPDO inspired by genetic algorithms. The process begins by initializing a population $G^{(0)} = \{y_1^{(0)}, \ldots, y_N^{(0)}\} \subset \mathcal{Y}$ of $N$ candidate

---

**Algorithm 1 MEMETRON: Memetic Optimization for RPDO**

---

1: $g \leftarrow 0$
2: Initialize population $G^{(0)} = \{y_1^{(0)}, \ldots, y_N^{(0)}\}$ by sampling from model $\pi(y \mid x)$
3: Evaluate rewards and initialize history buffer: $\mathcal{H} \leftarrow G^{(0)}$
4: **while** termination criterion not met **do**
5:     Apply GENETRON's genetic operators to $G^{(g)}$ to produce offspring $O^{(g)} = \{y_1^*, \ldots, y_N^*\}$
6:     **for all** $y^* \in O^{(g)}$ **do**
7:         Refine $y^*$ using ANNETRON with respect to reward $r$
8:     **end for**
9:     Update history: $\mathcal{H} \leftarrow \mathcal{H} \cup O^{(g)}$
10:     Form next population:

$$G^{(g+1)} = \arg\max_{\substack{S \subseteq \mathcal{H} \\ |S| = N}} \sum_{y \in S} r(x, y)$$

11:     $g \leftarrow g + 1$
12: **end while**
13: **Output:** $\mathcal{H}$ and $\hat{y} = \arg\max_{y \in \mathcal{H}} r(x, y)$

---

responses, sampled from one or multiple models $\{\pi_m(y \mid x)\}_{m=1}^{M}$, with $M$ denoting the total number of models and $x$ being the input prompt. Each response $y_i^{(0)} \in G^{(0)}$ is evaluated using the scalar evaluator $r(x, y)$, yielding scores $R^{(0)} = \{r(x, y_1^{(0)}), \ldots, r(x, y_N^{(0)})\}$.

At each subsequent generation $g$, a parent set $P^{(g)} \subseteq G^{(g)}$ is selected via binary tournament selection: for each parent, two candidates are sampled randomly from the current population, and the one with the higher reward $r(x, y)$ is retained. To generate offspring, we apply ***crossover*** via ***LLM-as-Crossover-Operator***. For each offspring, a pair of parents $y_i, y_j \in P^{(g)}$ is sampled uniformly at random, and and a crossover prompt $x_{\text{cross}}(x, y_i, y_j)$ is constructed to condition on the original input prompt $x$ and the two parent responses. Concretely, $x_{\text{cross}}$ instructs the LLM to (i) carefully analyze the user query and both parent responses with respect to task requirements and reward-relevant criteria, and (ii) produce a structured synthesis plan describing how to combine the strongest complementary elements from $y_i$ and $y_j$ into an improved solution. $x_{\text{cross}}$ is designed to emphasize properties targeted by $r(x, y)$, thereby aligning crossover generation with the optimization objective. A crossover draft is then sampled as

$$y^{\text{cross}} \sim \pi(y \mid x_{\text{cross}}(x, y_i, y_j)) .$$

Next, ***mutation*** is performed via the ***LLM-as-Mutator-Operator***, which refines the crossover draft while remaining grounded in the original prompt and the parent solutions. Specifically, a mutation prompt $x_{\text{mut}}(x, y_i, y_j, y^{\text{cross}})$ is constructed from $x$, the two parents $y_i, y_j$, and the crossover draft $y^{\text{cross}}$. The mutation prompt instructs the LLM to treat the crossover draft as a primary guide and to produce a single final response that best satisfies the task requirements in $x$. Analogously, $x_{\text{mut}}$ is crafted to encourage edits that improve reward under $r(x, y)$, aligning mutation with the same objective. From each mutation prompt, we draw $n$ mutated candidates,

$$\{y_1^{\text{mut}}, \ldots, y_n^{\text{mut}}\} \sim \pi(y \mid x_{\text{mut}}(x, y_i, y_j, y^{\text{cross}})) ,$$

which implements multi-try stochastic mutation within the neighborhood induced by $x_{\text{mut}}$, reducing sensitivity to any single sample and increasing the chance of escaping low-reward local edits. We select the highest-reward mutation:

$$y^* = \arg\max_{k \in \{1, \ldots, n\}} r(x, y_k^{\text{mut}}) .$$

Prompt templates for both ***LLM-as-Crossover-Operator*** and ***LLM-as-Mutator-Operator*** operators are provided in Appendix D for both instruction-following and mathematical-reasoning tasks.

The resulting set of all generated offsprings $y^*$ of generation $g$ is denoted as $O^{(g)}$. Throughout the process, we maintain a cumulative search history

$$\mathcal{H} = G^{(0)} \cup \bigcup_{g=1}^{L} O^{(g)},$$

where $L$ stands for the last generation index. The next population $G^{(g+1)}$ is formed with **elitism**, by selecting the top $N$ responses from the full history $\mathcal{H}^{(g)}$, based on reward:

$$G^{(g+1)} = \arg\max_{\substack{S \subseteq \mathcal{H}^{(g)} \\ |S|=N}} \sum_{y \in S} r(x, y).$$

This step allows high-reward individuals from any previous generation to persist if they outperform newly generated candidates.

The evolutionary process continues until a convergence criterion is met, such as hitting a fixed computational budget (e.g., total number of model calls), reaching a predefined reward threshold, or a fixed number of generations $L$ is reached. In practice, convergence is determined by monitoring the improvement in maximum reward across generations. Specifically, if the best reward in the current population $G^{(g)}$ has not improved by more than a small threshold $\delta$ over the past $L_{\text{patience}}$ generations, the search is considered to have converged. Formally, we stop if

$$\max_{y \in G^{(g)}} r(x, y) - \max_{y \in G^{(g-L_{\text{patience}})}} r(x, y) < \delta.$$

This patience-based criterion prevents unnecessary computation once the search plateaus and further improvements are unlikely. The highest quality response for a given prompt can then be extracted from the history by computing:

$$\hat{y} = \arg\max_{y \in \mathcal{H}} r(x, y)$$

**Remark 1.** *GENETRON implements crossover and mutation via the **LLM-as-Crossover-Operator** and **LLM-as-Mutation-Operator** rather than hand-crafted combinatorial operators, enabling semantically coherent offspring generation directly in natural-language response space. These operators are coupled in a two-stage pipeline: the crossover operator analyzes the current parent responses and produces a synthesis plan and draft, which the mutation operator then refines. Both operators are prompted to align with the $r$ objective without exposing scalar reward scores to the LLM, so that candidate generation is guided by semantic understanding rather than direct score feedback. Per offspring, a small number $n$ of mutation candidates are sampled to simulate stochastic mutation, and the highest-reward one is retained. This differs from naive best-of-n in that candidates are not independently sampled from the base LLM but are conditioned on the crossover draft and parent responses, making them semantically informed proposals rather than random draws. Unlike task-specific evolutionary approaches that rely on programmatic verifiers, all selection, elitism, and convergence decisions of GENETRON are driven exclusively by scalar values $r(x, y)$, keeping it reward-agnostic and broadly applicable across tasks and $r$ instantiations.*

## 3.2 ANNETRON: Simulated Annealing for RPDO

We introduce ANNETRON, a local refinement procedure for RPDO inspired by simulated annealing. The optimization process begins by initializing a candidate response $y_0 \in \mathcal{Y}$, sampled from a conditional LLM $\pi(y \mid x)$, where $x$ is the input prompt. The corresponding reward is computed as $r_0 = r(x, y_0)$. A temperature parameter $T_0 > 0$ is also initialized, and the search proceeds over discrete time steps indexed by $t$.

At each iteration $t$, **LLM-as-Refinement-Operator** is first applied to the current candidate $y_t$ to derive a targeted improvement blueprint. In particular, a refinement prompt $x_{\text{refine}}(x, y_t)$ is constructed to instruct the LLM to (i) carefully analyze the user query $x$ and the current response $y_t$ with respect to task requirements and reward-relevant criteria, and (ii) produce a structured refinement plan that specifies concrete edits and improvements to improve $r(x, y)$. The resulting refinement artifact is sampled as

$$y_t^{\text{refine}} \sim \pi\left(y \mid x_{\text{refine}}(x, y_t)\right).$$

**Algorithm 2 GENETRON: Genetic Optimization for RPDO**

1: Initialize $g \leftarrow 0$
2: Initialize population $G^{(0)} = \{y_1^{(0)}, \ldots, y_N^{(0)}\}$ by sampling from model $\pi(y \mid x)$
3: Evaluate rewards: $R^{(0)} = \{r(x, y_1^{(0)}), \ldots, r(x, y_N^{(0)})\}$
4: Initialize history buffer: $\mathcal{H}^{(0)} \leftarrow G^{(0)}$
5: **while** termination criterion not met **do**
6:   *Selection:* Select parents $P^{(g)} \subseteq G^{(g)}$ via binary tournament selection
7:   Initialize offspring set: $O^{(g)} \leftarrow \emptyset$
8:   **for** $k = 1$ to $N$ **do**
9:     *Crossover:* Sample parent pair $y_i, y_j \in P^{(g)}$
10:    Construct prompt $x_{\text{cross}} \leftarrow x_{\text{cross}}(x, y_i, y_j)$
11:    Sample crossover draft $y^{\text{cross}} \sim \pi(y \mid x_{\text{cross}})$
12:    *Mutation:* Construct prompt $x_{\text{mut}} \leftarrow x_{\text{mut}}(x, y_i, y_j, y^{\text{cross}})$
13:    Sample $n$ candidates $\{y_1^{\text{mut}}, \ldots, y_n^{\text{mut}}\} \sim \pi(y \mid x_{\text{mut}})$
14:    Select best candidate $y^* = \underset{k \in \{1, \ldots, n\}}{\arg\max} \, r(x, y_k^{\text{mut}})$
15:    Add $y^*$ to $O^{(g)}$
16:  **end for**
17:  Update history: $\mathcal{H}^{(g+1)} \leftarrow \mathcal{H}^{(g)} \cup O^{(g)}$
18:  *Elitism:* Form next population

$$G^{(g+1)} = \underset{\substack{S \subseteq \mathcal{H}^{(g+1)} \\ |S|=N}}{\arg\max} \sum_{y \in S} r(x, y)$$

19:  $g \leftarrow g + 1$
20: **end while**
21: **Output:** $\mathcal{H}$ and best response $\hat{y} = \arg\max_{y \in \mathcal{H}} r(x, y)$

**Algorithm 3 ANNETRON: Simulated Annealing for RPDO**

1: Initialize $t \leftarrow 0$
2: Sample initial response $y_0 \sim \pi(y \mid x)$
3: Evaluate reward $r_0 = r(x, y_0)$
4: Set temperature $T_0 > 0$, decay factor $\alpha \in (0, 1)$
5: Initialize history buffer $\mathcal{H} \leftarrow \{y_0\}$
6: **while** termination criterion not met **do**
7:   *Refinement:* Construct prompt $x_{\text{refine}} \leftarrow x_{\text{refine}}(x, y_t)$
8:   Sample refinement draft $y_t^{\text{refine}} \sim \pi(y \mid x_{\text{refine}})$
9:   *Perturbation :* Construct prompt $x_{\text{pert}} \leftarrow x_{\text{pert}}(x, y_t, y_t^{\text{refine}})$
10:  Sample $n$ candidates $\{y_{t,1}^{\text{pert}}, \ldots, y_{t,n}^{\text{pert}}\} \sim \pi(y \mid x_{\text{pert}})$
11:  Select best candidate $y_t^* = \underset{k \in \{1, \ldots, n\}}{\arg\max} \, r(x, y_{t,k}^{\text{pert}})$
12:  Compute reward difference $\Delta r = r(x, y_t^*) - r(x, y_t)$
13:  **if** $\Delta r \geq 0$ **then**
14:    Accept: $y_{t+1} \leftarrow y_t^*$
15:  **else**
16:    Accept with probability $p = \exp(\Delta r / T_t)$
17:    **if** $\texttt{random()} < p$ **then**
18:      $y_{t+1} \leftarrow y_t^*$
19:    **else**
20:      $y_{t+1} \leftarrow y_t$
21:    **end if**
22:  **end if**
23:  Update temperature: $T_{t+1} \leftarrow \alpha T_t$
24:  **if** $y_{t+1} \neq y_t$ **then**
25:    Update history: $\mathcal{H} \leftarrow \mathcal{H} \cup \{y_{t+1}\}$
26:  **end if**
27:  $t \leftarrow t + 1$
28: **end while**
29: **Output:** $\mathcal{H}$ and best response $\hat{y} = \arg\max_{y \in \mathcal{H}} r(x, y)$

Next, a perturbed candidate is produced via an ***LLM-as-Perturbation-Operator***, which is analogous to the mutation operator used in population-based methods but operates on a *single* current response. A perturbation prompt $x_{\text{pert}}(x, y_t, y_t^{\text{refine}})$ is constructed from the original prompt $x$, the current candidate $y_t$, and the refinement draft $y_t^{\text{refine}}$. The prompt instructs the LLM to treat $y_t^{\text{refine}}$ as the primary guide and to produce a single candidate response that best satisfies the task requirements in $x$, while making targeted improvements relative to $y_t$. From the perturbation prompt, $n$ candidates are sampled:

$$\{y_{t,1}^{\text{pert}}, \ldots, y_{t,n}^{\text{pert}}\} \sim \pi\big(y \mid x_{\text{pert}}(x, y_t, y_t^{\text{refine}})\big),$$

which provides an implicit local random-restart effect within the perturbation neighborhood induced by $x_{\text{pert}}$, improving robustness and increasing the chance of escaping low-reward local edits. We select the highest-reward candidate:

$$y_t^* = \underset{k \in \{1, \ldots, n\}}{\arg\max} \, r\Big(x, y_{t,k}^{\text{pert}}\Big).$$

Prompt templates for both the ***LLM-as-Refinement-Operator*** and ***LLM-as-Perturbation-Operator*** operators are provided in Appendix D for both instruction-following and mathematical-reasoning tasks.

We then compute the reward difference as $\Delta r = r(x, y^*) - r(x, y_t)$, which determines whether the new candidate is accepted. If $y^*$ improves upon $y_t$ (i.e., $\Delta r \geq 0$), it is always accepted. If the reward decreases ($\Delta r < 0$), the new candidate may still be accepted with a probability given by the *Metropolis criterion*:

$$P(\text{accept}) = \begin{cases} 1, & \text{if } \Delta r \geq 0 \\ \exp\left(\dfrac{\Delta r}{T_t}\right), & \text{if } \Delta r < 0, \end{cases}$$

where temperature $T_t$ is updated at each step using a geometric decay schedule:

$$T_{t+1} = \alpha T_t, \quad \text{with } \alpha \in (0, 1).$$

This probabilistic acceptance criterion allows non-greedy exploration, enabling the search to escape local optima and increasing the likelihood of discovering globally optimal responses. At high temperatures, AN-NETRON explores more widely, frequently accepting worse responses to traverse the response landscape. As the temperature decreases, the algorithm becomes increasingly greedy, primarily accepting responses that yield reward improvements.

Throughout the optimization process, we maintain a history set $\mathcal{H}$ of all accepted responses, beginning with the initial candidate and including each accepted refinement:

$$\mathcal{H} = \{y_0\} \cup \{y_1^*, y_2^*, \ldots, y_L^*\},$$

where each $y_t^*$ denotes a response accepted at iteration $t$, and $L$ is the last iteration index.

The process continues for a fixed number of steps or until a termination condition is met (e.g., reward plateau or convergence). The best response observed across all accepted candidates is returned as the final output:

$$\hat{y} = \arg\max_{y \in \mathcal{H}} r(x, y).$$

**Remark 2.** *ANNETRON implements local refinement via the **LLM-as-Refinement-Operator** and **LLM-as-Perturbation-Operator**, enabling semantically coherent local search directly in natural-language response space. These operators are coupled in a two-stage pipeline: the refinement operator analyzes the current candidate and produces a targeted improvement plan, which the perturbation operator then uses to generate a perturbed candidate. Both operators are prompted to align with the r objective without exposing scalar reward values $r(x, y)$ to the LLM, so that candidate generation is guided by semantic understanding rather than direct score feedback. Per step, a small number n of perturbation candidates are sampled to simulate stochastic perturbation, and the highest-reward candidate under r is used as the proposal for the Metropolis acceptance criterion, which allows non-greedy exploration and escape from local optima. Unlike approaches that rely on natural language feedback models or task-specific evaluation pipelines, ANNETRON operates under any scalar reward signal, keeping it reward-agnostic and broadly applicable across tasks and r instantiations.*

## 4 Experiments

### 4.1 Mathematical reasoning

**Dataset.** We curated 42 non-figure problems from the 2025 AMC 12 A and B tests, administered in November 2025; problem statements and answer keys were obtained from Art of Problem Solving (Art of Problem Solving, 2026). AMC 12 is a U.S. national mathematics competition for high-school students presented in a multiple-choice format.

We additionally evaluate on all 30 problems from AIME 2026 I and II, administered in February 2026, using the MathArena release (Dekoninck et al., 2026). AIME requires an integer answer in $[0, 999]$ rather than a multiple-choice selection and is substantially harder than AMC 12, providing a stricter test of whether the observed gains generalize.

**Generator details.** We generate responses with Qwen3-8B (Yang et al., 2025) in non-thinking mode using temperature = 1.5, min-$p$ = 0.1, top-$k$ = 50, and a maximum generation length of 16,384 tokens. For AMC 12, we provide the model with problem statements only, omitting the multiple-choice options to encourage free-form reasoning rather than elimination-based strategies. Generations are run with vLLM (Kwon et al., 2023) on a single NVIDIA A100 (80 GB) for AMC 12 2025 and on a single NVIDIA RTX PRO 6000 Blackwell Server Edition for AIME 2026; the hardware differs only in throughput and does not affect the sampling protocol. Because both benchmarks postdate Qwen3-8B's April 2025 release, they reduce the risk of training-data contamination. All LLM operators of our proposed methods use the same model and generation settings.

**Reward model details.** MEMETRON and its algorithms are designed to work with any black-box scalar reward signal, making them compatible with general-purpose reward models (RMs) such as the Skywork-Reward-V2 family, which can be reused across tasks by changing only the operator prompts. In this experiment, we infer reward signals $r$ using Skywork-Reward-V2-Llama-3.1-8B-40M (Liu et al., 2025), a variant in the Skywork-Reward-V2 discriminative outcome RM family trained on the full 40M-pair preference dataset, which performs strongly on RewardBench v2 (Malik et al., 2025). Reward inference uses HuggingFace Transformers (Wolf et al., 2020) with batched scoring, served through an internal Gradio API hosted on a single NVIDIA A100 (80 GB) for both benchmarks.

**Baselines.** Base-16 consists of 16 independently sampled responses and serves as the shared initialization for all RPDO methods on both benchmarks. For AMC 12, to separate the effect of our proposed algorithms from simply increasing the sampling budget, we additionally report pure-sampling baselines Base-128 and Base-1024. ANNETRON and GENETRON also serve as ablations of MEMETRON, isolating local refinement and population-based search respectively.

**Proposed RPDO methods.** We evaluate all three of our proposed deep RPDO algorithms: AN-NETRON, GENETRON, and MEMETRON. All methods are initialized from Base-16. ANNETRON runs for five iterations and GENETRON runs for five generations; in both cases, each step retains exactly 16 responses. Over five steps, this yields a final history buffer $\mathcal{H}$ with $|\mathcal{H}| = 96$. At each step, the **LLM-as-Perturbation-Operator** in ANNETRON and the **LLM-as-Mutation-Operator** in GENETRON each sample $n = 3$ candidates conditioned on the refinement draft and crossover draft, respectively, and retain the highest-reward candidate. ANNETRON's annealing schedule uses an initial temperature $T_0 = 1.5$ and a cooling rate $\alpha = 0.975$, applied multiplicatively at each iteration ($T_{t+1} = \alpha T_t$), with the standard Metropolis acceptance criterion. MEMETRON combines both operators, running for five generations, where each MEMETRON generation consists of one GENETRON generation followed by five ANNETRON iterations, and the annealing temperature is reset to $T_0$ at the start of each refinement stage. All LLM operators are prompted to align with the mathematical reasoning objective without access to numerical reward scores.

To isolate the effect of correctness-signal augmentation, we additionally evaluate MEMETRON with *correctness shaping* on AMC 12, which modifies the reward as $r' = r + c \cdot \mathbb{1}[\text{answer is correct}]$, where $r$ is the base reward score and $c = 20$. Like Pass@$|\mathcal{H}|$, this is an oracle-assisted diagnostic requiring ground-truth labels rather than a label-free inference-time strategy. All other aspects of the optimization procedure remain identical to MEMETRON. Although the proposed methods support reward-based early stopping, we disable adaptive stopping to ensure controlled and comparable evaluation across methods. Prompt templates for the LLM operators are provided in Appendix D.2.

**Evaluation metrics and protocol.** We evaluate performance using four metrics: *Pass@$|\mathcal{H}|$* accuracy, *Best RM*, *Best-of-$|\mathcal{H}|$* accuracy, and *Self-consistency* accuracy. Here, $|\mathcal{H}|$ denotes the size of the history buffer accumulated by an algorithm up to that step; *Pass@$|\mathcal{H}|$* and *Best-of-$|\mathcal{H}|$* are therefore the standard *Pass@k* and *Best-of-N* metrics evaluated over the entire history buffer. We emphasize that $|\mathcal{H}|$ is a buffer size, not a compute budget: variation operators issue substantially more LLM calls per step than the number of responses retained in $\mathcal{H}$ (Appendix G). For the sampling baselines, which maintain no history, $|\mathcal{H}|$ coincides with the number of responses sampled. *Pass@$|\mathcal{H}|$* measures whether at least one correct solution appears in the candidate pool and serves as an upper-bound indicator of correctness coverage; it is most useful when automatic answer verification is cheap and reliable. *Best RM* is the highest reward score observed in $\mathcal{H}$, reported as the mean across problems with 95% confidence intervals. *Best-of-$|\mathcal{H}|$* accuracy and *Self-consistency* accuracy measure the accuracy of the highest-reward answer and the most frequent answer in $\mathcal{H}$, respectively, both reflecting label-free selection rules. Full definitions of *Pass@k*, *Best-of-N*, and *Self-consistency* are provided in Appendix E.

The $|\mathcal{H}|$ axis is intended to expose a structural difference between the two families of methods. Best-of-$N$ scores responses independently and in parallel, with no carry-over across samples, whereas RPDO methods maintain a growing history buffer in which each successive step is conditioned on previously scored responses.

Table 1: Performance of sampling baselines and proposed RPDO methods on 42 non-figure AMC 12 2025 problems, using Qwen3-8B (non-thinking) as the generator and Skywork-Reward-V2-Llama-3.1-8B-40M as the reward model. Base-16 is the shared initialization pool for all RPDO methods; Base-128 and Base-1024 are pure sampling baselines to illustrate their limitations. Each RPDO step appends 16 new responses to the history buffer, so all methods at five steps reach $|\mathcal{H}| = 96$. MEMETRON with correctness shaping is an oracle variant that augments the reward signal with ground-truth correctness. Metric definitions are in Appendix E; computational cost in Appendix G; experimental details in Subsection 4.1.

| System | Pass@$|\mathcal{H}|$ | Best RM (CI$_{95}$) | Best-of-$|\mathcal{H}|$ | Self-consistency | $|\mathcal{H}|$ |
|---|---|---|---|---|---|
| Base-16 | 36/42 (85.71%) | 25.64 [22.94, 28.34] | 26/42 (61.90%) | 30/42 (71.43%) | 16 |
| Base-128 | 41/42 (97.62%) | 29.90 [27.69, 32.11] | 23/42 (54.76%) | 29/42 (69.05%) | 128 |
| Base-1024 | 42/42 (100.00%) | 32.92 [30.81, 35.04] | 20/42 (47.62%) | 29/42 (69.05%) | 1024 |
| ANNETRON (Base-16 + 1 iteration) | 38/42 (90.48%) | 28.25 [25.65, 30.86] | 33/42 (78.57%) | 31/42 (73.81%) | 32 |
| ANNETRON (Base-16 + 2 iterations) | 38/42 (90.48%) | 30.29 [27.80, 32.77] | 33/42 (78.57%) | 33/42 (78.57%) | 48 |
| ANNETRON (Base-16 + 3 iterations) | 38/42 (90.48%) | 30.73 [28.24, 33.21] | 33/42 (78.57%) | 34/42 (80.95%) | 64 |
| ANNETRON (Base-16 + 4 iterations) | 38/42 (90.48%) | 31.60 [29.17, 34.03] | 34/42 (80.95%) | 34/42 (80.95%) | 80 |
| ANNETRON (Base-16 + 5 iterations) | 38/42 (90.48%) | 32.21 [29.86, 34.57] | 34/42 (80.95%) | 34/42 (80.95%) | 96 |
| GENETRON (Base-16 + 1 generation) | 36/42 (85.71%) | 29.47 [27.18, 31.76] | 33/42 (78.57%) | 33/42 (78.57%) | 32 |
| GENETRON (Base-16 + 2 generations) | 36/42 (85.71%) | 31.23 [28.92, 33.54] | 34/42 (80.95%) | 35/42 (83.33%) | 48 |
| GENETRON (Base-16 + 3 generations) | 36/42 (85.71%) | 32.25 [29.91, 34.59] | 35/42 (83.33%) | 36/42 (85.71%) | 64 |
| GENETRON (Base-16 + 4 generations) | 36/42 (85.71%) | 33.40 [30.98, 35.83] | 35/42 (83.33%) | 36/42 (85.71%) | 80 |
| GENETRON (Base-16 + 5 generations) | 36/42 (85.71%) | 34.01 [31.62, 36.40] | 35/42 (83.33%) | 36/42 (85.71%) | 96 |
| MEMETRON (Base-16 + 1 generation) | 39/42 (92.86%) | 33.39 [31.01, 35.78] | 33/42 (78.57%) | 36/42 (85.71%) | 32 |
| MEMETRON (Base-16 + 2 generations) | 39/42 (92.86%) | 34.62 [32.25, 36.99] | 35/42 (83.33%) | 36/42 (85.71%) | 48 |
| MEMETRON (Base-16 + 3 generations) | 40/42 (95.24%) | 35.74 [33.39, 38.08] | 35/42 (83.33%) | 36/42 (85.71%) | 64 |
| MEMETRON (Base-16 + 4 generations) | 40/42 (95.24%) | 36.81 [34.60, 39.03] | 35/42 (83.33%) | 36/42 (85.71%) | 80 |
| MEMETRON (Base-16 + 5 generations) | **40/42 (95.24%)** | **37.48 [35.32, 39.64]** | **36/42 (85.71%)** | **37/42 (88.10%)** | 96 |
| MEMETRON + Shaping (Base-16 + 1 generation) | 39/42 (92.86%) | 49.74 [46.30, 53.18] | 39/42 (92.86%) | 36/42 (85.71%) | 32 |
| MEMETRON + Shaping (Base-16 + 2 generations) | 39/42 (92.86%) | 51.74 [48.59, 54.90] | 39/42 (92.86%) | 38/42 (90.48%) | 48 |
| MEMETRON + Shaping (Base-16 + 3 generations) | 41/42 (97.62%) | 53.46 [50.58, 56.34] | 41/42 (97.62%) | 39/42 (92.86%) | 64 |
| MEMETRON + Shaping (Base-16 + 4 generations) | 41/42 (97.62%) | 54.50 [51.61, 57.39] | 41/42 (97.62%) | 39/42 (92.86%) | 80 |
| MEMETRON + Shaping (Base-16 + 5 generations) | **41/42 (97.62%)** | **55.29 [52.43, 58.15]** | **41/42 (97.62%)** | **40/42 (95.24%)** | 96 |

**Results.  Correctness coverage under automatic verification.** The underlying generator is capable of producing correct solutions for nearly all problems, but requires wide sampling to do so: Base-128 reaches 97.62% and Base-1024 achieves 100% Pass@$|\mathcal{H}|$ across all 42 problems.

Starting from the same Base-16 initialization (85.71%), the proposed deep RPDO methods guide the generator toward higher-coverage solutions *without access to ground-truth verification during optimization*: AN-NETRON reaches 90.48%, GENETRON remains at 85.71%, and MEMETRON increases to 95.24%. To assess the ceiling of RPDO under ideal reward conditions, MEMETRON with correctness shaping simulates a perfect reward model by incorporating ground-truth correctness into the reward signal, reaching 97.62%.

However, Pass@$|\mathcal{H}|$ itself requires oracle verification, which is rarely available in practice: programmatic verification is not generally available for open-ended mathematical reasoning, manual verification across hundreds of responses is highly impractical, and RMs are not yet reliable enough to substitute for ground truth.

**Label-free selection without automatic verification.** In the absence of ground-truth verification, practitioners must rely on imperfect proxies: RM-based selection such as Best-of-$N$, and aggregation heuristics such as Self-consistency. However, under pure sampling, simply increasing the response pool is counterproductive: as the pool grows, reward misranking is exacerbated by the RM's imperfect calibration over an increasingly diverse set of responses. Best RM rises monotonically ($25.64 \rightarrow 29.90 \rightarrow 32.92$ for Base-16, Base-128, and Base-1024), yet selection accuracy degrades: Best-of-$|\mathcal{H}|$ accuracy drops from 61.90% to 54.76% to 47.62%, indicating that larger response pools increasingly surface high-reward but incorrect responses; and Self-consistency accuracy remains at 69.05% for both Base-128 and Base-1024, suggesting that additional responses yield diminishing returns for majority-based aggregation. Because this degradation is monotone in pool size, matching MEMETRON's response budget with independent sampling would not recover its accuracy but reduce it further: compute spent on wider sampling does not convert into selectable accuracy.

Our proposed RPDO methods mitigate this failure mode by optimizing the response distribution against the RM, rather than merely expanding pool size. At $|\mathcal{H}| = 96$, ANNETRON reaches a Best RM of 32.21 with 80.95% on both Best-of-$|\mathcal{H}|$ and Self-consistency, while GENETRON reaches 34.01 with 83.33% and 85.71%. MEMETRON yields the strongest overall performance, reaching a Best RM of 37.48 with Best-of-$|\mathcal{H}|$ and Self-consistency accuracies of 85.71% and 88.10%, indicating that deeper, multi-step RPDO produces response sets that are both higher-scoring and more reliably selected under proxy supervision. Notably, all proposed methods attain non-overlapping 95% confidence intervals in Best RM relative to Base-16: AN-NETRON from iteration four onward, GENETRON from generation two onward, and MEMETRON from the first generation onward, providing evidence that the observed reward improvements are statistically reliable.

**Reward recognition bottleneck.** Despite these substantial gains, a gap between Pass@$|\mathcal{H}|$ and Best-of-$|\mathcal{H}|$ accuracy persists: for MEMETRON at five generations, correct solutions exist for 40 of 42 problems (95.24%), yet the RM fails to rank them highest for 4 of those, yielding a Best-of-$|\mathcal{H}|$ of 36/42 (85.71%). Best RM of MEMETRON continues to improve beyond Base-1024 (37.48 vs. 32.92), suggesting that RPDO finds extreme high-reward but incorrect responses beyond the reach of pure sampling.

To address this, MEMETRON with correctness shaping augments the RM signal with a 20-point bonus for correct responses, steering the model toward responses that are both high-reward and correct. While Best RM reaches 55.29, this reflects the 20-point bonus; accounting for it, the underlying RM scores are comparable to unaugmented MEMETRON, yet the highest-reward responses are now correct for problems where the generator can produce correct answers, and therefore selected by Best-of-$|\mathcal{H}|$. As a result, accuracies reach 97.62% and 95.24% for Best-of-$|\mathcal{H}|$ and Self-consistency after five generations, closing the gap with Pass@$|\mathcal{H}|$ (97.62%).

The contrastive pairs produced by MEMETRON with and without correctness shaping provide a concrete diagnostic and training resource, enabling reward model fine-tuning, rejection-sampling SFT warmups for RL-based training pipelines such as PPO and GRPO, and direct preference learning such as DPO, which we analyze further in Appendix H.3.

**Compute-quality tradeoff and diminishing returns.** The improvements of our algorithms come at the cost of increased inference time. Base-16, Base-128, and Base-1024 require 34.19, 93.13, and 601.13 seconds per question. Running all five steps, and including the shared Base-16 initialization, ANNETRON requires 468.50 seconds per question, GENETRON 525.29, and MEMETRON 2352.64. The higher cost of MEMETRON reflects its nested schedule: each MEMETRON generation runs one GENETRON generation followed by five ANNETRON iterations, and therefore issues six times the LLM and RM calls of either component method. This additional compute is not interchangeable with a larger sampling budget: Base-1024 spends 601.13 seconds to reach 47.62% Best-of-$|\mathcal{H}|$, while a single MEMETRON generation reaches 78.57% at an average of 497.88 seconds end-to-end, including the Base-16 initialization. Appendix G reports the full breakdown, including completions generated, candidates scored, and request counts for every system.

Substantial gains arrive early, however. The trajectory analysis for AMC 12 2025 in Appendix F.1 shows that the largest marginal gain occurs at the first step for all three methods, and is largest for MEMETRON ($\Delta = 7.75$), after which improvements accumulate at a diminishing rate: MEMETRON's Best RM rises from 33.39 at one generation to 37.48 at five. A single MEMETRON generation, at 497.88 seconds per question, already raises Best-of-$|\mathcal{H}|$ from Base-16's 61.90% to 78.57% and Self-consistency from 71.43% to 85.71%. The number of steps is therefore an operating point to be chosen against the available budget rather than a fixed requirement, and the reward-based early stopping supported by our methods can reduce average cost in deployment.

All times are problem-dependent, as harder problems tend to induce longer generations. Because this benchmark is relatively challenging, the reported runtimes likely skew toward the upper end of what we observe in easier settings. Measurements are end-to-end and include response generation, RM scoring, and the orchestration between them. RM calls are served over an internal Gradio API, so the reported times include serialization, tunnel transport, and queueing overhead in addition to RM inference itself. All runs use a Python implementation on fixed hardware, processing one problem at a time.

**Generalization to a harder benchmark.** AIME 2026 was released after our method and hyperparameters were fixed, providing an uncontaminated test of whether the observed gains generalize. Table 2 applies the same protocol on a benchmark where the Base-16 pool contains a correct solution for only 10 of 30 problems (33.33%), against 85.71% on AMC 12. The pattern holds: ANNETRON and GENETRON again perform comparably, reaching 46.67% and 50.00% Pass@$|\mathcal{H}|$, while MEMETRON is strongest at 73.33%, more than doubling the coverage of its initialization. Label-free selection improves alongside coverage: from 20.00% on both Best-of-$|\mathcal{H}|$ and Self-consistency at Base-16, ANNETRON and GENETRON reach 33.33% and 30.00% respectively on the two metrics, while MEMETRON reaches 46.67% on both. Best RM improves monotonically across steps for all three methods.

The reward recognition bottleneck is far more severe here. At generation five, MEMETRON's buffer contains a correct solution for 22 of 30 problems, yet the RM's highest-scoring response is correct for only 14: on 8 problems the answer was in the buffer and the RM preferred something else. The corresponding gap on AMC 12 is 40 versus 36. The search continues to locate correct responses as problems grow harder; what degrades is the RM's ability to prefer them. Since RPDO is agnostic to the source of the reward signal, this points to reward modeling, rather than the search itself, as the limiting factor on realizable accuracy.

We report AIME 2026 as a generalization check. Sampling baselines at matched budgets and the correctness-shaping variant were not run, so the comparisons available on AMC 12 cannot be made here.

## 4.2 Instruction Following

**Dataset.** To evaluate the effectiveness of our proposed methods on instruction-following tasks, we use the TinyAlpacaEval dataset (Polo et al., 2024), which comprises 100 examples selected from the full 805-example AlpacaEval 2.0 benchmark (Li et al., 2023) via Item Response Theory (IRT) to maximize informativeness while preserving the core evaluation properties of the original benchmark.

**Generator details.** For the instruction-following experiment, we use Llama-3.2-3B-Instruct. Responses are generated with vLLM (Kwon et al., 2023) using sampling hyperparameters: temperature $= 1.5$, min-$p$

Table 2: Performance on 30 AIME 2026 problems, using Qwen3-8B (non-thinking) as the generator and Skywork-Reward-V2-Llama-3.1-8B-40M as the reward model. Base-16 is the shared initialization pool for all RPDO methods. All settings follow Subsection 4.1.

| System | Pass@$|\mathcal{H}|$ | Best RM (CI$_{95}$) | Best-of-$|\mathcal{H}|$ | Self-consistency | $|\mathcal{H}|$ |
|---|---|---|---|---|---|
| Base-16 | 10/30 (33.33%) | 25.18 [22.85, 27.51] | 6/30 (20.00%) | 6/30 (20.00%) | 16 |
| ANNETRON (Base-16 + 1 iteration) | 11/30 (36.67%) | 28.28 [25.66, 30.89] | 6/30 (20.00%) | 6/30 (20.00%) | 32 |
| ANNETRON (Base-16 + 2 iterations) | 14/30 (46.67%) | 29.43 [26.82, 32.04] | 7/30 (23.33%) | 6/30 (20.00%) | 48 |
| ANNETRON (Base-16 + 3 iterations) | 14/30 (46.67%) | 30.07 [27.50, 32.64] | 8/30 (26.67%) | 8/30 (26.67%) | 64 |
| ANNETRON (Base-16 + 4 iterations) | 14/30 (46.67%) | 31.18 [28.74, 33.62] | 9/30 (30.00%) | 8/30 (26.67%) | 80 |
| ANNETRON (Base-16 + 5 iterations) | 14/30 (46.67%) | 31.85 [29.50, 34.19] | 10/30 (33.33%) | 9/30 (30.00%) | 96 |
| GENETRON (Base-16 + 1 generation) | 10/30 (33.33%) | 28.55 [26.05, 31.06] | 7/30 (23.33%) | 8/30 (26.67%) | 32 |
| GENETRON (Base-16 + 2 generations) | 12/30 (40.00%) | 30.05 [27.56, 32.55] | 8/30 (26.67%) | 8/30 (26.67%) | 48 |
| GENETRON (Base-16 + 3 generations) | 14/30 (46.67%) | 31.71 [29.31, 34.12] | 9/30 (30.00%) | 9/30 (30.00%) | 64 |
| GENETRON (Base-16 + 4 generations) | 15/30 (50.00%) | 32.59 [30.19, 35.00] | 10/30 (33.33%) | 9/30 (30.00%) | 80 |
| GENETRON (Base-16 + 5 generations) | 15/30 (50.00%) | 33.50 [31.19, 35.89] | 10/30 (33.33%) | 9/30 (30.00%) | 96 |
| MEMETRON (Base-16 + 1 generation) | 16/30 (53.33%) | 32.06 [29.72, 34.40] | 9/30 (30.00%) | 9/30 (30.00%) | 32 |
| MEMETRON (Base-16 + 2 generations) | 18/30 (60.00%) | 34.84 [32.61, 37.06] | 11/30 (36.67%) | 11/30 (36.67%) | 48 |
| MEMETRON (Base-16 + 3 generations) | 20/30 (66.67%) | 36.20 [33.95, 38.44] | 12/30 (40.00%) | 13/30 (43.33%) | 64 |
| MEMETRON (Base-16 + 4 generations) | 22/30 (73.33%) | 37.08 [34.90, 39.25] | 13/30 (43.33%) | 14/30 (46.67%) | 80 |
| MEMETRON (Base-16 + 5 generations) | **22/30 (73.33%)** | **37.84 [35.73, 39.96]** | **14/30 (46.67%)** | **14/30 (46.67%)** | 96 |

$= 0.1$, top-$k = 50$, and a maximum generation length of 4,096 tokens. All experiments are conducted on a single NVIDIA A100 (80 GB). All LLM operators of the proposed method in this experiment also use the same model and generation settings.

**Reward model details.** As with the mathematical reasoning experiments, MEMETRON utilizes a general-purpose RM from the Skywork-Reward-V2 family. In this experiment, $r$ signals are inferred using Skywork-Reward-V2-Llama-3.2-3B (Liu et al., 2025). We use the 3B variant to keep the reward model at a similar scale to the base generator. Reward inference uses HuggingFace Transformers (Wolf et al., 2020) with batched scoring, served via an internal Gradio API hosted on a single NVIDIA A100 GPU (80 GB).

**MEMETRON.** We run MEMETRON initialized from Base-16 for five generations, where each generation consists of one generation of GENETRON followed by five iterations of ANNETRON, with LLM operators prompted to align with the instruction-following objective without access to numerical reward scores. At each step, both the LLM-as-Perturbation-Operator in ANNETRON and the LLM-as-Mutation-Operator in GENETRON sample $n = 3$ candidates, retaining the highest-reward one. ANNETRON's annealing schedule uses an initial temperature $T_0 = 1.5$ and a cooling rate $\alpha = 0.975$, applied multiplica-

Table 3: Performance of MEMETRON on 100 TinyAlpacaEval prompts, using Llama-3.2-3B-Instruct as the generator and Skywork-Reward-V2-Llama-3.2-3B as the reward model. Base-16 is the shared initialization pool; each additional generation appends 16 new candidates to the cumulative pool, so MEMETRON at five generations reaches $|\mathcal{H}| = 96$. Metric definitions are in Appendix E; experimental details in Section 4.2.

| System | Best RM (CI$_{95}$) | Mean $\Delta$ (CI$_{95}$) | Preference (CI$_{95}$) | Margin (CI$_{95}$) | $|\mathcal{H}|$ |
|---|---|---|---|---|---|
| Base-16 | 16.60 [15.37, 17.82] | — | — | — | 16 |
| MEMETRON (Base-16 + 1 generation) | 17.93 [16.74, 19.13] | 1.34 [0.98, 1.72] | 58.65% [53.35%, 64.25%] | +17.30 pp [+06.70 pp, +28.50 pp] | 32 |
| MEMETRON (Base-16 + 2 generations) | 18.56 [17.36, 19.76] | 0.62 [0.47, 0.78] | 63.80% [57.70%, 69.85%] | +27.60 pp [+15.40 pp, +39.70 pp] | 48 |
| MEMETRON (Base-16 + 3 generations) | 19.04 [17.83, 20.25] | 0.47 [0.33, 0.63] | 64.45% [58.30%, 70.45%] | +28.90 pp [+16.60 pp, +40.90 pp] | 64 |
| MEMETRON (Base-16 + 4 generations) | 19.36 [18.16, 20.56] | 0.33 [0.20, 0.47] | 67.60% [61.20%, 74.00%] | +35.20 pp [+22.40 pp, +48.00 pp] | 80 |
| MEMETRON (Base-16 + 5 generations) | 19.62 [18.42, 20.81] | 0.25 [0.16, 0.37] | 68.30% [62.40%, 74.25%] | +36.60 pp [+24.80 pp, +48.50 pp] | 96 |

tively at each iteration, with the standard Metropolis acceptance criterion; the temperature is reset to $T_0$ at the start of each generation's refinement stage. Each generation retains 16 responses, yielding a history buffer $\mathcal{H}$ with $|\mathcal{H}| = 96$. Instruction-following prompt templates for the operators are provided in Appendix D.3.

**Evaluation metrics and protocol.** We evaluate performance using four metrics: *Best RM*, marginal gain $\Delta_g$, *Preference* win rate, and *Margin*. $|\mathcal{H}|$ denotes the size of the cumulative candidate pool available up to generation $g$ (including the Base-16 initialization). *Best RM* is the highest reward score observed in $\mathcal{H}$, reported as the mean across prompts with 95% confidence intervals. The marginal gain is defined as $\Delta_g = \text{best}(g) - \text{best}(g-1)$, where $\text{best}(g)$ denotes the best RM score in the cumulative pool at generation $g$; we report the mean $\Delta_g$ across prompts with 95% bootstrap confidence intervals. To complement RM-based evaluation, we additionally compare the *Best RM*-selected response against the Base-16 baseline using GPT-5-nano with two presentation orders (A/B and B/A; 5 judgments each, 10 total per prompt), reporting tie-aware *Preference* and *Margin* averaged across prompts with 95% hierarchical bootstrap confidence intervals (Appendix E).

**Results.** Table 3 reports MEMETRON performance across generations on TinyAlpacaEval. Best RM increases monotonically from 16.60 at Base-16 to 19.62 after five generations, with non-overlapping confidence intervals relative to Base-16 from generation 3 onward, and mean marginal gains diminishing from 1.34 at generation 1 to 0.25 at generation 5, consistent with the pattern of decreasing returns observed in the mathematical reasoning experiments. Notably, all marginal gain confidence intervals are strictly positive, indicating that every generation still contributes meaningfully to reward improvement. Pairwise preference evaluations with GPT-5-nano as the judge confirm that reward optimization translates to instruction-following quality improvements recognized by an independent judge: MEMETRON is preferred over Base-16 by 58.65% at generation 1, increasing to 68.30% at generation 5, with confidence intervals remaining above the 50% baseline across all generations, indicating that the preference gains are statistically reliable from the first generation onward. Furthermore, the win-loss margin grows from +17.30 pp to +36.60 pp, indicating that MEMETRON not only wins more often but by increasingly larger margins over generations. We provide a detailed analysis of the MEMETRON optimization trajectory in Appendix F.2.

# 5 Conclusion

We studied reward-guided post-decoding optimization (RPDO), framing inference-time improvement as discrete black-box optimization over generated responses under a scalar reward. We introduced MEMETRON, a memetic optimization framework that couples population-based optimization via GENETRON with annealing-based local refinement via ANNETRON, using frozen LLMs as search operators for semantically

meaningful transformations. Across instruction-following and mathematical reasoning tasks, MEMETRON reliably increases RM scores. On mathematical reasoning, it improves correctness coverage Pass@$k$ and selection accuracy under Best-of-$N$ and Self-consistency; on instruction following, it improves downstream quality under LLM-judge evaluation. When a verifiable correctness signal is available, correctness-conditioned shaping provides a practical diagnostic for RM-correctness misalignment by surfacing systematic misrankings and producing contrastive response pairs that can support reward model fine-tuning, rejection-sampling SFT warmups for RL-based training pipelines such as PPO and GRPO, and direct preference learning such as DPO. The gap between the correctness present in the search buffer and the correctness the RM selects widens sharply as problems grow harder, indicating that reward modeling, rather than search, is the limiting factor on realizable accuracy. Overall, our results support deeper, reward-guided post-decoding search even with imperfect reward signals, while emphasizing that reward quality and alignment remain central factors in its effectiveness.

**Limitations and future work**  MEMETRON assumes access to a reasonably informative RM and a capable base LLM, and incurs higher inference-time cost than single-pass decoding or Best-of-$N$. Because MEMETRON is anytime, search depth can be traded against budget, but reducing the cost of each generation and allocating budget adaptively across queries remain important directions for future work. Tighter integration with training pipelines is a promising avenue: MEMETRON-generated responses could supply higher-reward examples during SFT warmup or serve as off-policy data during RL training with PPO or GRPO. More broadly, alternating reward-guided search with online policy or RM updates may yield more robust closed-loop systems, and extending MEMETRON to multi-objective settings could enable principled trade-offs among correctness, factuality, style, and safety under deployment constraints.

## Broader Impact Statement

MEMETRON searches deeply over the response space, raising two main concerns. The first is environmental. Iteratively generating and scoring many responses per query consumes substantially more energy than single-pass decoding and could increase emissions from LLM serving at scale. We therefore report wall-clock time and LLM and RM call counts for every configuration and, because MEMETRON is anytime, recommend adaptive stopping in deployment. The second concern is misuse. MEMETRON strongly optimizes the objective it is given through the operators that guide its search, so an adversary could supply a harmful reward objective and operator prompts aligned to it, amplifying undesirable behavior. This capacity for aggressive optimization also makes MEMETRON useful as a diagnostic. By driving rewards toward their extremes, it reveals cases in which reward models prefer incorrect responses over correct ones and produces contrastive pairs that can be used to improve those models.

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

# A  Related Work

## A.1  Training-Time Response Optimization

Supervised fine-tuning, preference optimization, and reinforcement learning have become the dominant paradigms for training-time response optimization, as exemplified by SFT (Radford et al., 2018), DPO (Rafailov et al., 2023), and PPO (Ouyang et al., 2022). Off-policy methods such as SFT and DPO update model parameters using pre-collected responses that are independently sampled and then filtered, paired, or weighted using reward or preference models, a one-shot selection process that can leave substantial performance gains unrealized.

**Reinforcement Learning for Alignment and Reasoning** On-policy reinforcement learning methods have evolved from preference alignment to reasoning optimization. PPO (Ouyang et al., 2022) was among the first to apply RL to LLMs for preference alignment, sampling responses from the current policy and optimizing against a learned reward model. More recently, methods such as GRPO (Shao et al., 2024) have adapted on-policy RL to learn inference-time reasoning policies, explicitly trading additional test-time compute for improved performance on challenging tasks. Representative examples include *o1*, subsequent *o*-series models, and *GPT-5* from OpenAI, as well as DeepSeek-R1, which incentivize reasoning behaviors through RL-style optimization.

Post-decoding optimization methods such as MEMETRON can improve the quality of responses collected during off-policy finetuning stages such as SFT and DPO, as well as the SFT warmup stage of RL pipelines, by discovering higher-reward candidates before they are used for parameter updates.

## A.2  Inference-Time Response Optimization

Even when inference-time reasoning behaviors are learned through post-training optimization, responses can often be further improved at deployment time using algorithmic inference-time strategies that do not modify model parameters. Existing inference-time optimization methods can be broadly categorized into three paradigms: pre-decoding prompt engineering, which steers model behavior by modifying or augmenting the input prompt; guided decoding, which incorporates intermediate evaluation or control signals while tokens are being generated; and post-decoding response engineering, which refines outputs after complete candidates have been produced through selection, aggregation, or refinement.

**Pre-decoding prompt engineering.** Pre-decoding methods steer model behavior by modifying the input prompt before any tokens are generated. Manual prompt engineering includes few-shot and in-context learning (Brown et al., 2020), reasoning-oriented templates such as Chain-of-Thought prompting (Wei et al., 2022), and decomposition strategies like Least-to-Most prompting (Zhou et al., 2023), which restructure complex tasks into simpler subproblems. More recently, automated prompt engineering methods perform algorithmic search over instructions and exemplars, directly optimizing prompt configurations against task-level performance metrics, as in MIPROv2 (Opsahl-Ong et al., 2024) and GEPA (Agrawal et al., 2025). These approaches improve task performance without intervening during decoding or refining generated outputs.

**Guided decoding.** Beyond basic sampling-based decoding strategies such as temperature scaling or top-$k$/top-$p$, guided decoding incorporates intermediate evaluation signals during generation, such as verifier-guided lookahead and beam-style decoding (Wang et al., 2023a; Lightman et al., 2023; Snell et al., 2024), as well as structured exploration frameworks such as Tree of Thoughts (Yao et al., 2023; Long, 2023), Graph of Thoughts (Besta et al., 2024), and MCTS-based decoding.

**Post-decoding response engineering.** Post-decoding methods operate on complete candidates. Techniques such as Best-of-$N$ sampling, Self-consistency (Wang et al., 2023b), and prompt ensembling (Jiang et al., 2023) improve outputs by selecting or aggregating multiple generated responses using internal log-probability scores or external reward signals. We refer to this family of methods collectively as Reward-Guided Post-Decoding Optimization (RPDO). However, these approaches are largely one-shot and shallow, as they do not perform explicit iterative search or global optimization under a reward objective, limiting their ability to exploit reward signals for systematic exploration and refinement. Iterative post-generation methods such as Mixture-of-Agents (Wang et al., 2024) and Self-Refine (Madaan et al., 2023) go further but

are not formulated as reward-guided optimization processes: MoA relies on aggregation and synthesis across agent layers without an explicit optimization objective, while Self-Refine iteratively refines responses using natural-language feedback rather than scalar reward signals.

### A.3 Metaheuristic Search

**Metaheuristic Search Strategies.** Genetic Algorithms (GAs) (Holland, 1975) are population-based metaheuristics inspired by natural selection, employing operators such as selection, crossover, and mutation to evolve candidate solutions. Simulated Annealing (SA) (Kirkpatrick et al., 1983), in contrast, is a single-solution stochastic optimization method that mimics the physical annealing process and escapes local optima by probabilistically accepting worse solutions early in the search. While both approaches are effective for combinatorial and continuous optimization, they exhibit complementary strengths: GAs emphasize global exploration through population diversity, whereas SA focuses on local refinement with a controlled exploration-exploitation trade-off. Memetic Algorithms (MAs) (Moscato, 1989) combine population-based global search with local improvement procedures, often incorporating problem-specific heuristics, to achieve faster convergence and higher solution quality. Across a wide range of optimization domains, MAs have been shown to outperform pure evolutionary or local search methods by more effectively balancing exploration and exploitation (Moscato, 1989; Ong et al., 2010; Omran et al., 2005).

### A.4 Metaheuristic Search for LLM Post-Decoding Optimization

Metaheuristic algorithms have been explored in recent work on post-decoding optimization for LLMs. Several studies apply population-based evolutionary search to program synthesis and code generation, using execution results or unit tests as programmatic fitness signals (Liventsev et al., 2023; Hemberg et al., 2024). Related approaches extend LLM-guided evolutionary strategies to formal optimization problems with precisely specified objectives and constraints, where candidate solutions are directly verifiable (Liu et al., 2024; Brahmachary et al., 2025). Mind Evolution (Lee et al., 2025) adapts this paradigm to natural language reasoning by performing population-based genetic search over full-text responses, while still relying on programmatic or verifiable task signals. These methods demonstrate the effectiveness of population-level exploration when reliable programmatic evaluators are available, but their applicability is limited to settings with explicit correctness checks.

Beyond verifier-based tasks, metaheuristic optimization has also been explored for open-ended language generation. LLMRefine (Xu et al., 2024) performs iterative local refinement using a simulated-annealing-style process guided by a separately trained natural language feedback model, whose outputs are mapped to scalar scores through manually designed rules. This two-stage evaluation pipeline adds task-specific complexity and may introduce noise by collapsing rich feedback into a brittle scalar.

Overall, existing approaches either rely on programmatic verifiers that limit applicability to verifiable tasks, or on task-specific feedback modeling and scoring design that introduces additional complexity. Moreover, they typically prioritize either population-based exploration or local refinement rather than unifying both within a single framework. MEMETRON addresses both limitations by formulating post-decoding optimization as discrete black-box optimization under direct scalar reward supervision, combining population-level exploration and annealing-style local refinement while remaining broadly applicable across tasks and RM instantiations.

# B  Qualitative Mind-twisting Examples

**Example 1**

**Query**
Five monkeys are jumping around on a four poster bed while three chickens stand and watch. How many legs are on the floor?

**Before**
There are 0 legs on the floor. The four-poster bed has legs, but they are on the bed, not on the floor. The five monkeys and three chickens are on the bed, so their legs are not on the floor. No other legs are mentioned.

**After**
There are 10 legs on the floor. The four-poster bed has four legs resting on the floor. Three chickens are standing on the floor, contributing 6 legs (3 chickens × 2 legs each). The five monkeys are jumping on the bed, so their legs are not on the floor. Therefore, the total number of legs on the floor is $4 + 6 = 10$.

**Example 2**

**Query**
Sally is a girl. She has 3 brothers. Each brother has 2 sisters. How many sisters does Sally have?

**Before**
Sally is a girl. She has 3 brothers. Each brother has 2 sisters, which includes Sally and one other sister. Therefore, Sally has 2 sisters.

**After**
Sally is a girl and has 3 brothers. Each of her brothers has 2 sisters. Since Sally is one of the sisters, and each brother has 2 sisters, there must be 2 girls in the family. One of them is Sally, so the other girl is her sister. Therefore, Sally has 1 sister.

**Example 3**

**Query**
Kevin currently has 8 apples. He ate 3 apples yesterday. How many apples does Kevin have now?

**Before**
Kevin currently has 8 apples. He ate 3 apples, so he has $8 - 3 = 5$ apples left.

Answer: 5 apples.

**After**
Kevin currently has 8 apples. The information that he ate 3 apples yesterday refers to a past action. Since the question asks how many apples he has now, and it is explicitly stated that he currently has 8 apples, the answer is 8.

Figure 2: Qualitative examples of non-thinking Qwen3-4B model responses on mind-twisting questions before and after optimization with GENETRON.

## C MEMETRON's Flowchart

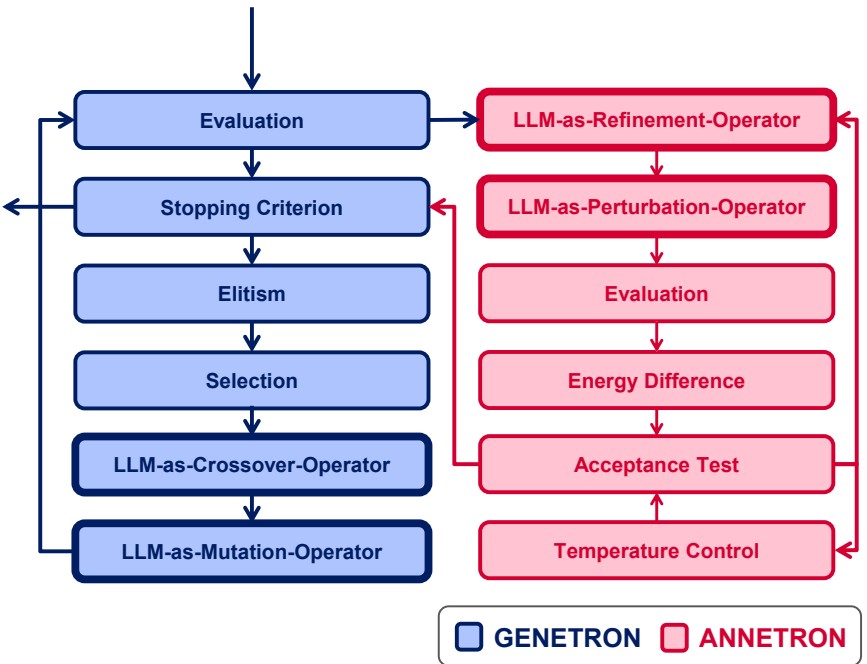

Figure 3: Flowchart of the **MEMETRON** algorithm, which performs iterative memetic search by alternating between two complementary modules: **GENETRON** for population-based genetic optimization and **ANNETRON** for simulated-annealing-style local refinement. Detailed descriptions of the algorithm are provided in Sections 3.1 and 3.2.

## D Prompt Templates

This appendix provides full prompt templates used to instantiate (i) a shared base-sampling prompt for initializing candidates, and (ii) four LLM operators: *LLM-as-Crossover-Operator*, *LLM-as-Mutation-Operator*, *LLM-as-Refinement-Operator*, and *LLM-as-Perturbation-Operator*. Operator templates are provided for two task families: Mathematical Reasoning and Instruction Following.

### D.1 Base Sampling Prompt (Shared)

---
**Base Sampling Prompt ($\pi(y \mid x)$)**

**Purpose.** Sample an initial candidate response for prompt $x$.
**Inputs.** Prompt $x$.
**Output.** A single candidate response $y$.

**Template.**
You are a helpful, reliable, and concise assistant: provide clear, relevant, direct, and accurate answers that address the user's query with essential details.
When solving math problems, show each step clearly and logically with proper formatting. Label steps for clarity, include correct units, and highlight the final answer in $\boxed{x}$ format.

Query: {query}

---

## D.2 Mathematical Reasoning Operator Prompts

---

**LLM-as-Crossover-Operator ($x_{\text{cross}}$) — Mathematical Reasoning**

**Purpose.** Produce a structured synthesis plan from two candidate solutions.
**Inputs.** Problem $x$, candidates $y_i$, $y_j$.
**Output.** A structured synthesis plan (no final answer).

**Template.**
```
You are impartial, highly confident, logical and intelligent.  Your task is to carefully evaluate the
reasoning quality of two responses to a query by checking the validity of their steps, correcting any
errors, and integrating the strongest points into a single accurate and well-supported answer.  You do not,
under any circumstances, rely on standard assumptions, stock answers, standard approaches, or conventional
interpretations of well-known, famous, classic, typical, or popular puzzles, riddles, or math problems,
nor on their stated constraints and restrictions, as these can introduce errors from recall rather than
reasoning.  Classic solutions may be invalid or misleading outside their original context.  Instead, you
must ground all reasoning in the problem itself, making only those assumptions that logically follow from
its exact wording, given constraints, and explicit structure, without importing external conventions or
interpretations.  Start your analysis right away after reading this instruction without any text wrapper.

Step 1 - Critically and carefully analyze the query and each of the responses step by step in depth before
giving any judgment of correctness.  The problem itself may be designed as a trick question or an altered
version of a popular puzzle, intentionally modified to be misleading or deceptive.  Also, do not assume
by default that any part of any responses, including its individual statements, assumptions, reasoning
steps, calculations, derivations, intermediate results, or conclusions, is correct, valid, or reliable
as they may include serious errors.  Any or all components of either response can be incorrect or wrong,
whether partially or entirely.  Even if multiple parts appear consistent with one another or align with
the other response, this does not imply accuracy or truth.  Therefore, Decompose and verify carefully and
diligently the query requirements, and each of the responses' strengths, weaknesses, key ideas, overlap,
complementary insights, unique points, direct conflicts, and reasoning paths.  Evaluate each response's
reasoning carefully for tense, causal, temporal, spatial, mathematical, constraint, counterfactual
and overall logical consistency.  Check intensely at every single step of each response for hidden
assumptions, misinterpretations, reading comprehension errors, overgeneralizations, oversimplifications,
mathematical/arithmetic/computation errors, temporal errors, spatial errors, causal errors, contradictions,
inconsistencies, constraint violations, and gaps in logic.  Do not ignore temporal or spatial anchors.
- When decompose and evaluate the query and individual solutions to ensure accuracy and detect logical or
  reasoning errors, begin by carefully compiling a complete, detailed, and accurate summary of all given
  information, facts, variables, hints, quantities, actions and their consequences in the exact correct,
  accurate, logical, and causal order presented or implied in the query, so that the structure of the
  problem is as clear and correct as possible.  Do not hallucinate, adding any information not explicitly
  stated or logically and factually implied in the wording of the query.  For each quantity, identify
  whether it functions as an input, output, condition, or intermediate value.  Pay close attention to its
  wording, including action verb tense (stole, is placed, will release, etc.), condition order (if A then B,
  B only if A, etc.), modality (certainty vs possibility:  will, may, must, etc.), logical connectors (if,
  then, and, or, not, unless, only if, if and only if, etc.), adjectives describing states (dead, closed,
  full, etc.), and quantifiers (all, some, exactly one, none, etc.), to establish causal relationships
  among variables.  Clearly specify the mathematical operation associated with each quantity, whether
  it is addition, subtraction, multiplication, division, exponentiation, or comparison.  Always check
  dependencies so that if one variable's value changes, all subsequent calculations that rely on it are
  updated accordingly, and confirm at each step that the operation chosen is consistent with the problem's
  meaning and preserves logical integrity.  During verification, never group operations together; break all
  operations into small, explicit substeps.  Always show intermediate results, carries or borrows, units,
  and signs.  Recompute each substep carefully, defer rounding until the end, cross-check when possible,
  and label each step as Verified or Corrected with a brief note.  Continuously cross-reference each
  intermediate result against the problem's exact requirements.  Finally, ensure that the overall solution
  directly answers the original question and passes a thorough reasonableness check before proceeding.

Step 2 - Based on the individually analyzed and corrected outputs from Step 1, begin constructing the most
correct and accurate reasoning and position point by point by integrating only the most accurate, relevant,
and well-supported content.
- Build the reasoning in a sequential, structured manner to ensure that each point is clear, logically
  connected, and directly responsive to the query.  Throughout this process, avoid including speculative,
  incorrect, or irrelevant material, and ensure that the emerging synthesis maintains coherence, precision,
  and fidelity to the verified logic established in Step 1.
- Systematically compare the Step-1 analyses of both responses on a point-by-point basis to determine where
```

---

they align, complement one another, diverge, or conflict. This comparison should examine assumptions,
definitions, causal relationships, temporal and spatial claims, logical and arithmetic steps, and
conclusions to explicitly surface all areas requiring integration, reconciliation, or adjudication before
the correct reasoning is assembled.
- For each point, integrate the strongest vetted content when the ideas from both responses are compatible
or mutually reinforcing, merging them into a single improved statement. When differences or apparent
contradictions arise, attempt to reconcile them by clarifying definitions, correcting flawed or incomplete
reasoning, resolving temporal or causal inconsistencies, or adjusting scope so that the two perspectives
can be unified into an accurate, coherent interpretation. If reconciliation is not possible because the
positions are genuinely incompatible, adjudicate by selecting the reasoning path that is more coherent,
better supported, and more consistent with the corrected analyses from Step 1. When both interpretations
remain correct but apply only under different conditions, present a forked conclusion that explicitly
specifies the circumstances under which each interpretation is valid.
- If both responses' analyses continues to fail to adequately justify a required point, do not rely on
their conclusions. Instead, independently reconstruct the point by reasoning from first principles,
proceeding step by step, explicitly articulating all causal, temporal, spatial, and logical relationships,
rigorously identifying and testing hidden assumptions, and revising as needed until the point is either
fully supported or shown to be unsupported by the available information.
- After drafting each integrated or adjudicated point, verify its consistency with the prior points to
maintain a coherent overall reasoning structure. If, even after reconstruction, no reliable conclusion
can be reached due to insufficient evidence or irresolvable ambiguity, explicitly state that the available
information does not permit a definitive answer.

Query: {query}

Responses:
Response A:
{y_i}

Response B:
{y_j}

---

## LLM-as-Mutation-Operator ($x_{mut}$) — Mathematical Reasoning

**Purpose.** Produce a single improved solution guided by the provided analyses.
**Inputs.** Problem $x$, parents $y_i$, $y_j$, reasoning analysis $y^{cross}$ (or ana).
**Output.** A single final response (reasoning + boxed final answer).

**Template.**
Task:
For the query below, you have been provided with two responses from different AI models and their
reasoning analysis. Your goal is to carefully study both responses and the reasoning analysis in order
to produce a response that is better than either given original response.

Step 1 – Analyze query requirements, what it asks for and what a complete and relevant response should
include. Analyze each response's strengths and weaknesses in writing quality: clarity, grammar,
conciseness, delivery, readability, and formatting style. Analyze each response's completeness and
relevance to the query. Identify improvements to strengthen writing quality and ensure adherence to the
query requirements. This analysis will guide refinements in writing style.

Step 2 – Using the writing quality analysis from Step 1 and the reasoning analysis provided only as
references, write a completely new response to the query, including both the reasoning and the final
answer. The new response must be entirely in your own words and must not copy the text of any of the
original responses directly. Your response must be derived from and accurately reflect the correct
and accurate final reasoning and conclusion identified in the given reasoning analysis below, without
introducing random or unsupported ideas. Avoid selecting or combining elements from the original
responses at random; instead, ensure your response is clearly and logically derived from the reasoning
analysis itself. It should also be more polished and well-written than the original responses, addressing
the issues identified in the Step 1 writing quality analysis. The response must be clear, concise without
losing depth, logically coherent, contextually relevant, well-structured, easy to read, and fully and
accurately address the query. The formatting style should be similar to the provided responses, but with
corrections and improvements where needed to ensure consistency, accuracy, and readability. When solving
math problems, show each step clearly and logically with proper formatting. Label steps for clarity,
include correct units, and highlight the final answer in $\boxed{x}$ format.

```
Output Format:  Provide only the new response to the query, including both the reasoning and the final
answer.  Avoid at all costs any wrapper phrases such as 'Here's the result', 'Improved response' or 'Final
response.'  Do not output any internal processes used while operating under these prompt instructions,
including intermediate analysis, evaluations, summaries, commentary, references, or explanations of how
the response was created and how it is better or adhere to the requirements.  Do not mention 'Response
A' and/or 'Response B.' Output only the new response itself, written freshly in your own words, with
formatting that matches the style of the provided responses while correcting and improving it where
appropriate.

Query:  {query}

Responses:
Response A:
{y_i}

Response B:
{y_j}

Reasoning analysis:  {ana}
```

## LLM-as-Refinement-Operator ($x_{\text{refine}}$) — Mathematical Reasoning

**Purpose.** Produce a structured refinement draft/plan from a single candidate solution.
**Inputs.** Problem $x$, current response $y_t$.
**Output.** A refinement draft (analysis + improvement blueprint).

**Template.**
```
Task:
You are impartial, highly confident, logical and intelligent.  Your task is to evaluate the reasoning
quality of a response by checking the validity of its steps, correcting any errors, preserving correct
reasoning, and producing a single accurate and well-supported answer.  You do not, under any circumstances,
rely on standard assumptions, stock answers, standard approaches, or conventional interpretations of
well-known, famous, classic, typical, or popular puzzles, riddles, or math problems, nor on their
stated constraints and restrictions, as these can introduce errors from recall rather than reasoning.
Classic solutions may be invalid or misleading outside their original context.  Instead, you must
ground all reasoning in the problem itself, making only those assumptions that logically follow from
its exact wording, given constraints, and explicit structure, without importing external conventions or
interpretations.  Start your analysis right away after reading this instruction without any text wrapper.

Step 1 - Critically and carefully analyze the query and the response step by step in depth before giving
any judgment of correctness.  The problem itself may be designed as a trick question or an altered
version of a popular puzzle, intentionally modified to be misleading or deceptive.  Also, do not assume
by default that any part of a response, including its individual statements, assumptions, reasoning
steps, calculations, derivations, intermediate results, or conclusions, is correct, valid, or reliable
as they may include serious errors.  Any or all components of either response can be incorrect or wrong,
whether partially or entirely.  Even if multiple parts appear consistent with one another or align with
the other response, this does not imply accuracy or truth.  Therefore, Decompose and verify carefully
and diligently the query requirements, and the response's strengths, weaknesses, key ideas, overlap,
complementary insights, unique points, direct conflicts, and reasoning paths.  Evaluate the response's
reasoning carefully for tense, causal, temporal, spatial, mathematical, constraint, counterfactual
and overall logical consistency.  Check intensely at every single step of each response for hidden
assumptions, misinterpretations, reading comprehension errors, overgeneralizations, oversimplifications,
mathematical/arithmetic/computation errors, temporal errors, spatial errors, causal errors, contradictions,
inconsistencies, constraint violations, and gaps in logic.  Do not ignore temporal or spatial anchors.
- When decompose and evaluate the query and the solution to ensure accuracy and detect logical or reasoning
errors, begin by carefully compiling a complete, detailed, and accurate summary of all given information,
facts, variables, hints, quantities, actions and their consequences in the exact correct, accurate,
logical, and causal order presented or implied in the query, so that the structure of the problem is
as clear and correct as possible.  Do not hallucinate, adding any information not explicitly stated or
logically and factually implied in the wording of the query.  For each quantity, identify whether it
functions as an input, output, condition, or intermediate value.  Pay close attention to its wording,
including action verb tense (stole, is placed, will release, etc.), condition order (if A then B, B
only if A, etc.), modality (certainty vs possibility:  will, may, must, etc.), logical connectors (if,
then, and, or, not, unless, only if, if and only if, etc.), adjectives describing states (dead, closed,
full, etc.), and quantifiers (all, some, exactly one, none, etc.), to establish causal relationships
among variables.  Clearly specify the mathematical operation associated with each quantity, whether
```

```
it is addition, subtraction, multiplication, division, exponentiation, or comparison.  Always check
dependencies so that if one variable's value changes, all subsequent calculations that rely on it are
updated accordingly, and confirm at each step that the operation chosen is consistent with the problem's
meaning and preserves logical integrity.  During verification, never group operations together; break all
operations into small, explicit substeps.  Always show intermediate results, carries or borrows, units,
and signs.  Recompute each substep carefully, defer rounding until the end, cross-check when possible,
and label each step as Verified or Corrected with a brief note.  Continuously cross-reference each
intermediate result against the problem's exact requirements.  Finally, ensure that the overall solution
directly answers the original question and passes a thorough reasonableness check before proceeding.

Step 2 - Based on the analyzed and corrected outputs from Step 1, begin constructing the most correct
and accurate reasoning and position point by point by integrating only the most accurate, relevant, and
well-supported content.
- Build the reasoning in a sequential, structured manner to ensure that each point is clear, logically
connected, and directly responsive to the query.  Throughout this process, avoid including speculative,
incorrect, or irrelevant material, and ensure that the emerging synthesis maintains coherence, precision,
and fidelity to the verified logic established in Step 1.
- If the response's analysis continues to fail to adequately justify a required point, do not rely on its
conclusion.  Instead, independently reconstruct the point by reasoning from first principles, proceeding
step by step, explicitly articulating all causal, temporal, spatial, and logical relationships, rigorously
identifying and testing hidden assumptions, and revising as needed until the point is either fully
supported or shown to be unsupported by the available information.
- After drafting each integrated or adjudicated point, verify its consistency with the prior points to
maintain a coherent overall reasoning structure.  If, even after reconstruction, no reliable conclusion
can be reached due to insufficient evidence or irresolvable ambiguity, explicitly state that the available
information does not permit a definitive answer.

Query:  {query}

Response:
{y_t}
```

## LLM-as-Perturbation-Operator ($x_{\text{pert}}$) — Mathematical Reasoning

**Purpose.** Generate a single improved candidate guided by a refinement draft/analysis.
**Inputs.** Problem $x$, current response $y_t$, refinement draft $y_t^{\text{refine}}$ (reasoning analysis).
**Output.** A single candidate response (reasoning + boxed final answer).

**Template.**
```
Task:
For the query below, you have been provided with a single response from a AI model and its reasoning
analysis.  Your goal is to carefully study the response and the reasoning analysis in order to produce a
response that is better than the given original response.

Step 1 - Analyze query requirements, what it asks for and what a complete and relevant response should
include.  Analyze the response's strengths and weaknesses in writing quality:  clarity, grammar,
conciseness, delivery, readability, and formatting style.  Analyze the response's completeness and
relevance to the query.  Identify improvements to strengthen writing quality and ensure adherence to the
query requirements.  This analysis will guide refinements in writing style.

Step 2 - Using the writing quality analysis from Step 1 and the reasoning analysis provided only as
references, write a completely new response to the query, including both the reasoning and the final
answer.  The new response must be entirely in your own words and must not copy the text of the original
response directly.  Your response must be derived from and accurately reflect the correct and accurate
final reasoning and conclusion identified in the given reasoning analysis below, without introducing
random or unsupported ideas.  Avoid selecting or combining elements from the original response at random;
instead, ensure your response is clearly and logically derived from the reasoning analysis itself.  It
should also be more polished and well-written than the original response, addressing the issues identified
in the Step 1 writing quality analysis.  The response must be clear, concise without losing depth,
logically coherent, contextually relevant, well-structured, easy to read, and fully and accurately address
the query.  The formatting style should be similar to the provided responses, but with corrections and
improvements where needed to ensure consistency, accuracy, and readability.  When solving math problems,
show each step clearly and logically with proper formatting.  Label steps for clarity, include correct
units, and highlight the final answer in  x  format.
```

```
Output Format:  Provide only the new response to the query, including both the reasoning and the final
answer.  Avoid at all costs any wrapper phrases such as 'Here's the result', 'Improved response' or 'Final
response.'  Do not output any internal processes used while operating under these prompt instructions,
including intermediate analysis, evaluations, summaries, commentary, references, or explanations of how
the response was created and how it is better or adhere to the requirements.  Do not mention the given
response.  Output only the new response itself, written freshly in your own words, with formatting that
matches the style of the provided responses while correcting and improving it where appropriate.

Query:
{query}

Response:
{y_t}

Reasoning analysis:
{y_t^refine}
```

## D.3 Instruction Following Operator Prompts

**LLM-as-Crossover-Operator ($x_{\mathbf{cross}}$) — Instruction Following**

**Purpose.** Produce a structured synthesis plan from two parent responses under an Instruction Following objective.
**Inputs.** Prompt $x$, parents $y_i$, $y_j$.
**Output.** A structured synthesis plan (no final answer).

**Template.**
```
You are impartial, precise, attentive to detail, and highly disciplined in rule adherence.  Your task is
to carefully evaluate how well two responses follow a user's instruction by identifying every explicit
requirement, constraint, and formatting rule, determining whether each response satisfies them, and
clearly noting any deviations, omissions, or unnecessary additions.  Correct any misinterpretations of
the instruction, identify the strongest supported elements from each response, and specify how missing
elements should be reconstructed when necessary by producing a structured synthesis plan that defines how
the responses will be integrated into a final response that most completely and accurately fulfills what
was explicitly asked in a subsequent step.

Step 1 - Critically and carefully analyze the user instruction and each of the two responses step by step
in depth before giving any judgment of instruction adherence.
-The instruction itself may contain multiple constraints, hidden requirements, conditional tasks,
formatting rules, priorities, or prohibitions that are easy to overlook or misinterpret.  Begin
by compiling a complete, detailed, and accurate breakdown of every explicit and reasonably implied
requirement in the instruction, including content requirements, exclusions, structure, tone, length,
ordering, scope, and output format, preserving their exact meaning and priority as written.  Do not infer
intentions beyond what is clearly supported by the wording of the instruction, and do not introduce new
requirements that were not stated or logically implied.
- When analyzing the two responses, do not assume by default that any part of either response, including
its interpretations, stated requirements, implied constraints, formatting choices, omissions, or
compliance claims, is correct, complete, or reliable, since either response may contain serious
Instruction Following failures.  Any or all components of either response can be noncompliant, whether
partially or entirely.  Even if both responses appear similar or mutually consistent, this does not imply
they correctly followed the instruction.  Therefore, using the precise requirements from the instruction
decomposition, verify carefully and diligently each response's compliance, noting strengths, weaknesses,
overlap, complementary coverage, unique contributions, and direct conflicts.
- Evaluate each response for faithful reading of the instruction; correct handling of constraints and
prohibitions; proper ordering and structure; correct formatting; appropriate tone and style; adherence to
any length or scope limits; and accuracy, factual correctness, relevance, internal consistency, coherence,
and safety/bias.  Check every part for hidden assumptions about what the user wanted, misinterpretations,
added unstated requirements, unnecessary content that violates constraints, missing required elements,
contradictions with the instruction, and gaps in coverage.  Continuously cross-reference each part of each
response against the instruction's exact requirements.

Step 2 - Using the requirement-by-requirement instruction-compliant analysis from Step 1 as your primary
reference, develop a structured synthesis plan that specifies how the best instruction-compliant response
will be produced.  Preserve all mandatory format, ordering, tone, length, and prohibition rules defined by
```

```
the instruction.  Identify any noncompliant material that must be removed or revised, resolve conflicts
by prioritizing the instruction's explicit constraints, and ensure the plan accounts for every required
element.  Ensure the plan also addresses deficiencies in accuracy, factual correctness, relevance,
consistency, coherence, and safety/bias where applicable.
- Systematically compare both responses on a requirement-by-requirement basis to determine where they
align, complement one another, diverge, or conflict in their interpretation and satisfaction of the
instruction.  Examine how each response handles constraints and prohibitions, required content, ordering
and structure, formatting rules, tone and style requirements, length and scope limits, and any conditional
or edge-case instructions in order to surface all areas requiring integration, correction, reconstruction,
or adjudication.
- For each requirement, specify which elements from each response can be retained, which must be modified
or removed, and which response provides the stronger compliant material when overlap exists.  Define
what new content must be constructed to address missing requirements, and clarify how conflicts between
the responses or between a response and the instruction will be resolved.  When the approaches are
genuinely incompatible, specify which path will be followed and justify that decision based on instruction
fidelity.  If multiple interpretations remain valid due to genuine ambiguity in the instruction, specify
the compliant alternatives and the conditions under which each would apply.
- If both responses fail to adequately satisfy a required element, do not rely on their content.  Instead,
derive the requirement directly from the instruction and well-established knowledge, outline how that
element will be constructed from first principles without introducing unsupported assumptions.
- Organize the synthesis plan in a sequential, structured manner so that each planned component clearly
maps to a specific instruction requirement and the eventual response can follow the required order, format,
and style.  Ensure the plan provides proportional depth by specifying sufficient detail to fully satisfy
the instruction without introducing unnecessary expansion.  Avoid verbatim copying from either response
except where exact wording or formatting is explicitly required by the instruction.
- After completing the plan, verify that all requirements are addressed, integration decisions are
coherent, contradictions have been resolved, and no new violations have been introduced.

Query:  {query}

Responses:
Response A:
{y_i}

Response B:
{y_j}
```

## LLM-as-Mutation-Operator ($x_{\mathrm{mut}}$) — Instruction Following

**Purpose.** Produce a single final response guided by a structured synthesis plan derived from two parent responses.
**Inputs.** Prompt $x$, parents $y_i$, $y_j$, crossover draft $y^{\mathrm{cross}}$ (synthesis plan / reasoning analysis).
**Output.** A single final response.

**Template.**
```
You are impartial, precise, attentive to detail, and highly disciplined in rule adherence.  For the query
below, you have been provided with a structured synthesis plan derived from a requirement-by-requirement
instruction-compliant analysis of two AI-generated responses.  Your task is to use this plan as your
primary guide to produce a single final response that satisfies the user's instruction.

- Base the response on the synthesis plan so that the writing naturally reflects the decisions and
integrations already established.  Focus on translating the planned structure and content into a
clear, well-composed answer rather than revisiting the earlier comparison or synthesis process.  Do not
re-analyze the instruction or reassess the original responses; instead, rely on the plan as the foundation
for what should be communicated.
- Ensure the response addresses all requirements identified in the plan while preserving any mandated
format, ordering, tone, length, and prohibition rules.  Where the plan calls for reconstruction or newly
developed material, generate the necessary content directly from the instruction without introducing
unsupported assumptions or irrelevant additions.  Provide sufficient detail to fully satisfy the
instruction while avoiding unnecessary expansion.  If the synthesis plan is incomplete, incorporate any
clearly required instruction elements that are missing.  If the plan conflicts with the instruction,
follow the instruction.
- Allow the wording and structure to flow naturally so the response reads as a cohesive answer rather than
a procedural assembly.  Treat the synthesis plan as the authoritative foundation for the response while
maintaining flexibility in expression to support clarity, readability, and effective communication.  Avoid
introducing new approaches that were not implied by the plan unless doing so is necessary to maintain
coherence or instruction compliance.
```

```
- In addition to satisfying the plan's requirements, ensure the final response is high-quality along these
  dimensions where applicable:  accuracy (task correctness), factual correctness (no invented details
  or capabilities), relevance (stays on-topic and directly addresses the request), internal consistency
  (no contradictions), coherence (clear logical flow and readability), and safety/bias (no harmful,
  inappropriate, privacy-violating, insecure, or unfairly prejudicial content).
- Present the final response as a natural, user-facing reply written as if you are directly answering
  the user.  The response should read smoothly and conversationally while remaining precise and compliant
  with the instruction.  Maintain a helpful, clear, and professional tone without becoming overly formal or
  mechanical, and prioritize clarity so the user can immediately understand the answer.
- Output only the final response.  The response should stand alone as if it were written without any
  internal pipeline, and it should be seamless and unified using formatting that best supports clarity and
  aligns with the instruction.  Do not reference the analysis, synthesis plan, evaluation process, candidate
  responses, or how the response was created.  Do not include compliance notes, justification statements, or
  process-related commentary, and do not append any pipeline-related commentary or explanatory text after
  the response is complete.

Query:  {query}

Responses:
Response A:
{y_i}

Response B:
{y_j}

Reasoning analysis:  {y^cross}
```

## LLM-as-Refinement-Operator ($x_{\mathbf{refine}}$) — Instruction Following

**Purpose.** Produce a structured refinement (revision) plan for improving a single response under an Instruction Following objective.
**Inputs.** Prompt $x$, current response $y_t$.
**Output.** A structured refinement plan (no final answer).

**Template.**
```
You are impartial, precise, attentive to detail, and highly disciplined in rule adherence.  Your task
is to carefully evaluate how well a single response follows a user's instruction by identifying every
explicit requirement, constraint, and formatting rule, determining whether the response satisfies them,
and clearly noting any deviations, omissions, or unnecessary additions.  Correct any misinterpretations
of the instruction and specify how missing elements should be reconstructed when possible by producing a
structured revision plan that will be used to create a final response that most completely and accurately
fulfills what was explicitly asked in a subsequent step.

Step 1 - Critically and carefully analyze the user instruction and the response step by step in depth
before giving any judgment of instruction adherence.
-The instruction may contain multiple constraints, hidden requirements, conditional tasks, formatting
 rules, priorities, or prohibitions that are easy to overlook or misinterpret.  Begin by compiling a
 complete, detailed, and accurate breakdown of every explicit and reasonably implied requirement in the
 instruction, including required content, exclusions, structure, tone, length, ordering, scope, and output
 format, preserving their exact meaning and priority as written.  Do not infer intentions beyond what is
 clearly supported by the wording of the instruction, and do not introduce new requirements that were not
 stated or logically implied.
- When analyzing the response, do not assume by default that any part of it, including its interpretations
 of the instruction, implied constraints, formatting choices, omissions, or claims of compliance, is
 correct, complete, or reliable, as it may contain significant Instruction Following failures.  Any
 component of the response can be noncompliant, whether partially or entirely.  Even if the response
 appears well-structured, detailed, or persuasive, this does not imply it correctly followed the
 instruction.  Therefore, using the precise requirements from the instruction decomposition, verify the
 response's compliance carefully and diligently against each requirement, identifying strengths, weaknesses,
 missing elements, unnecessary additions, and constraint violations.
- Evaluate the response for faithful reading of the instruction; correct handling of constraints and
 prohibitions; proper ordering and structure; correct formatting; appropriate tone and style; adherence to
 any length or scope limits; and accuracy, factual correctness, relevance, internal consistency, coherence,
 and safety/bias.  Check every part for hidden assumptions about what the user wanted, misinterpretations,
 added unstated requirements, unnecessary content that violates constraints, missing required elements,
 contradictions with the instruction, and gaps in coverage.  Continuously cross-reference each part of the
```

```
response against the instruction's exact requirements.

Step 2 - Using the requirement-by-requirement instruction-compliant analysis as your primary reference,
develop a structured refinement plan that specifies how the best instruction-compliant response will be
produced.  Preserve all mandatory format, ordering, tone, length, and prohibition rules defined by the
instruction.  Identify any noncompliant material that must be removed or revised, resolve conflicts by
prioritizing the instruction's explicit constraints, and ensure the plan accounts for every required
element.
- Systematically evaluate the response on a requirement-by-requirement basis to determine where it
aligns with or diverges from the instruction in its interpretation and satisfaction of the requirements.
Examine how the response handles constraints and prohibitions, required content, ordering and structure,
formatting rules, tone and style requirements, length and scope limits, and any conditional or edge-case
instructions in order to surface all areas requiring correction, reconstruction, or clarification.
- For each requirement, specify which elements of the response can be retained, which must be modified
or removed, and what new content must be constructed to address missing requirements.  Clarify how any
conflicts between the response and the instruction will be resolved.  If multiple interpretations remain
valid due to genuine ambiguity in the instruction, specify the compliant alternatives and the conditions
under which each would apply.
- If the response fails to adequately satisfy a required element, do not rely on its content.  Instead,
derive the requirement directly from the instruction and well-established knowledge, outline how that
element will be constructed from first principles without introducing unsupported assumptions.
- Organize the refinement plan in a sequential, structured manner so that each planned component clearly
maps to a specific instruction requirement and the eventual response can follow the required order, format,
and style.  Ensure the plan provides proportional depth by specifying sufficient detail to fully satisfy
the instruction without introducing unnecessary expansion.  Avoid verbatim copying from either response
except where exact wording or formatting is explicitly required by the instruction.
- After completing the plan, verify that all requirements are addressed, the planned revisions are
coherent, contradictions have been resolved, and no new violations have been introduced.

Query:  {query}

Response:
{y_t}
```

## LLM-as-Perturbation-Operator ($x_{\text{pert}}$) — Instruction Following

**Purpose.** Produce a single final response guided by a structured refinement plan derived from one response.
**Inputs.** Prompt $x$, current response $y_t$, refinement draft $y_t^{\text{refine}}$ (refinement plan / reasoning analysis).
**Output.** A single candidate response.

**Template.**
```
You are impartial, precise, attentive to detail, and highly disciplined in rule adherence.  For the query
below, you have been provided with a structured refinement plan derived from a requirement-by-requirement
instruction-compliant analysis of an AI-generated response.  Your task is to use this plan as your primary
guide to produce the final response that satisfies the user's instruction.

- Base the response on the refinement plan while allowing for natural language flow and readability.
Follow the plan faithfully, but apply reasonable judgment when expressing the content so the result reads
as a clear and well-composed answer rather than a mechanical execution of instructions.  Do not re-analyze
the instruction or reassess the original response; instead, focus on translating the planned decisions
into a coherent response.
- Ensure that all requirements identified in the plan are addressed and that the response preserves any
mandated format, ordering, tone, length, and prohibition rules.  Where the plan calls for reconstruction,
generate the necessary content directly from the instruction without introducing unsupported assumptions
or irrelevant additions.  Avoid unnecessary expansion, but provide enough detail to fully satisfy the
instruction.  If the refinement plan is incomplete, incorporate any clearly required instruction elements
that are missing.  If the plan conflicts with the instruction, follow the instruction.
- Treat the plan as the authoritative foundation for the response while maintaining flexibility in wording
and structure to produce a natural result.  The goal is faithful implementation with smooth communication,
not rigid adherence that harms clarity or usability.
- In addition to satisfying the plan's requirements, ensure the final response is high-quality along these
dimensions where applicable:  accuracy (task correctness), factual correctness (no invented details
or capabilities), relevance (stays on-topic and directly addresses the request), internal consistency
(no contradictions), coherence (clear logical flow and readability), and safety/bias (no harmful,
inappropriate, privacy-violating, insecure, or unfairly prejudicial content).
```

```
- Present the final response as a natural, user-facing reply written as if you are directly answering
  the user.  The response should read smoothly and conversationally while remaining precise and
  instruction-compliant.  Maintain a helpful, clear, and professional tone without becoming overly formal
  or mechanical, and prioritize clarity so the user can immediately understand the answer.
- Output only the final response.  The response should stand alone as if it were written without any
  internal pipeline, and it should be seamless and unified using formatting that best supports clarity and
  aligns with the instruction.  Do not reference the analysis, synthesis plan, evaluation process, candidate
  responses, or how the response was created.  Do not include compliance notes, justification statements, or
  process-related commentary, and do not append any pipeline-related commentary or explanatory text after
  the response is complete.

Query:
{query}

Response:
{y_t}

Reasoning analysis:
{$y_t^{\text{refine}}$}
```

## D.4  GPT-Judge Prompt Templates

**GPT-Judge Instructions (Instruction Following Comparison)**

**Purpose.** Compare two candidate responses and select the one that follows the user instruction more accurately.
**Inputs.** User query $x$, Candidate A, Candidate B.
**Output.** JSON with keys `winner`, `confidence`, `notes`.

```
Template.
You are an impartial judge evaluating two candidate responses to a user query.

You will be given:
- INPUT (human user query)
- Candidate A response
- Candidate B response

Ignoring factors unrelated to instruction following, compare all instruction following aspects of the
two responses and determine which one follows the user's instructions more accurately and completely.

Decision rule:  - Ignore response order and judge independently.
- Choose "A" or "B" if one response is clearly better.
- Choose "TIE" only if the two responses are effectively indistinguishable in all aspects of
  instruction-following quality.

Output MUST be valid JSON with exactly these keys:
- winner:  "A", "B", or "TIE".
- confidence:  integer 0-100 (higher if the difference is obvious).
- notes:  <= 2 sentences, focusing on why the winner is clearer/cleaner.
```

# E  Evaluation Metrics

## E.1  Evaluation metrics for the Mathematical Reasoning experiment 4.1

We evaluate mathematical reasoning in two settings: (i) *automatic verification*, where candidate solutions can be automatically verified against ground-truth answers, and (ii) *proxy-based selection*, where ground truth is unavailable and selection must rely on learned proxy signals or agreement-based heuristics. We additionally report *pairwise preference* evaluations to assess response quality beyond correctness.

### E.1.1 Automatic Verification

When solutions admit automatic verification, we evaluate correctness coverage by estimating the probability that at least one sampled response matches the ground-truth answer.

**Pass@$k$.** Pass@$k$ is an evaluation metric for tasks with a well-defined ground-truth answer that can be obtained automatically. Given a prompt $x$, we draw $k$ candidate responses

$$y_i \sim \pi(y \mid x), \quad i = 1, \ldots, k,$$

using a fixed decoding procedure. Let $a^{\text{gt}}(x)$ denote the ground-truth final answer for $x$, and let $f : \mathcal{Y} \to \mathcal{A}$ be a task-specific answer extraction function. Pass@$k$ measures the probability that at least one sampled response yields the correct final answer:

$$\text{Pass@}k \;=\; \mathbb{P}\big(\exists i \in \{1, \ldots, k\} \text{ such that } f(y_i) = a^{\text{gt}}(x)\big).$$

### E.1.2 Proxy-Based Selection

When automatic verification is unavailable, human evaluation is often required but quickly becomes impractical at scale. We therefore select responses from sampled candidates using learned or agreement-based proxy signals intended to correlate with correctness and solution quality.

**Best RM.** Given a prompt $x$, we draw $N$ candidate responses

$$y_i \sim \pi(y \mid x), \quad i = 1, \ldots, N,$$

and score each candidate using a reward model $r(x, y)$. The Best RM score is

$$\text{BestRM}(x; N) \;=\; \max_{i \in \{1, \ldots, N\}} r(x, y_i).$$

We report the mean of $\text{BestRM}(x; N)$ across prompts with two-sided 95% confidence intervals.

**Best-of-$N$ sampling.** Best-of-$N$ sampling is a post-decoding strategy that selects the highest-reward response from a finite set of samples. Given a prompt $x$, we draw $N$ candidate responses

$$y_i \sim \pi(y \mid x), \quad i = 1, \ldots, N,$$

using a fixed decoding procedure. The Best-of-$N$ response is defined as

$$\hat{y}_{\text{BoN}} \;=\; \arg\max_{i \in \{1, \ldots, N\}} r(x, y_i).$$

In evaluation, we report Best-of-$N$ accuracy, defined as the accuracy of the selected response $\hat{y}_{\text{BoN}}$.

**Self-consistency.** As an alternative that does not rely on an RM, we draw $N$ candidate responses

$$y_i \sim \pi(y \mid x), \quad i = 1, \ldots, N,$$

using a fixed decoding procedure. Let $f : \mathcal{Y} \to \mathcal{Z}$ denote a task-specific answer extraction function. Self-consistency selects the response whose extracted answer is most frequent among the samples:

$$\hat{y}_{\text{SC}} \;=\; \arg\max_{i \in \{1, \ldots, N\}} \sum_{j=1}^{N} \mathbf{1}[f(y_i) = f(y_j)].$$

In evaluation, we report Self-consistency accuracy, defined as the accuracy of the selected response $\hat{y}_{\text{SC}}$.

## E.2 Evaluation metrics for the Instruction Following experiment 4.2

We define the reward-based and preference metrics used in the instruction-following experiments.

**Marginal gain.** Let $N_g$ denote the cumulative number of candidates available after generation $g$. Define

$$\text{best}(x, g) \;=\; \max_{i \in \{1, \dots, N_g\}} r(x, y_i).$$

The marginal gain at generation $g$ is

$$\Delta_g(x) \;=\; \text{best}(x, g) - \text{best}(x, g - 1).$$

We report the mean of $\Delta_g(x)$ across prompts with percentile bootstrap confidence intervals.

**Preference win rate.** For each prompt, we compare a selected response (system $B$) against a baseline response (system $A$) using a panel of judges and evaluate both presentation orders (A/B and B/A). Each judge outcome is mapped back to the system identities and scored as

$$z = \begin{cases} 1 & \text{if } B \text{ is preferred,} \\ 0 & \text{if } A \text{ is preferred,} \\ \frac{1}{2} & \text{if tied.} \end{cases}$$

The per-prompt preference score is the mean judge score across all votes, and the reported Preference is the mean of this score across prompts. We compute 95% confidence intervals using a hierarchical bootstrap that resamples prompts and, within each prompt, resamples judge votes.

**Preference margin.** We define the tie-aware Margin as a symmetric transformation of Preference:

$$\text{Margin} \;=\; 2\,\text{Preference} - 1,$$

and report it with 95% hierarchical bootstrap confidence intervals (converting to percentage points when reporting in pp).

## F   Complementary Analysis of Optimization Trajectories

### F.1   Mathematical Reasoning (AMC 12 2025)

We present complementary optimization trajectory analyses of ANNETRON, GENETRON, MEMETRON, and MEMETRON with correctness shaping for the AMC 12 2025 experiment reported in Subsection 4.1.

To understand the behavior of the optimization process, we analyze the trajectory of Best RM scores across generations, examining marginal gains and improvement across prompt difficulty levels. We approximate prompt difficulty using the Base-16 RM score, where prompts yielding lower Base-16 scores are considered harder, as the base model produces lower-quality responses for these prompts prior to optimization. Figure 4 presents these views for all three methods.

All three methods share a consistent pattern: the largest marginal gain occurs at generation 1, after which improvements diminish rapidly. MEMETRON achieves the highest first-generation gain ($\Delta = 7.75$), followed by GENETRON ($\Delta = 3.83$) and ANNETRON ($\Delta = 2.61$), demonstrating that the integration of GENETRON and ANNETRON within MEMETRON yields coherent and additive optimization gains.

Across all methods, harder prompts benefit more from additional generations. ANNETRON shows Q1 gaining 9.34 points versus 4.45 points for Q4; GENETRON shows Q1 gaining 12.00 points versus 6.55 points for Q4; and MEMETRON shows the largest overall gains, with Q1 improving by 15.70 points and Q4 by 8.30 points. This consistent pattern suggests that MEMETRON and its component optimizers are most beneficial precisely where the base model struggles the most.

Figure 5 presents the optimization trajectory for MEMETRON with correctness shaping. The marginal gain pattern remains similar to vanilla MEMETRON, with a first-generation gain of $\Delta = 7.62$, confirming that the additive correctness bonus does not alter the optimization dynamics. Comparing the generation-level gains,

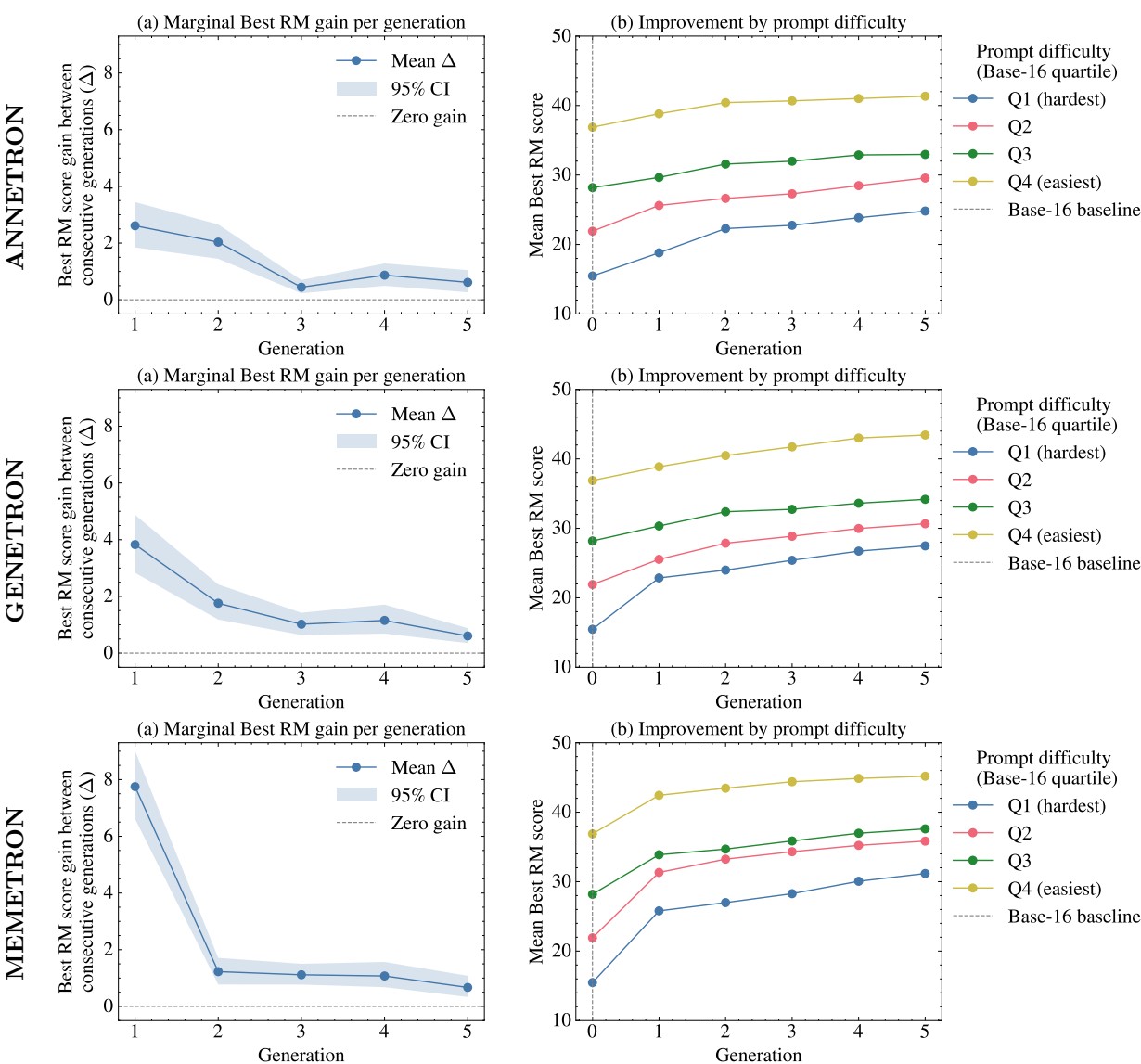

Figure 4: Optimization trajectory across generations for ANNETRON, GENETRON, and MEMETRON on AMC 2025 mathematical reasoning prompts using non-thinking Qwen-3-8B. **(a)** Marginal Best RM score gain between consecutive generations ($\Delta$) with 95% confidence intervals (shaded). Positive values above the zero-gain line (dashed) indicate continued improvement. **(b)** Mean Best RM score across generations stratified by prompt difficulty quartile, defined by Base-16 score. Q1 contains the hardest 25% of prompts and Q4 the easiest. The dashed vertical line marks the Base-16 baseline prior to optimization.

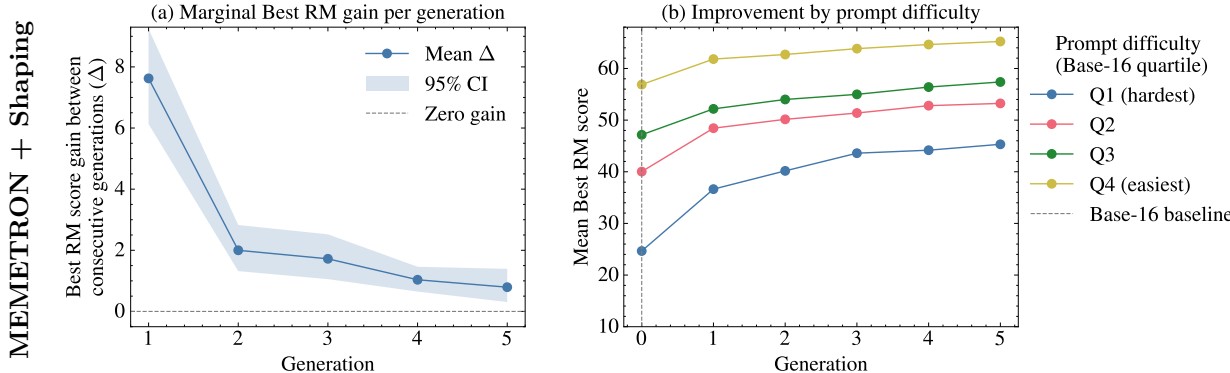

Figure 5: Optimization trajectory across generations for MEMETRON with correctness shaping on AMC 2025 mathematical reasoning prompts using non-thinking Qwen-3-8B. **(a)** Marginal Best RM score gain between consecutive generations ($\Delta$) with 95% confidence intervals (shaded). Positive values above the zero-gain line (dashed) indicate continued improvement. **(b)** Mean Best RM score across generations stratified by prompt difficulty quartile, defined by Base-16 score. Q1 contains the hardest 25% of prompts and Q4 the easiest. The dashed vertical line marks the Base-16 baseline prior to optimization.

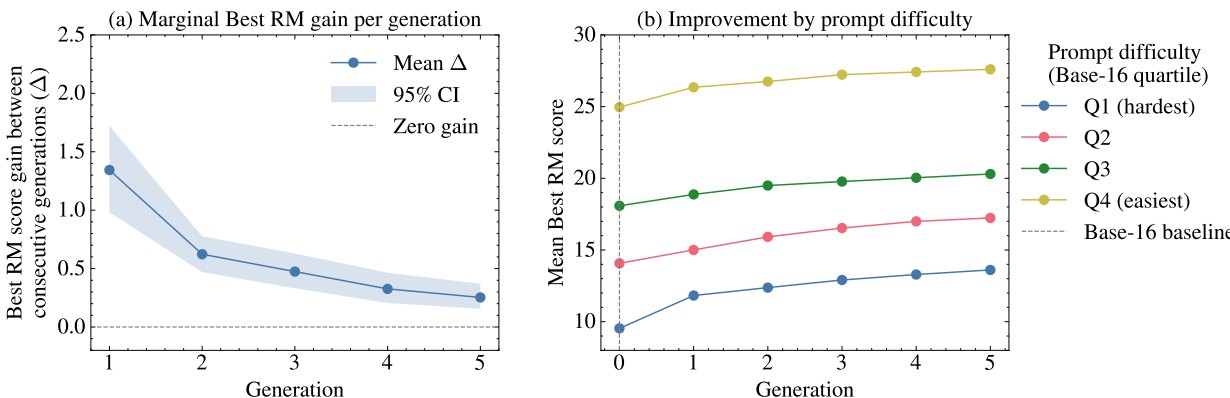

Figure 6: Optimization trajectory across generations for MEMETRON on 100 TinyAlpacaEval Instruction Following prompts using Llama-3.2-3B-Instruct. **(a)** Marginal Best RM score gain between consecutive generations ($\Delta$) with 95% confidence intervals (shaded). Positive values above the zero-gain line (dashed) indicate continued improvement. **(b)** Mean Best RM score across generations stratified by prompt difficulty quartile, defined by Base-16 score. Q1 contains the hardest 25% of prompts and Q4 the easiest. The dashed vertical line marks the Base-16 baseline prior to optimization.

which are invariant to the constant reward shift, reveals that correctness shaping disproportionately benefits harder prompts: Q1 achieves a gain of 20.66 points compared to 15.70 points for vanilla MEMETRON, while Q4 remains largely unchanged (8.34 vs. 8.30 points). This suggests that MEMETRON progressively improves the Best RM score across generations for all difficulty levels, and correctness shaping amplifies this effect for harder prompts by providing an additional reward signal when correct solutions are found.

## F.2 Instruction Following

We present complementary optimization trajectory analysis of MEMETRON for the instruction following experiment reported in Section 4.2.

Figure 6 presents the optimization trajectory for MEMETRON on the instruction following task. The pattern is consistent with the mathematical reasoning experiments: the largest marginal gain occurs at generation 1 ($\Delta = 1.34$), after which improvements diminish gradually but remain positive throughout. Across all diffi-

culty levels, the Best RM score improves monotonically across generations, with harder prompts benefiting more overall: Q1 achieves an absolute gain of 4.08 points compared to 2.64 points for Q4, suggesting that MEMETRON is most beneficial where the base model struggles the most, consistent with the findings on mathematical reasoning.

## G   Computational Cost

Table 4 reports the computational cost of every system in the AMC 12 2025 experiment of Subsection 4.1.

We define a *sub-step* as a single ANNETRON iteration or GENETRON generation. This is the atomic unit of search: all three methods are built from it. A *request* is one batched call, returning many completions at once. A sub-step issues two generator requests: the first produces $N$ drafts, the second samples $n$ candidates per draft. It therefore generates $N(1 + n)$ completions in total. One RM request scores the $Nn$ candidates; the drafts themselves are never scored. For each draft, the highest-scoring of its $n$ candidates is selected, and these $N$ responses are carried forward.

A *step* is the unit indexed by the rows of Table 4, and $g$ denotes the number of completed steps. A step consists of $s$ sub-steps and appends to $\mathcal{H}$ the $N$ best responses found across them. For standalone ANNETRON and GENETRON, a step coincides with a single sub-step, so $s = 1$. A MEMETRON step, corresponding to one MEMETRON generation in Tables 1 and 2, comprises one GENETRON generation followed by five ANNETRON iterations, so $s = 1 + 5 = 6$. ANNETRON therefore has different buffer behavior in the two settings: standalone, each iteration is its own step and appends $N$ responses to $\mathcal{H}$; within MEMETRON, its five iterations are sub-steps of a single step and contribute $N$ responses in total.

Counting the shared Base-$N$ initialization as one generator request, one RM request, and $N$ completions, all quantities in Table 4 follow. With $N = 16$ and $n = 3$ in this experiment:

$$|\mathcal{H}| = N(1 + g) = 16 + 16g,$$
$$\text{completions generated} = N\big(1 + (1 + n)sg\big) = 16 + 64sg,$$
$$\text{candidates scored} = N(1 + nsg) = 16 + 48sg,$$
$$\text{generator requests} = 1 + 2sg,$$
$$\text{RM requests} = 1 + sg.$$

Note that $|\mathcal{H}|$ is the only quantity that does not scale with $s$: a MEMETRON step issues six times the calls of a GENETRON step while growing $\mathcal{H}$ by the same amount. Wall-clock time is also linear in $g$: the variation operators condition on selected parents rather than on the full buffer, so prompt length is independent of $|\mathcal{H}|$ and per-sub-step cost is constant.

Table 4: Cumulative computational cost per step in the AMC 12 2025 experiment (Subsection 4.1). *Init* is the sampling cost of the initial pool (for RPDO methods, the shared Base-16 initialization, identical across methods); *Search* is the cumulative cost of completed steps, and *Total* their sum. Wall-clock time is measured end-to-end over the full five-step run; time per step does not depend on $g$, so intermediate rows ($\sim$) scale the measured per-step average. Completion, candidate, and request counts are exact and follow the formulas above. $|\mathcal{H}|$ records only retained responses and is not a compute budget.

| System | Init (s/q) | Search (s/q) | Total (s/q) | Completions Generated | Candidates Scored | Requests (Gen / RM) | $|\mathcal{H}|$ |
|---|---|---|---|---|---|---|---|
| Base-16 | 34.19 | — | 34.19 | 16 | 16 | 1 / 1 | 16 |
| Base-128 | 93.13 | — | 93.13 | 128 | 128 | 1 / 1 | 128 |
| Base-1024 | 601.13 | — | 601.13 | 1024 | 1024 | 1 / 1 | 1024 |
| ANNETRON (Base-16 + 1 iteration) | 34.19 | $\sim$86.86 | $\sim$121.05 | 80 | 64 | 3 / 2 | 32 |
| ANNETRON (Base-16 + 2 iterations) | 34.19 | $\sim$173.72 | $\sim$207.91 | 144 | 112 | 5 / 3 | 48 |
| ANNETRON (Base-16 + 3 iterations) | 34.19 | $\sim$260.59 | $\sim$294.78 | 208 | 160 | 7 / 4 | 64 |
| ANNETRON (Base-16 + 4 iterations) | 34.19 | $\sim$347.45 | $\sim$381.64 | 272 | 208 | 9 / 5 | 80 |
| ANNETRON (Base-16 + 5 iterations) | 34.19 | 434.31 | 468.50 | 336 | 256 | 11 / 6 | 96 |
| GENETRON (Base-16 + 1 generation) | 34.19 | $\sim$98.22 | $\sim$132.41 | 80 | 64 | 3 / 2 | 32 |
| GENETRON (Base-16 + 2 generations) | 34.19 | $\sim$196.44 | $\sim$230.63 | 144 | 112 | 5 / 3 | 48 |
| GENETRON (Base-16 + 3 generations) | 34.19 | $\sim$294.66 | $\sim$328.85 | 208 | 160 | 7 / 4 | 64 |
| GENETRON (Base-16 + 4 generations) | 34.19 | $\sim$392.88 | $\sim$427.07 | 272 | 208 | 9 / 5 | 80 |
| GENETRON (Base-16 + 5 generations) | 34.19 | 491.10 | 525.29 | 336 | 256 | 11 / 6 | 96 |
| MEMETRON (Base-16 + 1 generation) | 34.19 | $\sim$463.69 | $\sim$497.88 | 400 | 304 | 13 / 7 | 32 |
| MEMETRON (Base-16 + 2 generations) | 34.19 | $\sim$927.38 | $\sim$961.57 | 784 | 592 | 25 / 13 | 48 |
| MEMETRON (Base-16 + 3 generations) | 34.19 | $\sim$1391.07 | $\sim$1425.26 | 1168 | 880 | 37 / 19 | 64 |
| MEMETRON (Base-16 + 4 generations) | 34.19 | $\sim$1854.76 | $\sim$1888.95 | 1552 | 1168 | 49 / 25 | 80 |
| MEMETRON (Base-16 + 5 generations) | 34.19 | 2318.45 | 2352.64 | 1936 | 1456 | 61 / 31 | 96 |
| MEMETRON + Shaping (Base-16 + 1 generation) | 34.19 | $\sim$476.30 | $\sim$510.49 | 400 | 304 | 13 / 7 | 32 |
| MEMETRON + Shaping (Base-16 + 2 generations) | 34.19 | $\sim$952.59 | $\sim$986.78 | 784 | 592 | 25 / 13 | 48 |
| MEMETRON + Shaping (Base-16 + 3 generations) | 34.19 | $\sim$1428.89 | $\sim$1463.08 | 1168 | 880 | 37 / 19 | 64 |
| MEMETRON + Shaping (Base-16 + 4 generations) | 34.19 | $\sim$1905.18 | $\sim$1939.37 | 1552 | 1168 | 49 / 25 | 80 |
| MEMETRON + Shaping (Base-16 + 5 generations) | 34.19 | 2381.48 | 2415.67 | 1936 | 1456 | 61 / 31 | 96 |

## H  Implications

MEMETRON provides a unified mechanism for reward-guided optimization over complete LLM responses without modifying model parameters or decoding procedures. Because optimization is formulated purely at the response level and relies only on black-box reward evaluations, the same search process can be deployed

both at inference time to improve final outputs and during off-policy training to generate higher-quality training candidates. Moreover, when verifiable correctness signals are available, search-based optimization naturally exposes reward-correctness mismatches, enabling systematic analysis and mitigation of reward misalignment.

### H.1 Inference-Time Response Optimization

At inference time, model parameters are fixed, and performance can only be improved by modifying how candidate responses are explored and selected. Rather than relying on independent sampling followed by one-shot selection as in Best-of-$N$ and Self-consistency, MEMETRON performs structured exploration and iterative refinement of complete responses guided by the reward function, returning

$$\hat{y} = \arg\max_{y \in \mathcal{H}(x)} r(x, y),$$

where $\mathcal{H}(x)$ denotes the set of candidates explored during search. Because optimization operates entirely at the response level and requires only black-box reward evaluations, MEMETRON can be deployed as a post-generation layer on top of any frozen LLM and reward function, without modifying model parameters or token-level decoding procedures.

### H.2 Training-Time Response Optimization

MEMETRON can also be used during off-policy training as a response-level search operator that improves how candidate responses are generated. Given a prompt $x$, a policy $\pi_\theta$, and a reward function $r$, MEMETRON produces a set of evaluated candidate responses

$$\mathcal{H}(x) = \text{MEMETRON}(x, \pi_\theta, r),$$

together with their associated reward scores $\{r(x, y)\}_{y \in \mathcal{H}(x)}$. The Best-of-search response $\hat{y} = \arg\max_{y \in \mathcal{H}} r(x, y)$ can then be extracted directly from this candidate set.

**SFT with reward-optimized targets.** Instead of sampling $y \sim \pi_\theta(y \mid x)$ and filtering, we train on the approximate solution $\hat{y}$ produced by MEMETRON, using the standard supervised objective

$$\mathcal{L}_{\text{SFT}}(\theta) = -\mathbb{E}_{x \sim \mathcal{D}} \log \pi_\theta(\hat{y} \mid x).$$

This replaces shallow generation-and-filtering with reward-guided search for constructing training targets, with $\hat{y}$ treated as a fixed target during optimization. Since RL pipelines such as PPO and GRPO typically rely on an SFT warmup stage to initialize the policy, MEMETRON can also improve this initialization by supplying higher-reward training targets, leading to a stronger starting point before RL training begins.

**DPO with search-induced preference pairs.** From the candidate pool $\mathcal{H}(x)$, we construct preference pairs by selecting a high-reward and a low-reward response,

$$y^+ \in \text{Top}_k(\mathcal{H}(x)), \qquad y^- \in \text{Bottom}_k(\mathcal{H}(x)),$$

inducing a preference dataset $\mathcal{P} = \{(x, y^+, y^-)\}$ used in DPO via

$$\mathcal{L}_{\text{DPO}}(\theta) = -\mathbb{E}_{(x, y^+, y^-) \sim \mathcal{P}} \log \sigma\Big( \beta\big( \log \pi_\theta(y^+ \mid x) - \log \pi_\theta(y^- \mid x) \big) \Big).$$

MEMETRON thus supplies informative preferred and dispreferred responses drawn from structured search, while leaving the DPO objective unchanged.

### H.3 Reward Misalignment Analysis and Mitigation in Verifiable Tasks

In tasks with verifiable final answers, a learned RM may assign high scores to responses that are incorrect or low-utility, due to limited model capacity, distribution shift, or systematic biases in how surface features are

scored. In these cases, reward-guided selection may fail to identify the best candidate even when high-quality solutions are present in the explored set.

We address reward misalignment through a search-driven diagnostic and mitigation workflow centered on correctness shaping, covering three steps: detecting misalignment through coverage-selection discrepancies amplified by search-based optimization, isolating misranking failures using correctness-conditioned reward shaping, and mitigating misalignment via targeted reward model refinement. Crucially, MEMETRON performs search-based discrete optimization rather than independent sampling, which actively concentrates candidates in high-reward regions of the response space, amplifying discrepancies between reward scores and true task outcomes and making misalignment easier to surface and analyze.

**Exposing and Detecting Misalignment via Selection Gaps under Search.** We detect reward misalignment by comparing coverage-oriented metrics such as Pass@$n$, which indicate whether correct solutions are present among explored candidates, with selection-based metrics such as Best-of-$N$, which measure whether the RM successfully selects them. When correct solutions exist but are not selected, this indicates reward model misranking. Search-based optimization amplifies such discrepancies by intentionally concentrating candidates in high-reward regions, producing more challenging candidate sets that stress-test the RM and make selection failures easier to quantify.

**Correctness Shaping for Isolating Failures and Optimizing Correct Solutions.** To isolate reward model misranking and guide search toward correct solutions, we introduce correctness-conditioned reward shaping, which augments the base reward with an explicit correctness signal:

$$r'(x, y) = r(x, y) + c \cdot \mathbb{1}[\text{answer}(y) \text{ is correct}],$$

where $c > 0$ is a constant bonus applied to correct responses. Running search under both the base reward $r$ and the shaped reward $r'$ induces two complementary candidate distributions: responses that achieve high reward under $r$ but are incorrect, and responses that achieve high reward under $r'$ while satisfying correctness constraints. Because both sets are generated using the same search operators, correctness shaping isolates the effect of answer recognition while controlling for search dynamics, and provides a practical oracle-assisted upper reference for achievable selection performance.

**Mitigating Reward Misalignment via Search-Generated Preference Data.** Correctness shaping naturally yields informative preference pairs $(y^+, y^-)$, where $y^+$ denotes a correct response selected under $r'$ and $y^-$ denotes an incorrect response that attains high reward under the base reward $r$. These pairs directly expose cases where the learned RM favors incorrect but highly scored responses.

Such search-generated preference data can be used to refine the RM using standard preference-based objectives such as the Bradley-Terry loss. For a parameterized RM $r_\phi$, the loss for a single pair is

$$\mathcal{L}_{\text{BT}}(\phi) = -\log \sigma\big(r_\phi(x, y^+) - r_\phi(x, y^-)\big),$$

which encourages the RM to assign higher scores to correct, high-quality responses than to incorrect but highly scored ones. Once reward model ranking is improved through such updates, the same high-quality responses can be reused for downstream policy optimization using standard alignment pipelines.

