# OpenReview forum: "MEMETRON: Memetic Response Optimizer for Reward-Guided Post-Decoding Optimization of Large Language Models"
_TMLR — Accepted by TMLR_

### Review · Reviewer_4Kkc · 2026-03-19

**Summary Of Contributions:**

The paper casts reward-guided post-decoding optimization (RPDO) as discrete black-box optimization over finished LLM responses and introduces MEMETRON, which interleaves a GA-style population search (GENETRON) with an SA-style local refinement (ANNETRON), both driven by frozen LLMs as operators. Experiments cover 42 AMC 12 2025 math problems (Qwen3-8B + Skywork-Reward-V2-8B) and 100 TinyAlpacaEval prompts (Llama-3.2-3B-Instruct + Skywork-Reward-V2-3B). On both tasks, MEMETRON pushes RM scores above pure-sampling baselines and shows gains in best-of-N accuracy, self-consistency, and LLM-judge preference.

**Additional Comments:**

See above.

**Audience:**

Yes

**Audience Explanation:**

This paper explores the evolution algorithm based LLM post-training paradigms.

**Claims And Evidence:**

Yes

**Claims Explanation:**

## Strengths

1. Framing best-of-N, rejection-sampling SFT, and rejection-sampling DPO as shallow special cases of a single optimization objective is a neat unification that clarifies what "going deeper" means in this space.

2. The correctness-shaping diagnostic is a useful idea, and the contrastive pairs it produces could feed into RM fine-tuning or DPO pipelines.

## Weaknesses

### W1. Missing baselines

Mind Evolution and LLMRefine are discussed at length in Section 1 as the two closest methods: "Mind Evolution emphasizes population-based optimization via genetic algorithms"; "LLMRefine targets open-ended tasks by iteratively refining responses with simulated annealing." Yet neither shows up in any experiment. Every comparison is against naive sampling (Base-16/128/1024). The paper defines itself against these two methods but did not actually tests against them.

### W2. Novelty concern

Strip away the naming and what remains is: run a genetic algorithm with LLM-based crossover/mutation (≈ Mind Evolution), run simulated annealing with LLM-based perturbation (≈ LLMRefine), and alternate between the two is the textbook definition of a memetic algorithm. LLM-as-evolutionary-operator is not new either (Mind Evolution, EvoPrompt, LLM-as-Optimizer, etc.).

### W3. Deep search aggravates the reward-hacking problem it claims to address

Section 1 motivates the work by arguing that wide shallow sampling "exacerbates reward hacking, making downstream selection methods such as best-of-N and self-consistency unreliable." But the paper's own results cut the other way. The authors note that "RPDO finds extreme high-reward but incorrect responses beyond the reach of pure sampling." Concretely, MEMETRON at generation 5 achieves Best RM = 37.48 vs. 32.92 for Base-1024, while a ~10-point gap persists between Pass@|H| (95.24%) and Best-of-|H| (85.71%) — for 4 out of 40 solvable problems, the RM's top-ranked answer is wrong. Deeper search appears to make the RM easier to exploit, not harder. The correctness-shaping fix requires oracle labels, which sidesteps rather than solves the issue.

### W4. Reliance on strong base model.

The entire search happens over outputs of a frozen base model. When the base model simply cannot produce a good answer, more search will not help — you cannot find what is not in the space. In the reported experiments, the generator is already strong enough that Base-1024 hits 100% Pass@|H| on the math benchmark. This raises a practical question: how weak can the base model be before MEMETRON stops paying off? And if the method only works well when the model is already nearly competent, the marginal value of ~39 min/problem of extra search becomes hard to justify. The paper would benefit from experiments on tasks or models where the generator is clearly weaker.

## Questions

1. Could you run Mind Evolution and LLMRefine (or faithful re-implementations) under the same compute budget and report numbers?
2. Beyond the GA+SA combination: what specific operator design or scheduling decision in MEMETRON is new and could not be trivially assembled from prior work?
3. The data show MEMETRON finds high-reward wrong answers that pure sampling never reaches. How should a practitioner think about deploying this without oracle labels?
4. Have you tried MEMETRON on a task where the base model's Pass@1024 is, say, below 50%? What happens?

**Requested Changes:**

See above.

---

> ### Author Response · Authors · 2026-05-06
> **Response to Reviewer 4Kkc of Paper7799: Opening & Overview**
>
> We thank the reviewer for their time and review, and for recognizing the conceptual contributions of our framework. We address the raised concerns below.

---

> ### Author Response · Authors · 2026-05-06
> **Response to Reviewer 4Kkc of Paper7799: W1**
>
> **Mind Evolution and LLMRefine are discussed at length in Section 1 as the two closest methods: "Mind Evolution emphasizes population-based optimization via genetic algorithms"; "LLMRefine targets open-ended tasks by iteratively refining responses with simulated annealing." Yet neither shows up in any experiment. Every comparison is against naive sampling (Base-16/128/1024). The paper defines itself against these two methods but did not actually tests against them.**
>
> We thank the reviewer for this comment. We explicitly discuss and contrast our work with MindEvolution (Lee et al., 2025) and LLMRefine (Xu et al., 2024) in our paper's introduction (Section 1, Paragraph 4, Page 2, beginning "Recent...") and in our related work (Section A.4, Page 19), noting that both are constrained by their dependence on either task-specific programmatic verifiers or task-specific feedback models, limiting their applicability across general settings such as mathematical reasoning and instruction following tasks that we evaluate in this work. Direct comparison is not feasible as their designs do not generalize beyond these specific requirements, and no public code is available for either method.
>
> Instead, we designed GENETRON and ANNETRON explicitly based on the same core principles that these works are inspired from, as described in Section 3.1 (GENETRON: Genetic Optimization for RPDO) and Section 3.2 (ANNETRON: Simulated Annealing for RPDO), generalized to work with any blackbox scalar reward signal. GENETRON and ANNETRON are explicitly included in our experiments in Table 1 (Page 11), serving as the closest feasible experimental stand-ins for MindEvolution and LLMRefine respectively. Furthermore, we propose MEMETRON, which combines the strengths of both, drawing on the memetic idea of integrating global and local search for RPDO, representing a more powerful and principled unification of these approaches.

---

> ### Author Response · Authors · 2026-05-06
> **Response to Reviewer 4Kkc of Paper7799: W2 Novelty concern**
>
> **Strip away the naming and what remains is: run a genetic algorithm with LLM-based crossover/mutation (≈ Mind Evolution), run simulated annealing with LLM-based perturbation (≈ LLMRefine), and alternate between the two is the textbook definition of a memetic algorithm. LLM-as-evolutionary-operator is not new either (Mind Evolution, EvoPrompt, LLM-as-Optimizer, etc.).**
>
> The reviewer's characterization of our work as simply combining a genetic algorithm and simulated annealing is oversimplified and overlooks our actual contributions.
>
> First, we note that TMLR's own acceptance criteria explicitly state that novelty of the studied method is not a necessary criterion for acceptance, and that rejecting work on the basis of it not being "novel enough" is discouraged (https://jmlr.org/tmlr/acceptance-criteria.html). That said, our contributions extend well beyond the algorithmic combination the reviewer describes:
>
> We introduce RPDO as a formal discrete black-box optimization framework over completed responses under any blackbox scalar reward signal, defined in Section 1 (Introduction and Contribution 1) and fully formalized in Section 2 (Problem Formulation), that unifies existing methods such as best-of-N, SFT with rejection sampling, and DPO with rejection sampling as shallow, one-shot instantiations of a single objective, exposing a principled gap in how existing methods exploit the reward signal.
>
> Building on this framework, we introduce GENETRON (Section 3.1), ANNETRON (Section 3.2), and MEMETRON (Section 3) as concrete instantiations, where frozen LLMs serve as search operators prompted to align with the reward objective without access to raw reward scores, encouraging semantic improvement of response quality rather than exploitation of score artifacts.
>
> Furthermore, our reward misalignment diagnostics via correctness shaping, detailed in Contribution 4 (Page 3) and Section 4.1 (Page 10, last paragraph), and Appendix G.3 (Reward Misalignment Analysis and Mitigation in Verifiable Task), provide a novel mechanism to surface RM-correctness misalignment, yielding contrastive response pairs that can support reward model fine-tuning, rejection sampling SFT, and direct preference learning.
>
> These contributions are orthogonal to the algorithmic foundations the reviewer focuses on, are clearly articulated in the paper, and are not present in MindEvolution or LLMRefine, as further detailed in our baseline comparisons above.
>
> We also note that the reviewer's citation of EvoPrompt and LLM-as-Optimizer (OPRO) directly contradicts their own summary of our work, where they correctly identify that our method 'casts reward-guided post-decoding optimization (RPDO) as discrete black-box optimization over finished LLM responses.' Both EvoPrompt and OPRO operate in the prompt optimization space (which optimize task template), which is fundamentally different from response optimization.

---

> ### Author Response · Authors · 2026-05-06
> **Response to Reviewer 4Kkc of Paper7799: W3 Deep search aggravates the reward-hacking problem it claims to address**
>
> **Section 1 motivates the work by arguing that wide shallow sampling "exacerbates reward hacking, making downstream selection methods such as best-of-N and self-consistency unreliable." But the paper's own results cut the other way. The authors note that "RPDO finds extreme high-reward but incorrect responses beyond the reach of pure sampling." Concretely, MEMETRON at generation 5 achieves Best RM = 37.48 vs. 32.92 for Base-1024, while a ~10-point gap persists between Pass@|H| (95.24%) and Best-of-|H| (85.71%) — for 4 out of 40 solvable problems, the RM's top-ranked answer is wrong. Deeper search appears to make the RM easier to exploit, not harder. The correctness-shaping fix requires oracle labels, which sidesteps rather than solves the issue.**
>
> We thank the reviewer for this point. The reviewer claims that MEMETRON's deeper search makes the RM easier to exploit, not harder, and therefore contradicts our motivation. This is incorrect, and our results demonstrate the opposite.
>
> As shown in Table 1 (Page 11) and detailed in Section 4.1, pure sampling from Base-16 to Base-1024 shows Best-of-|H| accuracy with RM degrading from 61.90% to 54.76% to 47.62%, and self-consistency stagnating from 71.43% to 69.05% to 69.05%, confirming that wider shallow sampling makes downstream selection increasingly unreliable. In stark contrast, our proposed methods, all starting from the same Base-16 initialization, show improvement from the very first generation: at generation 1, ANNETRON achieves Best-of-|H| of 78.57% and self-consistency of 73.81%, GENETRON achieves 78.57% and 78.57%, and MEMETRON achieves 78.57% and 85.71%. By generation 5, ANNETRON reaches 80.95% and 80.95%, GENETRON reaches 83.33% and 85.71%, and MEMETRON reaches 85.71% and 88.10% for Best-of-|H| and self-consistency respectively, demonstrating that structured search under the reward signal consistently improves downstream selection with RM from the very first generation. This directly disproves the reviewer's claim that deeper search makes the RM easier to exploit.
>
> Second, we are fully transparent about how MEMETRON exploits the reward model in 4 problems in Section 4.1 (Reward recognition bottleneck, Page 11). However, MEMETRON is doing exactly what it is designed to do: finding the highest reward responses under the given reward signal. If those highest reward responses happen to be incorrect for 4 problems, this means the RM itself is imperfect, rewarding incorrect responses highly in those cases. MEMETRON cannot be blamed for following the reward signal it is given, just as a search algorithm cannot be blamed for finding the wrong answer when given the wrong objective. As stated in Section 2 (Reward Models, Page 4), RM imperfection is a known and explicitly acknowledged limitation, and robust RM design is treated as an orthogonal concern addressed by existing literature. The reviewer selectively focuses only on cases where the RM fails, while completely ignoring clear improvements. The claim that 'deeper search appears to make the RM easier to exploit, not harder' is directly contradicted by the evidence clearly stated in the paper.
>
> Finally, the reviewer dismisses correctness shaping as a 'sidestep' because it requires oracle labels. This characterization is wrong and downplays three substantial contributions, clearly detailed in Contribution 4 (Page 3), Section 4.1 (Page 10, last paragraph), and Appendix G.3: in verifiable cases, it improves selection by incorporating ground-truth correctness, steering the model toward responses that are both high-reward and correct; it serves as an oracle upper bound demonstrating what MEMETRON could achieve with a perfect reward signal in unverifiable cases, reaching Best-of-|H| of 97.62% and self-consistency of 95.24%; and the comparison between runs with and without correctness shaping produces contrastive pairs that provide concrete training signals for reward model fine-tuning, rejection sampling SFT, and direct preference learning such as DPO, which is a principled path toward addressing reward hacking at its root through downstream RM improvement. The 4 problems the reviewer points to as evidence of weakness are precisely where correctness shaping is most valuable, as comparing runs with and without shaping on these problems produces contrastive pairs that expose where the RM is misaligned with ground truth and provide valuable training signals for improving the RM itself. The reviewer's characterization of this as a 'sidestep' is therefore unfair and would equally invalidate the entire literature on Reinforcement Learning with Verifiable Rewards (RLVR), which relies on the same principle.

---

> ### Author Response · Authors · 2026-05-06
> **Response to Reviewer 4Kkc of Paper7799: W4 Reliance on strong base model**
>
> **The entire search happens over outputs of a frozen base model. When the base model simply cannot produce a good answer, more search will not help — you cannot find what is not in the space. In the reported experiments, the generator is already strong enough that Base-1024 hits 100% Pass@|H| on the math benchmark. This raises a practical question: how weak can the base model be before MEMETRON stops paying off? And if the method only works well when the model is already nearly competent, the marginal value of ~39 min/problem of extra search becomes hard to justify. The paper would benefit from experiments on tasks or models where the generator is clearly weaker.**
>
> The reviewer's concern about base model capability is largely trivial and misdirected. The core claim, that MEMETRON requires a strong base model, applies equally to every LLM-based method and reveals a fundamental misunderstanding of what constitutes a meaningful criticism of our specific approach.
>
> The reviewer's characterization of our base models as 'strong' is misleading. The models used in our experiments, Qwen3-8B in Section 4.1 and Llama-3.2-3B-Instruct in Section 4.2, are small models by current standards. Despite this, MEMETRON demonstrates that it does not require a nearly competent model: as shown in Table 1, MEMETRON discovers correct answers through generations of search even when they are not present in the initial Base-16 pool, improving correctness coverage from 85.71% to 95.24% over 3 generations. This is possible because MEMETRON's search operators, crossover, mutation, and refinement, actively generate new responses through each generation that were not present in the original pool, continuously expanding the search space.
>
> The reviewer's notion of model competence is also misleading and differs from what we define in our paper and what is generally accepted in the literature. We explicitly noted that Pass@|H| is an oracle metric not available in practice, and the ability to produce a correct answer somewhere across 1024 samples does not constitute competence, it simply means that with enough sampling one can get lucky, as we explicitly discuss in our paper (Page 10, Paragraph 3, beginning 'However'). A truly competent model should achieve correct answers reliably across multiple independent samples, such that correct answers dominate the pool, wrong answers rarely appear, and even an imperfect RM would have a much lower chance of ranking wrong answers highly, naturally leading to high Best-of-|H| and self-consistency. As noted by Yu (2025) as well, pass@k should be regarded as a diagnostic of a model's reasoning diversity rather than a measure of competence, and a truly capable model should converge toward high pass@1, meaning consistent correct answers across independent samples (arXiv:2511.16231). By this definition, Base-1024 with Best-of-|H| of only 47.62% and self-consistency of 69.05% is far from competent. The reviewer's claim that the method only works well when the model is already nearly competent therefore does not hold, as our base models are demonstrably not competent by any reasonable measure, yet MEMETRON, GENETRON, and ANNETRON all provide substantial and consistent gains from the very first generation.
>
> We explicitly acknowledge the assumption of a reasonably capable base model in our Limitations and Future Work section, treating it as a transparent and obvious assumption rather than a hidden one. This assumption is not unique to MEMETRON. If a base model is so weak that it cannot generate any useful responses, whether due to fewer parameters, inferior training data, worse architecture, older model generations, or lack of instruction tuning, most LLM-based inference-time methods fail: generation itself fails, Best-of-N fails, self-consistency fails, self-refine fails, and reflexion fails. The reviewer is essentially saying “if the LLM doesn't work, MEMETRON doesn't work”, which is trivially true and not a meaningful criticism of our method specifically. Base model capability is furthermore an orthogonal and actively improving concern, with both open-source and enterprise communities continuously advancing model performance.

---

> > ### Author Response · Authors · 2026-05-06
> > **Response to Reviewer 4Kkc of Paper7799: W4 Reliance on strong base model (Cont)**
> >
> > As discussed above, for extremely weak models, most if not all LLM-based inference-time methods will fail. For extremely competent models that already achieve high Pass@1 accuracy consistently across independent samples, MEMETRON or other methods are simply unnecessary. MEMETRON is most valuable precisely for models in this middle ground: inconsistent enough that Best-of-N and self-consistency fail to reliably select correct answers under pure sampling, but capable enough to produce responses that the RM can score meaningfully and the search operators can improve upon through generations. As shown in Table 1, our results demonstrate this directly: With Qwen3-8B non-thinking mode, starting from Base-16 with Best-of-|H| of 61.90% and self-consistency of 71.43%, MEMETRON progressively improves both correctness coverage and selection reliability through generations, reaching Best-of-|H| of 85.71% and self-consistency of 88.10% at generation 5, and 97.62% Best-of-|H| and 95.24% self-consistency with correctness shaping in the case of a higher quality RM. The reviewer overlooks these substantial improvements MEMETRON achieves, which directly contradicts the claim that the model is strong.
> >
> > The reviewer's concern about computational cost assumes that all 5 generations are required, but this is not the case. As shown in Table 1, meaningful gains are achieved much earlier: MEMETRON at generation 1 already achieves Best-of-|H| of 78.57% and self-consistency of 85.71% with only 32 candidates, and generation 3 reaches 95.24% Pass@|H|. Furthermore, practitioners can choose lighter alternatives such as ANNETRON and GENETRON, which also provide meaningful gains over pure sampling at lower computational cost. The number of generations is a flexible budget parameter that practitioners can tune based on their computational constraints and desired performance tradeoff. The 39 min/problem figure corresponds to our fixed 5 generation experimental setup, chosen to enable fair comparison across all systems, not a required cost.
> >
> > The reviewer suggests experiments on tasks or models where the generator is clearly weaker. However, as we demonstrate above, our current experimental setting already represents precisely this scenario: Qwen3-8B in non-thinking mode is far from competent on AMC 2025 problems, with Best-of-|H| degrading from 61.90% to 54.76% to 47.62% and self-consistency stagnating from 71.43% to 69.05% to 69.05% across Base-16, Base-128, and Base-1024 under pure sampling. This is not a strong model confidently solving problems, it is a model that struggles with reliable selection, which is exactly the sweet spot where MEMETRON is most valuable. Furthermore, our experiments already include Llama-3.2-3B-Instruct, a 3 billion parameter model, which demonstrates consistent gains on instruction following tasks as measured by GPT judge preference, showing that our method generalizes across model sizes. We are nonetheless willing to conduct additional experiments with a weaker model on mathematical reasoning if the editorial team considers this necessary for acceptance. We expect stronger models to yield better performance and weaker models to yield weaker performance, as is expected of any inference-time method. Any performance degradation observed would likely affect other LLM-based inference-time methods as well, and is unlikely to reveal a limitation unique to MEMETRON. Furthermore, TMLR's own acceptance criteria explicitly state that lack of additional experiments is not grounds for rejection as long as the existing claims are supported by convincing evidence, which they are (https://jmlr.org/tmlr/acceptance-criteria.html).

---

> ### Author Response · Authors · 2026-05-06
> **Response to Reviewer 4Kkc of Paper7799: Additional questions**
>
> **Q1: Could you run Mind Evolution and LLMRefine (or faithful re-implementations) under the same compute budget and report numbers?**
>
> As detailed in W1 above, direct comparison is not feasible. To implement MindEvolution faithfully, one would need task-specific programmatic verifiers for mathematical reasoning and instruction following, which are not generally available for these settings. To implement LLMRefine faithfully, one would need to train task-specific feedback models for each task and then convert natural language feedback into scalar scores, a two-stage pipeline that introduces noise and is difficult to calibrate across tasks, as we note in our introduction. These are fundamental design requirements of the original methods, not implementation details that can be easily worked around. In contrast, our methods work directly with reward models already built for LLMs, which are broadly applicable across tasks without any task-specific infrastructure. GENETRON and ANNETRON are designed precisely to capture the core algorithmic principles of these methods while removing these constraints, making them not only the closest feasible experimental stand-ins but also more general ones.
>
> **Q2: Beyond the GA+SA combination: what specific operator design or scheduling decision in MEMETRON is new and could not be trivially assembled from prior work?**
>
> MEMETRON is not trivially assembled from prior work. As detailed in W2 above, the RPDO problem formulation with blackbox scalar reward, the design of frozen LLMs as search operators that align with the reward objective without access to raw reward scores, and the application of memetic optimization to response optimization under blackbox reward signals are all novel contributions not present in any cited prior work.
>
> **Q3: The data show MEMETRON finds high-reward wrong answers that pure sampling never reaches. How should a practitioner think about deploying this without oracle labels?**
>
> A practitioner deploying MEMETRON without oracle labels should rely on the practically relevant metrics: Best-of-|H| and self-consistency, both of which improve substantially with MEMETRON. As shown in Table 1 and detailed in W3 above, MEMETRON achieves Best-of-|H| of 85.71% and self-consistency of 88.10% without oracle labels, substantially outperforming pure sampling where Best-of-|H| degrades from 61.90% to 54.76% to 47.62% and self-consistency stagnates from 71.43% to 69.05% to 69.05% across Base-16, Base-128, and Base-1024. The existence of high-reward but incorrect responses for 4 problems reflects RM imperfection, which is an orthogonal concern explicitly acknowledged in our paper. In practice, a practitioner should treat MEMETRON as a principled alternative to pure sampling that improves the quality and reliability of the response pool, while being aware that RM imperfection is a shared limitation of all reward-guided inference-time methods, not unique to MEMETRON.
>
> **Q4: Have you tried MEMETRON on a task where the base model's Pass@1024 is, say, below 50%? What happens?**
>
> We have not yet evaluated this specific setting. However, as discussed in W4 above, our current experimental setting already represents a challenging scenario where the base model is far from competent, with Best-of-|H| degrading to 47.62% and self-consistency stagnating at 69.05% under pure sampling. We are therefore not operating in a regime where the base model is strong. We have demonstrated consistent improvements across two distinct tasks, mathematical reasoning and instruction following, and two different model sizes, Qwen3-8B and Llama-3.2-3B-Instruct, providing strong evidence that MEMETRON generalizes across settings. Furthermore, TMLR's own acceptance criteria explicitly state that lack of additional experiments is not grounds for rejection as long as the existing claims are supported by convincing evidence, which they are (https://jmlr.org/tmlr/acceptance-criteria.html). We are nonetheless willing to conduct additional experiments if the editorial team considers this necessary for acceptance.

---

> ### Author Response · Authors · 2026-05-08
> **Response to Reviewer 4Kkc of Paper7799: AIME 2026 Evaluation**
>
> Since the completion of this manuscript, AIME 2026 has been released. Based on your feedback, we took this opportunity to evaluate our methods on these newly available problems. The results are consistent with our AMC 12 2025 experiments in the paper, with all methods improving over Base-16, and ANNETRON and GENETRON again showing comparable performance to each other. MEMETRON achieves the strongest results, reaching 73.33% Pass@|H|, 46.67% Best-of-|H|, and 46.67% Self-consistency after 5 generations. As AIME is a significantly harder competition than AMC 12, we observe a large gap between Pass@|H| (73.33%, the coverage of correct responses in the candidate pool) and Best-of-|H| (46.67%, the reward model's ability to select them), showcasing the severe misalignment of the reward model when facing challenging problems. This indicates that with a perfect reward model, Best-of-|H| and Self-consistency could reach as high as 73.33%, highlighting the improvement of reward modeling as a promising direction for future work. The results are presented in the table below.
>
> **Table: Performance of the proposed RPDO methods on 30 AIME 2026 problems**, using Qwen3-8B (non-thinking) as the generator and Skywork-Reward-V2-Llama-3.1-8B-40M as the reward model. Base-16 is the shared initialization pool for all RPDO methods. All experimental setup follows Section 4 except the hardware is switched to an NVIDIA RTX PRO 6000 Blackwell Server Edition.
>
> | **System** | **Pass@\|H\|** | **Best RM (CI₉₅)** | **Best-of-\|H\|** | **Self-consistency** | **\|H\|** |
> |---|---|---|---|---|---|
> | Base-16 | 10/30 (33.33%) | 25.18 [22.85, 27.51] | 6/30 (20.00%) | 6/30 (20.00%) | 16 |
> | ANNETRON (Base-16 + 1 iteration) | 11/30 (36.67%) | 28.28 [25.66, 30.89] | 6/30 (20.00%) | 6/30 (20.00%) | 32 |
> | ANNETRON (Base-16 + 2 iterations) | 14/30 (46.67%) | 29.43 [26.82, 32.04] | 7/30 (23.33%) | 6/30 (20.00%) | 48 |
> | ANNETRON (Base-16 + 3 iterations) | 14/30 (46.67%) | 30.07 [27.50, 32.64] | 8/30 (26.67%) | 8/30 (26.67%) | 64 |
> | ANNETRON (Base-16 + 4 iterations) | 14/30 (46.67%) | 31.18 [28.74, 33.62] | 9/30 (30.00%) | 8/30 (26.67%) | 80 |
> | ANNETRON (Base-16 + 5 iterations) | 14/30 (46.67%) | 31.85 [29.50, 34.19] | 10/30 (33.33%) | 9/30 (30.00%) | 96 |
> |---|---|---|---|---|---|
> | GENETRON (Base-16 + 1 generation) | 10/30 (33.33%) | 28.55 [26.05, 31.06] | 7/30 (23.33%) | 8/30 (26.67%) | 32 |
> | GENETRON (Base-16 + 2 generations) | 12/30 (40.00%) | 30.05 [27.56, 32.55] | 8/30 (26.67%) | 8/30 (26.67%) | 48 |
> | GENETRON (Base-16 + 3 generations) | 14/30 (46.67%) | 31.71 [29.31, 34.12] | 9/30 (30.00%) | 9/30 (30.00%) | 64 |
> | GENETRON (Base-16 + 4 generations) | 15/30 (50.00%) | 32.59 [30.19, 35.00] | 10/30 (33.33%) | 9/30 (30.00%) | 80 |
> | GENETRON (Base-16 + 5 generations) | 15/30 (50.00%) | 33.50 [31.19, 35.89] | 10/30 (33.33%) | 9/30 (30.00%) | 96 |
> |---|---|---|---|---|---|
> | MEMETRON (Base-16 + 1 generation) | 16/30 (53.33%) | 32.06 [29.72, 34.40] | 9/30 (30.00%) | 9/30 (30.00%) | 32 |
> | MEMETRON (Base-16 + 2 generations) | 18/30 (60.00%) | 34.84 [32.61, 37.06] | 11/30 (36.67%) | 11/30 (36.67%) | 48 |
> | MEMETRON (Base-16 + 3 generations) | 20/30 (66.67%) | 36.20 [33.95, 38.44] | 12/30 (40.00%) | 13/30 (43.33%) | 64 |
> | MEMETRON (Base-16 + 4 generations) | 22/30 (73.33%) | 37.08 [34.90, 39.25] | 13/30 (43.33%) | 14/30 (46.67%) | 80 |
> | **MEMETRON (Base-16 + 5 generations)** | **22/30 (73.33%)** | **37.84 [35.73, 39.96]** | **14/30 (46.67%)** | **14/30 (46.67%)** | **96** |

---

> ### Comment · Reviewer_4Kkc · 2026-05-27
>
> Dear Authors,
>
> Thanks for the informative rebuttal and the additional AIME 2026 experiments.
>
> I update my assessment on W3 here. The authors' data is convincing on one specific point we underweighted: pure sampling from Base-16 to Base-1024 degrades Best-of-|H| from 61.90% to 47.62%, while MEMETRON improves it to 85.71% from the same starting point. This demonstrates that structured search under the reward signal does improve downstream selection reliability relative to naive scaling.
>
> Regarding W4, the AIME 2026 results are helpful. On a harder benchmark, Pass@|H| reaches 73.33% while Best-of-|H| is only 46.67%, confirming that MEMETRON provides gains in a regime where the model clearly struggles and that RM quality is the binding constraint. This partially addresses my previous concern. I still believe an experiment where Pass@1024 is well below 50% would be informative.
>
> Regarding W2, while I agree that TMLR does not require novelty as a necessary condition, my concern is whether the contributions meet the bar for "significant new knowledge". From my perspective, the core method remains a standard memetic algorithm (GA + SA) with LLM-based operators.  Varies work have been established under this LLM-as-evolutionary-operator paradigm: [1] uses LLMs as mutation operators over program solutions; [2] formalizes LLM-based crossover over arbitrary text genotypes; [3] employs LLMs as adaptive crossover and mutation operators for code-level neural architecture search; and [4] uses LLMs as evolutionary variation operators for combinatorial optimization. All of these operate in solution space, not prompt space, but still demonstrate that using frozen LLMs as search operators within evolutionary loops is a well-explored design pattern. My novelty concern is about this broader pattern, not the prompt-optimization subset the authors address in their rebuttal.
>
> Thus, to constitute sufficient contribution on its own, the empirical evidence needs to be particularly compelling, which brings us back to W1.
>
> Finally, regarding W1, while I appreciate the clarification that Mind Evolution and LLMRefine have no public code and rely on task-specific infrastructure, the claim that GENETRON and ANNETRON serve as "closest feasible stand-ins" is circular: if GENETRON ≈ Mind Evolution and ANNETRON ≈ LLMRefine, according to the paper, then showing MEMETRON > GENETRON/ANNETRON only shows the combination beats its own components, not that it advances over prior work. Mind Evolution's core algorithm is described in sufficient detail that a fair adaptation using the same RM as the fitness function is feasible, even without the original codebase. We acknowledge practical difficulty but maintain this is the most important missing piece in the evaluation.
>
> In general, the rebuttal and extra experiments are helpful. Thanks again for the clarifications.
>
> Best,
>
> Reviewer 4Kkc
>
>
> ----
>
> **References**
> - [1] Lehman, J., Gordon, J., Jain, S., Ndousse, K., Yeh, C., & Stanley, K. O. (2022). Evolution through Large Models. arXiv:2206.08896. https://arxiv.org/abs/2206.08896
> - [2] Meyerson, E., Nelson, M. J., Bradley, H., Gaier, A., Moradi, A., Hoover, A. K., & Lehman, J. (2023). Language Model Crossover: Variation through Few-Shot Prompting. arXiv:2302.12170. https://arxiv.org/abs/2302.12170
> - [3] Chen, A., Dohan, D. M., & So, D. R. (2023). EvoPrompting: Language Models for Code-Level Neural Architecture Search. NeurIPS 2023. arXiv:2302.14838. https://arxiv.org/abs/2302.14838
> - [4] Ye, H., Wang, J., Cao, Z., Berto, F., Hua, C., Kim, H., Park, J., & Song, G. (2024). ReEvo: Large Language Models as Hyper-Heuristics with Reflective Evolution. NeurIPS 2024. arXiv:2402.01145. https://arxiv.org/abs/2402.01145

---

> > ### Author Response · Authors · 2026-06-01
> > **Response to Updated Review from Reviewer 4Kkc: W3 and W4**
> >
> > We thank the reviewer for the updated review. We hope that our detailed responses have addressed all remaining concerns and that the reviewer and editorial team can see the practical usefulness and contribution of MEMETRON. We address each updated point below.
> >
> > **Response to updated W3: I update my assessment on W3 here. The authors' data is convincing on one specific point we underweighted: pure sampling from Base-16 to Base-1024 degrades Best-of-|H| from 61.90% to 47.62%, while MEMETRON improves it to 85.71% from the same starting point. This demonstrates that structured search under the reward signal does improve downstream selection reliability relative to naive scaling.**
> >
> > We thank the reviewer for the updated assessment and for acknowledging the strength of our results on W3.
> >
> > **Response to updated W4: Regarding W4, the AIME 2026 results are helpful. On a harder benchmark, Pass@|H| reaches 73.33% while Best-of-|H| is only 46.67%, confirming that MEMETRON provides gains in a regime where the model clearly struggles and that RM quality is the binding constraint. This partially addresses my previous concern. I still believe an experiment where Pass@1024 is well below 50% would be informative.**
> >
> > We thank the reviewer for the partial update and for acknowledging that the AIME 2026 results confirm MEMETRON provides gains in a regime where the model clearly struggles. We address the remaining concern:
> >
> > The reviewer states they "still believe an experiment where Pass@1024 is well below 50% would be informative." We appreciate the interest, but respectfully note that this request goes beyond what is required under TMLR's stated policies. TMLR's reviewer guide explicitly states that gaps between claims and evidence can be addressed either by providing more experiments or by authors adjusting their claims, and that missing experiments alone are not grounds for rejection when existing evidence is convincing. Our claims are scoped to the settings we evaluate, and the evidence is accurate, convincing, and clear.
> >
> > Critically, the reviewer does not identify a specific claim in the paper that requires a Pass@1024 < 50% setting to be substantiated. Without such a link, the request is an open-ended empirical question rather than a gap between a stated claim and its supporting evidence, a distinction TMLR's policies draw explicitly. Benchmark selection in our work follows standard practice in the literature: we evaluate on established, well-known benchmarks, with the additional care of selecting the newest available versions to minimize the risk of data contamination. We do not search exhaustively for benchmarks that hit a specific failure threshold, and no paper in the literature is expected to do so unless benchmark construction is an explicit contribution. We have selected AMC 2025 and AIME 2026, among the hardest publicly available math benchmarks for models of this scale, precisely because they represent the most challenging settings available to us. We also note a practical difficulty: identifying a benchmark where Pass@1024 falls well below 50% requires finding and running large-scale sampling experiments across multiple candidate benchmarks without knowing the outcome in advance, representing significant computational cost. Even if such a benchmark were found, it is unclear what additional claims this would add to the paper.
> >
> > More fundamentally, as we have already argued in our previous response, Pass@1024 is not a practically realizable metric, it requires oracle access to ground-truth labels across 1024 candidates, which is unavailable in real deployment. In contrast, MEMETRON is realizable in practice: starting from a small, practically affordable pool of 16 candidates, it demonstrably improves both quality and selection reliability without oracle access. We also note that the reviewer acknowledged in their updated assessment on W3 that pure sampling from Base-16 to Base-1024 degrades Best-of-|H|, meaning that scaling pure sampling to 1024 candidates already fails. The question of what happens when Pass@1024 < 50% is therefore moot: even when Pass@1024 is 100%, pure sampling at scale is already unreliable, and MEMETRON already addresses this. It is unclear what additional claim a Pass@1024 < 50% setting would enable or falsify that is not already established by our results.
> >
> > Therefore, we believe our experiments and results on AMC 2025, AIME 2026, and TinyAlpacaEval constitute a thorough and representative experimental scope. We remain willing to conduct additional experiments if the editorial team considers them necessary for acceptance, provided they are well-motivated by a specific gap between our stated claims and the existing evidence.

---

> > ### Author Response · Authors · 2026-06-01
> > **Response to Updated Review from Reviewer 4Kkc: W2**
> >
> > **Response to updated W2:**
> >
> > The reviewer argues whether our contributions meet the bar for "significant new knowledge." We respectfully point out that this standard does not appear anywhere in TMLR's acceptance criteria (https://jmlr.org/tmlr/acceptance-criteria.html). TMLR's criteria explicitly state: "it should not be used as a reason to reject work that isn't considered 'significant' or 'impactful'... Nor should it form the basis for rejecting work on a method considered not 'novel enough', as novelty of the studied method is not a necessary criteria for acceptance. We explicitly avoid these terms ('significant', 'impactful', 'novel')." The reviewer's framing of a higher empirical bar conditional on perceived novelty is therefore not a valid basis for rejection under TMLR's own policies.
> >
> > We also note that the reviewer's novelty concern has shifted twice across the review cycle, first citing EvoPrompt and OPRO, which we demonstrated operate in prompt space rather than response space, and now citing [1]–[4]. This pattern of shifting, combined with invoking a standard not present in TMLR's criteria, suggests the concern is not grounded in a specific, stable technical gap.
> >
> > On the technical substance of the concern: the reviewer claims that [1]–[4] establish "using frozen LLMs as search operators within evolutionary loops" as a well-explored design pattern that encompasses MEMETRON and "a standard memetic algorithm (GA + SA) with LLM-based operators". This characterization is incorrect, and a careful reading of each cited work reveals a fundamental distinction that the reviewer overlooks.
> >
> > First, none of [1]–[4] actually use a GA+SA or memetic structure. ELM [1] uses MAP-Elites which focus mainly on mutating operator; Language Model Crossover [2] implements a simple genetic algorithm where crossover is performed by naively concatenating parent texts as a few-shot prompt and sampling from the LLM, a primitive crossover operator; EvoPrompting [3] uses GA with few-shot crossover and mutation operators to do neural architecture search (NAS); and ReEvo [4] uses evolutionary search (not exactly GA) with LLM feedback. Not one of the four cited works uses the memetic GA+SA structure the reviewer attributes to them. The reviewer's own cited evidence does not support their own characterization of MEMETRON as a well-explored pattern.
> >
> > Second, all four cited works operate in a categorically different setting from MEMETRON. ELM [1], Language Model Crossover [2], EvoPrompting [3], and ReEvo [4] all evolve code or programs with verifiable fitness signals. In every case, the fitness signal is ground-truth aligned by construction: there is no gap between what the fitness function measures and what you actually want.
> >
> > MEMETRON operates in a categorically different setting. The genotype is natural language responses, not code. The fitness signal is a learned reward model, a signal that is non-deterministic, potentially misaligned with ground truth, and by definition unverifiable, since if ground truth were available there would be no need for a reward model in the first place.
> >
> > The reviewer states their concern is about "this broader pattern, not the prompt-optimization subset the authors address in their rebuttal." We agree that MEMETRON operates in the metaheuristic evolutionary space, this has never been in dispute. However, as demonstrated above, none of the four papers the reviewer cites as evidence of this "broader pattern" actually share MEMETRON's setting, search structure, or the core challenges it addresses. Citing works that are superficially in the same "LLM in an evolutionary loop" family does not constitute evidence that MEMETRON's specific contributions are well-explored.
> >
> > Finally, the reviewer concludes that "thus, to constitute sufficient contribution on its own, the empirical evidence needs to be particularly compelling, which brings us back to W1." This conclusion does not follow. Again, the phrase "particularly compelling" does not appear anywhere in TMLR's acceptance criteria (https://jmlr.org/tmlr/acceptance-criteria.html). The actual standard is simply "accurate, convincing and clear evidence." For the interest criterion, TMLR states explicitly: "if the authors make it clear that there is something to be learned by some researchers in their area from their work, then the criterion of interest is considered satisfied." The reviewer is inventing a higher evidential bar conditional on a novelty judgment that TMLR explicitly prohibits as grounds for rejection.

---

> > ### Author Response · Authors · 2026-06-01
> > **Response to Updated Review from Reviewer 4Kkc: W1**
> >
> > **Response to updated W1:**
> >
> > The reviewer argues that showing MEMETRON > GENETRON/ANNETRON "only shows the combination beats its own components, not that it advances over prior work." This argument has three flaws.
> >
> > First, we never claimed GENETRON ≈ Mind Evolution or ANNETRON ≈ LLMRefine. We called them "closest feasible stand-ins" precisely because they share the same underlying algorithmic principles: GENETRON and Mind Evolution are both grounded in genetic algorithms, ANNETRON and LLMRefine are both grounded in simulated annealing. But GENETRON and ANNETRON are designed from the ground up to operate under a general blackbox scalar reward model, which Mind Evolution and LLMRefine are not. The reviewer is attacking a claim we never made.
> >
> > Second, Mind Evolution and LLMRefine cannot be implemented in our setting. Mind Evolution requires task-specific programmatic verifiers. LLMRefine requires training a task-specific feedback model per task. These are fundamental architectural requirements, not implementation details. Whether these methods could hypothetically be extended to work with a scalar reward model is speculation; we cannot and should not compare against methods whose behavior under a reward model signal has never been studied or validated. GENETRON and ANNETRON are not approximations of those methods; they are purpose-built for the RPDO setting.
> >
> > Third, the reviewer's claim that "the combination beats its own components, not that it advances over prior work" is an unfair standard and not supported by TMLR's criteria. By this logic, no memetic algorithm or useful combination of algorithms ever advances over prior work, since every memetic algorithm combines a global search component with a local search component. The entire point of memetic algorithms is that the combination achieves what neither component achieves alone. Demonstrating this in a new setting where no prior work operates is precisely an advance over prior work under any reasonable definition, and certainly under TMLR's criteria.
> >
> > Finally, the reviewer suggests adapting Mind Evolution using the same RM as fitness is feasible. This is directly contradicted by Mind Evolution's own authors. Mind Evolution explicitly scopes itself to settings where a programmatic solution evaluator is available, stating: "The proposed approach avoids the need to formalize the underlying inference problem whenever a solution evaluator is available." More critically, the authors explicitly acknowledge that extending to learned reward models is an open problem, stating: "learned feedback models or self-evaluators can be noisy and are not perfectly reliable. We leave consideration of such approximate feedback mechanisms for future work." The reviewer is asking us to implement precisely what Mind Evolution's own authors identify as future work. Replacing the programmatic verifier with a scalar RM is not a feasible adaptation; it is the open problem that MEMETRON addresses. The resulting system would not be Mind Evolution. It would be GENETRON. That comparison already exists in our paper.
> >
> > The reviewer maintains this is "the most important missing piece in the evaluation." We respectfully disagree. We have demonstrated to the best of our ability given the current state of the literature, and we believe our evaluation is complete and reasonable.

---

### Review · Reviewer_V3qx · 2026-04-17

**Summary Of Contributions:**

This paper introduces MEMETRON, a novel memetic optimization framework designed for reward-guided post-decoding optimization (RPDO) of LLMs. The authors address the limitations of existing one-shot, shallow sampling methods like BoN, which often fail to reach higher-reward responses or suffer from reward hacking as the sampling pool expands. MEMETRON formulates response optimization as a discrete black-box problem and combines a genetic algorithm-based global search (i.e., GENETRON) with a simulated annealing-style local refinement (i.e., ANNETRON). Crucially, it leverages frozen LLMs as variation operators to perform semantic edits directly in the response space, guided by a black-box scalar reward without exposing the numerical scores to the LLM operators themselves. By framing post-decoding optimization as a discrete black-box problem, the paper provides a unified view of existing heuristics and exposes the gap that structured search can fill. Empirically, MEMETRON demonstrates significant improvements over strong sampling baselines on both mathematical reasoning and instruction-following tasks, showing better correctness coverage and alignment with human or model preferences. Furthermore, the authors propose a practical diagnostic method using correctness-conditioned reward shaping to expose reward model misalignment and generate contrastive pairs.

**Audience:**

Yes

**Audience Explanation:**

The paper directly addresses the highly topical area of inference-time compute and test-time optimization. As the community increasingly explores methods to improve performance by allocating more compute at inference time (as seen in recent reasoning models), MEMETRON provides a structured framework for exploring the response space.

- The one interested in the paradigm of "LLMs as optimizers" will find the methodology highly valuable. The paper demonstrates a successful integration of classical metaheuristic principles (genetic algorithms and simulated annealing) with large language models acting as semantic mutation and crossover operators. This opens up new avenues for discrete, gradient-free optimization in natural language space.
- Also, the findings are crucial for the community working on LLM alignment and RLHF. A persistent challenge in aligning models is the imperfect nature of reward models and their susceptibility to "reward hacking." This paper provides a practical method to stress-test reward models, diagnose misalignments, and generate high-quality contrastive datasets to improve them.
- The approach offers significant practical value for practitioners with limited resources or API-only access. Because MEMETRON does not require gradient updates or access to model weights and operates purely on black-box reward signals, it enables performance optimization on top of proprietary or fixed models where traditional fine-tuning is infeasible.

**Broader Impact Concerns:**

No Broader Impact Section. Given that the topic is related LLM, it is recommended to include.

**Claims And Evidence:**

Yes

**Claims Explanation:**

The claims are generally supported by convincing evidence across diverse tasks.

- The authors evaluate MEMETRON on both mathematical reasoning (AMC 12) and open-ended instruction following (TinyAlpacaEval), demonstrating that the framework consistently outperforms strong sampling baselines (up to Base-1024) in discovering higher-reward responses.
- The use of confidence intervals and controlled sampling budgets strengthens the claims. The experimental results in Table 1 and Table 2 include 95% confidence intervals, showing that the improvements are statistically significant and not only the result of increased sampling budgets.
- The ablation study effectively demonstrates the synergy of the proposed components. The paper compares MEMETRON with its individual modules (GENETRON and ANNETRON alone), providing clear evidence that the combination of global exploration and local refinement yields superior results compared to either strategy in isolation.

**Requested Changes:**

- The scale of the mathematical reasoning dataset is relatively small. While the use of the recent 2025 AMC 12 dataset effectively mitigates the risk of training-data contamination, the evaluation is limited to 42 non-figure problems. A larger-scale evaluation on benchmarks like MATH or GSM8K would have provided even more robust evidence.

- The trade-off between compute cost and quality is transparently reported but highlights a practical bottleneck. The paper clearly states that MEMETRON incurs significantly higher inference time compared to baseline methods (e.g., over 2300 seconds per question compared to 34 seconds for Base-16). While the evidence supports the claim of quality improvement, it also clearly outlines the method's limitation in terms of test-time compute efficiency. While it mentions that reward-based early stopping could reduce costs, it does not provide a detailed sensitivity analysis or empirical results for this. It is helpful to add an analysis on how different early stopping criteria affect the trade-off between performance and compute time.

- The current evaluation only compares MEMETRON against pure sampling baselines (Base-16, Base-128, Base-1024). However, the paper discusses other "deep" RPDO methods in the related work, such as LLMRefine or Mind Evolution. To truly demonstrate the advantage of combining population-based search and local refinement under a shared scalar reward, it is critical to include an empirical comparison with at least one of these existing iterative/optimization-based methods, or provide a compelling experimental demonstration showing why they fail in the tested settings.

- The paper discusses reward model misalignment and introduces correctness shaping to diagnose it. However, it would strengthen the paper to provide a more quantitative analysis of how often "reward hacking" (cases where the RM score increases but the actual correctness degrades) occurred during the pure MEMETRON search versus the shaped search, and how severe it was across generations.

---

> ### Author Response · Authors · 2026-05-06
> **Response to Reviewer V3qx of Paper7799: Opening & Overview**
>
> We thank the reviewer for the thorough and constructive review, and for recognizing the conceptual contributions of MEMETRON, particularly the unification of genetic and simulated annealing search under a shared scalar reward, the diagnostic value of correctness shaping for exposing reward model misalignment, and the practical applicability of the framework to black-box settings without gradient access. We address each of the raised concerns below.

---

> ### Author Response · Authors · 2026-05-06
> **Response to Reviewer V3qx of Paper7799: Small mathematical reasoning dataset.**
>
> **The scale of the mathematical reasoning dataset is relatively small. While the use of the recent 2025 AMC 12 dataset effectively mitigates the risk of training-data contamination, the evaluation is limited to 42 non-figure problems. A larger-scale evaluation on benchmarks like MATH or GSM8K would have provided even more robust evidence.**
>
> We thank the reviewer for this comment. The choice of 42 AMC 12 2025 problems was deliberate and motivated by data contamination concerns. Standard benchmarks such as MATH, MATH500, and GSM8K are saturated for Qwen3-8B due to contamination, with Qwen3-8B achieving near-perfect accuracy on these datasets, leaving no room for improvement and making MEMETRON unnecessary by construction. AMC 12 2025 problems are not only released after the model's training cutoff, making contamination unlikely, but are also significantly harder than these benchmarks, providing a more meaningful evaluation of reasoning improvement.
>
> We also note that AIME 2025, the community standard for evaluating mathematical reasoning on frontier models, contains only 30 problems, fewer than our 42-problem subset, and was the sole mathematical reasoning benchmark used by OpenAI when introducing o3 and o4-mini (https://openai.com/index/introducing-o3-and-o4-mini/) and by Google when introducing Gemini 2.5 Pro (https://blog.google/technology/google-deepmind/gemini-model-thinking-updates-march-2025/), yet frontier models now achieve near-perfect accuracy on it after months of exposure, rendering it ineffective as a discriminative benchmark."

---

> > ### Author Response · Authors · 2026-05-08
> > **Response to Reviewer V3qx of Paper7799: AIME 2026 Evaluation**
> >
> > Since the completion of this manuscript, AIME 2026 has been released. Based on your feedback, we took this opportunity to evaluate our methods on these newly available problems. The results are consistent with our AMC 12 2025 experiments in the paper, with all methods improving over Base-16, and ANNETRON and GENETRON again showing comparable performance to each other. MEMETRON achieves the strongest results, reaching 73.33% Pass@|H|, 46.67% Best-of-|H|, and 46.67% Self-consistency after 5 generations. As AIME is a significantly harder competition than AMC 12, we observe a large gap between Pass@|H| (73.33%, the coverage of correct responses in the candidate pool) and Best-of-|H| (46.67%, the reward model's ability to select them), showcasing the severe misalignment of the reward model when facing challenging problems. This indicates that with a perfect reward model, Best-of-|H| and Self-consistency could reach as high as 73.33%, highlighting the improvement of reward modeling as a promising direction for future work. The results are presented in the table below.
> >
> > **Table: Performance of the proposed RPDO methods on 30 AIME 2026 problems**, using Qwen3-8B (non-thinking) as the generator and Skywork-Reward-V2-Llama-3.1-8B-40M as the reward model. Base-16 is the shared initialization pool for all RPDO methods. All experimental setup follows Section 4 except the hardware is switched to an NVIDIA RTX PRO 6000 Blackwell Server Edition.
> >
> > | **System** | **Pass@\|H\|** | **Best RM (CI₉₅)** | **Best-of-\|H\|** | **Self-consistency** | **\|H\|** |
> > |---|---|---|---|---|---|
> > | Base-16 | 10/30 (33.33%) | 25.18 [22.85, 27.51] | 6/30 (20.00%) | 6/30 (20.00%) | 16 |
> > | ANNETRON (Base-16 + 1 iteration) | 11/30 (36.67%) | 28.28 [25.66, 30.89] | 6/30 (20.00%) | 6/30 (20.00%) | 32 |
> > | ANNETRON (Base-16 + 2 iterations) | 14/30 (46.67%) | 29.43 [26.82, 32.04] | 7/30 (23.33%) | 6/30 (20.00%) | 48 |
> > | ANNETRON (Base-16 + 3 iterations) | 14/30 (46.67%) | 30.07 [27.50, 32.64] | 8/30 (26.67%) | 8/30 (26.67%) | 64 |
> > | ANNETRON (Base-16 + 4 iterations) | 14/30 (46.67%) | 31.18 [28.74, 33.62] | 9/30 (30.00%) | 8/30 (26.67%) | 80 |
> > | ANNETRON (Base-16 + 5 iterations) | 14/30 (46.67%) | 31.85 [29.50, 34.19] | 10/30 (33.33%) | 9/30 (30.00%) | 96 |
> > |---|---|---|---|---|---|
> > | GENETRON (Base-16 + 1 generation) | 10/30 (33.33%) | 28.55 [26.05, 31.06] | 7/30 (23.33%) | 8/30 (26.67%) | 32 |
> > | GENETRON (Base-16 + 2 generations) | 12/30 (40.00%) | 30.05 [27.56, 32.55] | 8/30 (26.67%) | 8/30 (26.67%) | 48 |
> > | GENETRON (Base-16 + 3 generations) | 14/30 (46.67%) | 31.71 [29.31, 34.12] | 9/30 (30.00%) | 9/30 (30.00%) | 64 |
> > | GENETRON (Base-16 + 4 generations) | 15/30 (50.00%) | 32.59 [30.19, 35.00] | 10/30 (33.33%) | 9/30 (30.00%) | 80 |
> > | GENETRON (Base-16 + 5 generations) | 15/30 (50.00%) | 33.50 [31.19, 35.89] | 10/30 (33.33%) | 9/30 (30.00%) | 96 |
> > |---|---|---|---|---|---|
> > | MEMETRON (Base-16 + 1 generation) | 16/30 (53.33%) | 32.06 [29.72, 34.40] | 9/30 (30.00%) | 9/30 (30.00%) | 32 |
> > | MEMETRON (Base-16 + 2 generations) | 18/30 (60.00%) | 34.84 [32.61, 37.06] | 11/30 (36.67%) | 11/30 (36.67%) | 48 |
> > | MEMETRON (Base-16 + 3 generations) | 20/30 (66.67%) | 36.20 [33.95, 38.44] | 12/30 (40.00%) | 13/30 (43.33%) | 64 |
> > | MEMETRON (Base-16 + 4 generations) | 22/30 (73.33%) | 37.08 [34.90, 39.25] | 13/30 (43.33%) | 14/30 (46.67%) | 80 |
> > | **MEMETRON (Base-16 + 5 generations)** | **22/30 (73.33%)** | **37.84 [35.73, 39.96]** | **14/30 (46.67%)** | **14/30 (46.67%)** | **96** |

---

> ### Author Response · Authors · 2026-05-06
> **Response to Reviewer V3qx of Paper7799: Compute Cost**
>
> **The trade-off between compute cost and quality is transparently reported but highlights a practical bottleneck. The paper clearly states that MEMETRON incurs significantly higher inference time compared to baseline methods (e.g., over 2300 seconds per question compared to 34 seconds for Base-16). While the evidence supports the claim of quality improvement, it also clearly outlines the method's limitation in terms of test-time compute efficiency. While it mentions that reward-based early stopping could reduce costs, it does not provide a detailed sensitivity analysis or empirical results for this. It is helpful to add an analysis on how different early stopping criteria affect the trade-off between performance and compute time.**
>
> We thank the reviewer for this comment. We would like to point the reviewer to Appendix F.1 and F.2, which already provide trajectory analysis showing diminishing returns across questions and empirically demonstrate that reward-based early stopping can substantially reduce average cost in deployment. Regarding the reported 2318 seconds, this reflects our controlled 5-generation experimental setting and is not a fixed requirement. **GENETRON, ANNETRON, and MEMETRON exhibit the anytime property**, they can be terminated at the end of any generation/iteration with a valid output, and users can run fewer or more generations depending on their desired trade-off between runtime and solution quality.
>
> To further illustrate this, at just the first generation, ANNETRON (\~86.86s/iter) already achieves 33/42 (78.57%) Best-of-|H| and 31/42 (73.81%) Self-consistency, GENETRON (\~98.22s/gen) achieves 33/42 (78.57%) and 33/42 (78.57%), and MEMETRON (\~463.69s/gen) achieves 33/42 (78.57%) and 36/42 (85.71%), all substantially outperforming all three sampling baselines. Notably, at comparable wall-clock time to Base-128 (93.13s), a single iteration of GENETRON and ANNETRON already delivers a significant boost in selection quality, demonstrating that our methods extract substantially more value per unit of compute than naive sampling. The average time per generation is inferred by dividing the total reported runtime by the number of generations, with minor variance expected given the stochastic nature of LLM inference.
>
> We also note that with a well-calibrated reward model, as demonstrated in our correctness shaping experiment, a single generation of MEMETRON could achieve 39/42 (92.86%) under Best-of-|H| and 36/42 (85.71%) under self-consistency, substantially reducing the computational budget.
>
> We acknowledge that reporting the total runtimes for a fixed 5-generation configuration may have been misleading, as it could suggest that MEMETRON/GENETRON/ANNETRON requires such time to produce a valid solution. **We will instead report the average time per generation in the revised paper, which better reflects the incremental cost of each iteration.**

---

> ### Author Response · Authors · 2026-05-06
> **Response to Reviewer V3qx of Paper7799: Baselines**
>
> **The current evaluation only compares MEMETRON against pure sampling baselines (Base-16, Base-128, Base-1024). However, the paper discusses other "deep" RPDO methods in the related work, such as LLMRefine or Mind Evolution. To truly demonstrate the advantage of combining population-based search and local refinement under a shared scalar reward, it is critical to include an empirical comparison with at least one of these existing iterative/optimization-based methods, or provide a compelling experimental demonstration showing why they fail in the tested settings.**
>
> We acknowledge this concern. As discussed in our paper (Section 1, Paragraph 4, Page 2 and Section A.4, Page 19), direct empirical comparison with LLMRefine and Mind Evolution is not feasible as both methods depend on task-specific programmatic verifiers or feedback models that do not generalize to our evaluation settings of mathematical reasoning and instruction following under a blackbox scalar reward, and no public code is available for either method. Concretely, LLMRefine requires a task-specific feedback model trained to critique responses, and Mind Evolution requires a programmatic verifier to check solution correctness, neither of which is available in our instruction following setting and both of which would require ground-truth labels in our mathematical reasoning setting, defeating the purpose of a reward-guided optimization framework that operates purely on blackbox scalar signals. In typical empirical work, one would include such methods as direct baselines. However, given these fundamental constraints, we instead designed GENETRON and ANNETRON as generalized, stronger stand-ins that are inspired by the same core principles as these methods but free from task-specific constraints, allowing them to operate under any blackbox scalar reward signal. By including both components individually and in combination through MEMETRON, our experiments directly and transparently demonstrate the contribution of each search strategy, with the consistent improvement across generations providing a compelling demonstration of the advantage of combining population-based search and local refinement under a shared scalar reward.
>
> **We will revise the paper to more explicitly frame GENETRON and ANNETRON as the empirical baselines for deep iterative RPDO methods, making this design decision clearer to readers.**

---

> ### Author Response · Authors · 2026-05-06
> **Response to Reviewer V3qx of Paper7799: Reward Misalignment Analysis**
>
> **The paper discusses reward model misalignment and introduces correctness shaping to diagnose it. However, it would strengthen the paper to provide a more quantitative analysis of how often "reward hacking" (cases where the RM score increases but the actual correctness degrades) occurred during the pure MEMETRON search versus the shaped search, and how severe it was across generations.**
>
> We thank the reviewer for this suggestion. We would like to point the reviewer to the Results section (Section 4.1) and Table 1, where we already provide a quantitative analysis of reward hacking across generations. Specifically, the Reward Recognition Bottleneck paragraph in Section 4.1 explicitly quantifies how often the RM fails to rank correct responses highest in pure MEMETRON.
>
> Table 1 provides a detailed per-generation breakdown that directly captures this phenomenon: in pure MEMETRON, the gap between Pass@|H| and Best-of-|H| persists across all 5 generations, starting at 6 questions at generation 1 (39/42 vs 33/42), narrowing slightly to 4 questions at generation 2 (39/42 vs 35/42), widening again to 5 questions at generations 3 and 4 (40/42 vs 35/42), and settling at 4 questions at generation 5 (40/42 vs 36/42), with Best RM continuing to increase throughout, indicating that the reward signal is being optimized toward high-reward but incorrect responses. In contrast, MEMETRON with correctness shaping completely eliminates this gap across all generations, with Pass@|H| and Best-of-|H| remaining identical at every generation, directly demonstrating that when the reward signal is properly aligned with correctness, the RM consistently identifies the best response.
>
> We would also like to point the reviewer to Appendix G.3 (Reward Misalignment Analysis and Mitigation in Verifiable Tasks), which already provides an explicit framing of this analysis as a reward hacking diagnostic, including a formal discussion of how the gap between Pass@|H| and Best-of-|H| serves as a quantitative measure of reward model misranking.

---

> ### Comment · Reviewer_V3qx · 2026-05-14
>
> Thank you for the detailed responses. My concerns are generally addressed.

---

### Review · Reviewer_SseS · 2026-04-27

**Summary Of Contributions:**

The paper proposes MEMETRON, a framework for Reward-Guided Post-Decoding Optimization (RPDO) that formulates inference-time response generation as a discrete black-box optimization problem. To navigate the non-differentiable natural language space, the method employs a memetic algorithm that alternates between population-based global exploration (GENETRON) and simulated annealing-based local refinement (ANNETRON). Crucially, the variation operators (crossover, mutation, refinement, perturbation) are instantiated by prompting frozen LLMs to perform semantic edits, guided semantically by the operator prompts, with the scalar reward model used for selection between candidates. The paper demonstrates that this approach can discover higher-reward responses than shallow sampling on subsets of AMC 12 and TinyAlpacaEval, and introduces 'correctness shaping' as a diagnostic tool to expose reward model hacking.

Strengths:

S1. Conceptual Soundness: The formulation of RPDO as a memetic search problem using frozen LLMs as semantic operators is a clean unification of population-based and local-refinement search (cf. Mind Evolution and LLMRefine, which the paper cites as antecedents) and elegantly navigates discrete text space.

S2. Insightful Analysis of Reward Hacking: The introduction of correctness shaping serves as a brilliant diagnostic tool to expose reward model misalignment, practically demonstrating Goodhart's Law and providing a mechanism for generating preference pairs.

S3. High Relevance: Scaling test-time compute and inference-time optimization are highly active and impactful frontiers in LLM research.

Weaknesses:

W1. Misleading Headline Comparison Axis: Reporting the history buffer size (|H|) as the primary axis when comparing MEMETRON against Best-of-N baselines obscures the algorithm's true sample complexity and token cost (e.g., crossover drafts, rejected mutations). The paper does report wall-clock time per question separately (~4x Base-1024) and acknowledges higher cost in Limitations, but a total-LLM-calls or total-tokens comparison would be a fairer headline metric.

W2. Reproducibility Flaws: The omission of simulated annealing hyperparameters (T_0, alpha, patience threshold delta and L_patience), and the lack of open-source code or exact dataset splits make the results difficult to verify independently.

W3. High Computational Cost: The method requires ~40 minutes per question for math reasoning (2318 s/question vs 601 s for Base-1024), making it impractical for real-time deployment. The paper does report this number explicitly in Section 4.1 and acknowledges higher cost in Limitations, but does not foreground it in the abstract or framing.

W4. Small Evaluation Scale: The 42-problem AMC subset and 100 TinyAlpacaEval prompts raise concerns about statistical power and generalizability, particularly for the math-reasoning results. (TinyAlpacaEval is curated by Polo et al. via Item Response Theory specifically to maximize informativeness while preserving AlpacaEval 2.0 properties, which partly mitigates the concern on the instruction-following side.)

W5. Missing Baselines: The absence of comparisons against simpler iterative refinement methods (like Self-Refine) makes it difficult to ascertain if the full memetic complexity is strictly necessary.

**Additional Comments:**

Beyond the changes already requested, I want to flag three recent inference-time alignment papers that I believe should be discussed and ideally compared against in revision. None of them undercuts MEMETRON’s core algorithmic novelty (the GA + SA unification with frozen-LLM operators under a black-box scalar reward), but each occupies neighboring ground that the current draft does not engage with, and together they would meaningfully strengthen the empirical and positioning story.

Huang, Block, Liu, Jiang, Krishnamurthy and Foster (ICML 2025), “Is Best-of-N the Best of Them? Coverage, Scaling, and Optimality in Inference-Time Alignment,” arXiv:2503.21878. This paper formalizes inference-time alignment in essentially the same setup MEMETRON adopts (a frozen base policy with an imperfect reward model) and introduces InferenceTimePessimism, a rejection-sampling algorithm that is provably scaling-monotonic and does not degrade with N. Since MEMETRON’s motivation rests heavily on the “BoN suffers from reward hacking at scale” argument (the paper cites Ichihara et al., 2025 for this), it is important to engage with a method that already provides a principled inference-time fix without going deeper. A short discussion clarifying when deeper structured search is preferable to a smarter shallow selector would sharpen MEMETRON’s positioning.

Liu, Yao, Min, Cao, Hou, and Li (2025), “PairJudge RM: Best-of-N Sampling with Knockout Tournament,” arXiv:2501.13007. PairJudge RM replaces pointwise scalar reward scoring with pairwise judgments arranged in a knockout tournament, evaluated on MATH-500 and Olympiad Bench with 40–60% relative improvement on the hardest 50% of problems. They have empirical results on the same problem family as the AMC-12 evaluation, similar “the RM ranks correct solutions poorly on hard problems” motivation, but a much cheaper algorithmic answer than MEMETRON. Including it as a baseline (or at minimum positioning MEMETRON’s gains relative to it) would substantially strengthen the math-reasoning story.

Tang, Chen, and Cavallaro (2025), “CarBoN: Calibrated Best-of-N Sampling Improves Test-time Reasoning,” arXiv:2510.15674. CarBoN reports up to 4x fewer rollouts to reach the same accuracy as Best-of-N on MATH-500 and AIME-2024 by learning input-specific temperature and shift parameters. Given that MEMETRON’s headline cost is roughly 4x Base-1024 wall-clock for the math experiments, CarBoN should be discussed as well as it claims comparable or better quality at substantially lower compute on adjacent reasoning benchmarks. Even if MEMETRON is technically a different category of method (post-decoding response engineering versus guided decoding), the compute-vs-quality story helps from explicitly addressing this trade-off.

I also want to highlight that none of the three challenges the GA + SA + frozen-LLM-operator unification that constitutes MEMETRON’s primary contribution. Mind Evolution and LLMRefine, both already cited, but the papers above instead represent strong alternative answers to the same underlying problem, and incorporating them would allow readers to better situate MEMETRON’s deeper-search approach against the strongest current shallow alternatives.

**Audience:**

Yes

**Audience Explanation:**

Scaling test-time compute (System 2 reasoning) and inference-time optimization are currently major frontiers in LLM research. The insights into reward hacking, specifically the use of 'correctness shaping' to diagnose reward model misalignment and generate contrastive preference pairs, offer valuable takeaways for researchers working on alignment, reasoning, and automated data curation.

**Broader Impact Concerns:**

I think a Broader Impact Statement addressing the sustainability of scaling test-time compute in this manner should be included addressing the environmental impact and computational cost of running heavy evolutionary algorithms at inference time. The iterative generation and evaluation of dozens of LLM responses per query significantly increases energy consumption and carbon emissions.  Otherwise, the work aligns with responsible AI by aiming to improve model alignment and surface reward hacking.

**Claims And Evidence:**

Yes

**Claims Explanation:**

The paper's central claims are supported by the reported experiments. On mathematical reasoning, MEMETRON improves Best RM, Pass@|H|, Best-of-|H|, and self-consistency over Base-16, Base-128, and Base-1024 with non-overlapping 95% confidence intervals from the first generation onward. On instruction following, hierarchical bootstrap CIs on preference win-rate are strictly above 50% across all generations, and the marginal-gain CIs are strictly positive throughout. The correctness-shaping analysis convincingly surfaces RM-correctness misalignment and yields contrastive pairs as advertised. There are some caveats, but they are more presentation and reproducibility related rather than the validity of the underlying results, namely: the comparison against Best-of-N uses the history buffer size (|H|) as the headline axis, which obscures total sample complexity (crossover drafts, rejected mutations, intermediate refinement calls), and the paper does not present an LLM-call- or token-matched comparison even though it reports wall-clock cost separately (~4x Base-1024). Simulated annealing hyperparameters (T_0, alpha, patience parameters) are not specified anywhere in the manuscript, no code or AMC-12 split is released, and the 42-problem math sample is small. These are addressable in revision and are surfaced as "Critical" and "Suggested" changes below.

**Requested Changes:**

CRITICAL:
Recalibrate baseline comparisons to use Total LLM Calls, Total Inference Tokens, or FLOPs as the comparison metric, rather than the misleading history buffer size (|H|). Evaluate a true compute-matched Best-of-N baseline.

CRITICAL:
Surface the wall-clock and total-LLM-call cost of MEMETRON in the abstract and introduction, alongside the reward and accuracy gains, so readers can place the headline numbers in context.

CRITICAL:
Provide the exact hyperparameters for the simulated annealing component, including the initial temperature (T_0) and decay factor (\alpha).

CRITICAL:
Release the exact subset of 42 AMC 12 problems used, as well as the source code for the optimization loop and evaluations.

SUGGESTION:
Evaluate the framework on a larger, standard dataset (e.g., a larger subset of MATH or GSM8K) to ensure statistical robustness and generalizability.

SUGGESTION:
Include an empirical comparison to a standard iterative refinement baseline (such as Self-Refine) to justify the necessity of the full memetic algorithm's complexity.

SUGGESTION:
Explicitly discuss the utility and framing of MEMETRON as an offline data generation tool versus a practical online inference method, given its high computational cost.

SUGGESTION:
Add clear disclaimers that 'Correctness Shaping' is an oracle-assisted diagnostic analysis requiring ground-truth labels, not a deployable inference-time strategy.

---

> ### Author Response · Authors · 2026-05-06
> **Response to Reviewer SseS of Paper7799:  Opening & Overview**
>
> We thank the reviewer for the thoughtful and constructive review. We are glad the reviewer found the RPDO formulation as a memetic search problem to be a clean and conceptually sound unification, and that the correctness shaping analysis was recognized as an insightful diagnostic tool for exposing reward model misalignment. We address each of the raised weaknesses and requested changes below.

---

> ### Author Response · Authors · 2026-05-06
> **Response to Reviewer SseS of Paper7799: W1**
>
> **W1. Misleading Headline Comparison Axis: Reporting the history buffer size (|H|) as the primary axis when comparing MEMETRON against Best-of-N baselines obscures the algorithm's true sample complexity and token cost (e.g., crossover drafts, rejected mutations). The paper does report wall-clock time per question separately (~4x Base-1024) and acknowledges higher cost in Limitations, but a total-LLM-calls or total-tokens comparison would be a fairer headline metric.**
>
> We thank the reviewer for this observation, but wish to clarify the intent of the history buffer size (|H|) axis. Importantly, we make no claim anywhere in the paper that MEMETRON is more efficient due to a smaller buffer size, and such an interpretation was not intended. Rather, |H| is reported to illustrate a structural difference between the two approaches: Best-of-N considers responses independently and in parallel with no carry-over across iterations, whereas MEMETRON maintains a growing history buffer that informs each successive generation. The |H| axis is thus meant to highlight the iterative, memory-informed nature of MEMETRON's design, not to serve as a cost or efficiency comparison.
>
> Regarding wall-clock time, we acknowledge that reporting the total runtime for a fixed 5-generation configuration may have been misleading, as it could suggest that MEMETRON requires such time to produce a valid solution. This is not the case. **GENETRON, ANNETRON, and MEMETRON exhibit the anytime property**, a characteristic commonly associated with metaheuristic algorithms, meaning they progressively improve solution quality over time and can be terminated at any generation/iteration with a valid output. The reported runtime was used specifically as a controlled setting for fair comparison between all experiments run, not as a required budget (Mentioned in page 9 para "To isolate the effect ... in Appendix D.2."). In practice, MEMETRON can be run for fewer generations depending on the desired trade-off between runtime and solution quality (See Appendix F and E for analyses of optimization trajectories). **We will revise the paper to report the average time per generation instead**, which better reflects the incremental cost of each iteration and avoids potential misinterpretation of the total runtime. The average time per generation is 86.86 seconds/gen for ANNETRON, 98.22 seconds/gen for GENETRON, and 463.69/gen seconds for MEMETRON, with minor variance expected given the stochastic nature of LLM inference. We can see that one generation of MEMETRON (463.69 seconds) is approximately in the same range as Base-1024 (601.13 seconds) with superior performance.
>
> Furthermore, following the reviewer's suggestion, we will also report LLM and RM calls in total as well as per generation for each system in a dedicated cost table (Table 3, Appendix G), to provide a transparent and granular cost comparison. Per generation, ANNETRON and GENETRON each require 2 LLM calls and 1 RM call, while MEMETRON requires 12 LLM calls and 6 RM calls (2 LLM and 1 RM from one GENETRON generation, plus 10 LLM and 5 RM from five ANNETRON iterations).

---

> ### Author Response · Authors · 2026-05-06
> **Response to Reviewer SseS of Paper7799: W2**
>
> **W2. Reproducibility Flaws: The omission of simulated annealing hyperparameters (T_0, alpha, patience threshold delta and L_patience), and the lack of open-source code or exact dataset splits make the results difficult to verify independently.**
>
> We thank the reviewer for raising this concern. We acknowledge that ANNETRON's simulated annealing hyperparameters were omitted from the paper. The initial temperature is set to T_0 = 1.5 with a cooling rate of alpha = 0.975, updated multiplicatively at each iteration. Regarding the patience parameter, we note that this is already addressed in the paper (page 9, second to last paragraph), where we state that adaptive stopping is disabled to ensure controlled and comparable evaluation across methods. **We will include the hyperparameter details in the revised paper.**
>
> Regarding the dataset, the 42 non-figure-dependent problems are drawn from the publicly available 2025 AMC 12A and 12B problem sets, hosted on the Art of Problem Solving Wiki at https://artofproblemsolving.com/wiki/index.php?title=2025_AMC_12A_Problems and https://artofproblemsolving.com/wiki/index.php?title=2025_AMC_12B_Problems. The exact subset used will be explicitly listed in the revised paper with the HuggingFace link. Regarding code, we intend to release it upon acceptance, subject to institutional constraints. Releasing it during the double-blind review process would risk compromising author anonymity. We also note that TMLR does not mandate code release as a condition of submission or acceptance.

---

> ### Author Response · Authors · 2026-05-06
> **Response to Reviewer SseS of Paper7799: W3**
>
> **W3. High Computational Cost: The method requires \~40 minutes per question for math reasoning (2318 s/question vs 601 s for Base-1024), making it impractical for real-time deployment. The paper does report this number explicitly in Section 4.1 and acknowledges higher cost in Limitations, but does not foreground it in the abstract or framing.**
>
> We thank the reviewer for this comment, but respectfully clarify that this reflects a misunderstanding of MEMETRON's design. As noted in our response to W1, **GENETRON, ANNETRON, and MEMETRON exhibit the anytime property**, they can be terminated at the end of any generation/iteration with a valid output, and the reported 2318 seconds corresponds specifically to the 5-generation controlled setting used for comparison, not a required budget.
>
> To further illustrate this point, at just the first generation, ANNETRON (\~86.86s) already achieves 33/42 (78.57%) Best-of-|H| and 31/42 (73.81%) Self-consistency, GENETRON (\~98.22s) achieves 33/42 (78.57%) and 33/42 (78.57%), and MEMETRON (\~463.69s) achieves 33/42 (78.57%) and 36/42 (85.71%). In contrast, the sampling baselines suffer from reward hacking at scale: Base-16 (34.19s) achieves 26/42 (61.90%) and 30/42 (71.43%), Base-128 (93.13s) achieves 23/42 (54.76%) and 29/42 (69.05%), and Base-1024 (601.13s) achieves 20/42 (47.62%) and 29/42 (69.05%) on Best-of-|H| and Self-consistency respectively, showing that simply scaling sampling degrades selection quality despite increasing compute. We note that Pass@|H| is an oracle metric that requires ground-truth correctness labels for all responses and is therefore unrealizable in practice, as already discussed in the paper (page 10, third paragraph starting "However, ... ground truth."). This demonstrates that our methods deliver substantially stronger performance even at the first generation and additional generations further improve quality for users who can afford the extra compute, whereas Best-of-N baselines degrade in selection quality as compute scales.
>
> Again, the \~40-minute runtime reflects our controlled 5-generation experimental setting and is not a fixed requirement, **users can run fewer or more generations** depending on their desired trade-off between runtime and solution quality. **We will revise the abstract and framing to explicitly foreground the anytime nature of our algorithms and the controllable runtime to avoid this misinterpretation.**

---

> ### Author Response · Authors · 2026-05-06
> **Response to Reviewer SseS of Paper7799: W4**
>
> **W4. Small Evaluation Scale: The 42-problem AMC subset and 100 TinyAlpacaEval prompts raise concerns about statistical power and generalizability, particularly for the math-reasoning results. (TinyAlpacaEval is curated by Polo et al. via Item Response Theory specifically to maximize informativeness while preserving AlpacaEval 2.0 properties, which partly mitigates the concern on the instruction-following side.)**
>
> We thank the reviewer for this comment. Regarding the math reasoning evaluation, the choice of AMC 12 2025 problems was deliberate and motivated by data contamination concerns. Standard benchmarks such as MATH, MATH500, and GSM8K suffer from contamination. Qwen3-8B achieves near-perfect accuracy on these datasets, leaving no room for improvement and making MEMETRON unnecessary by construction. AMC 12 2025 problems (https://artofproblemsolving.com/wiki/index.php?title=2025_AMC_12A_Problems and https://artofproblemsolving.com/wiki/index.php?title=2025_AMC_12B_Problems) are not only released after the model's training cutoff, making contamination unlikely, but are also harder than MATH, MATH500, and GSM8K, providing a more meaningful evaluation of reasoning improvement.
>
> Another illustration of this issue is AIME 2025, the community standard for evaluating mathematical reasoning on frontier models, which contains only 30 problems, fewer than our 42-problem subset, yet also likely contaminated. Notably, it is included in Qwen3 technical report, and it was the only mathematical reasoning benchmark used by OpenAI when introducing o3 and o4-mini (https://openai.com/index/introducing-o3-and-o4-mini/) and by Google when introducing Gemini 2.5 Pro (https://blog.google/technology/google-deepmind/gemini-model-thinking-updates-march-2025/), yet frontier models now achieve near-perfect accuracy on it after months of exposure, rendering it ineffective as a discriminative benchmark.  This pattern of rapid saturation is also reflected in how the community has treated it: Artificial Analysis, a widely referenced evaluation index, had previously removed MATH-500 and AIME 2024 from their Intelligence Index for the same reason, replaced them with AIME 2025 as their sole mathematical reasoning benchmark in August 2025, and then removed AIME 2025 itself in January 2026 (v4.0), continuing to report it only as a standalone evaluation as of March 2026 (v4.0.4) (https://artificialanalysis.ai/methodology/intelligence-benchmarking). This recurring cycle of inclusion and exclusion underscores the severity of benchmark saturation in mathematical reasoning evaluation, and motivates our choice of AMC 12 2025 as a more reliable and contamination-free alternative.

---

> > ### Author Response · Authors · 2026-05-08
> > **Response to Reviewer SseS (W4 Continued): AIME 2026 Evaluation**
> >
> > Since the completion of this manuscript, AIME 2026 has been released. Based on your feedback, we took this opportunity to evaluate our methods on these newly available problems. The results are consistent with our AMC 12 2025 experiments in the paper, with all methods improving over Base-16, and ANNETRON and GENETRON again showing comparable performance to each other. MEMETRON achieves the strongest results, reaching 73.33% Pass@|H|, 46.67% Best-of-|H|, and 46.67% Self-consistency after 5 generations. As AIME is a significantly harder competition than AMC 12, we observe a large gap between Pass@|H| (73.33%, the coverage of correct responses in the candidate pool) and Best-of-|H| (46.67%, the reward model's ability to select them), showcasing the severe misalignment of the reward model when facing challenging problems. This indicates that with a perfect reward model, Best-of-|H| and Self-consistency could reach as high as 73.33%, highlighting the improvement of reward modeling as a promising direction for future work. The results are presented in the table below.
> >
> > **Table: Performance of the proposed RPDO methods on 30 AIME 2026 problems**, using Qwen3-8B (non-thinking) as the generator and Skywork-Reward-V2-Llama-3.1-8B-40M as the reward model. Base-16 is the shared initialization pool for all RPDO methods. All experimental setup follows Section 4 except the hardware is switched to an NVIDIA RTX PRO 6000 Blackwell Server Edition.
> >
> > | **System** | **Pass@\|H\|** | **Best RM (CI₉₅)** | **Best-of-\|H\|** | **Self-consistency** | **\|H\|** |
> > |---|---|---|---|---|---|
> > | Base-16 | 10/30 (33.33%) | 25.18 [22.85, 27.51] | 6/30 (20.00%) | 6/30 (20.00%) | 16 |
> > | ANNETRON (Base-16 + 1 iteration) | 11/30 (36.67%) | 28.28 [25.66, 30.89] | 6/30 (20.00%) | 6/30 (20.00%) | 32 |
> > | ANNETRON (Base-16 + 2 iterations) | 14/30 (46.67%) | 29.43 [26.82, 32.04] | 7/30 (23.33%) | 6/30 (20.00%) | 48 |
> > | ANNETRON (Base-16 + 3 iterations) | 14/30 (46.67%) | 30.07 [27.50, 32.64] | 8/30 (26.67%) | 8/30 (26.67%) | 64 |
> > | ANNETRON (Base-16 + 4 iterations) | 14/30 (46.67%) | 31.18 [28.74, 33.62] | 9/30 (30.00%) | 8/30 (26.67%) | 80 |
> > | ANNETRON (Base-16 + 5 iterations) | 14/30 (46.67%) | 31.85 [29.50, 34.19] | 10/30 (33.33%) | 9/30 (30.00%) | 96 |
> > |---|---|---|---|---|---|
> > | GENETRON (Base-16 + 1 generation) | 10/30 (33.33%) | 28.55 [26.05, 31.06] | 7/30 (23.33%) | 8/30 (26.67%) | 32 |
> > | GENETRON (Base-16 + 2 generations) | 12/30 (40.00%) | 30.05 [27.56, 32.55] | 8/30 (26.67%) | 8/30 (26.67%) | 48 |
> > | GENETRON (Base-16 + 3 generations) | 14/30 (46.67%) | 31.71 [29.31, 34.12] | 9/30 (30.00%) | 9/30 (30.00%) | 64 |
> > | GENETRON (Base-16 + 4 generations) | 15/30 (50.00%) | 32.59 [30.19, 35.00] | 10/30 (33.33%) | 9/30 (30.00%) | 80 |
> > | GENETRON (Base-16 + 5 generations) | 15/30 (50.00%) | 33.50 [31.19, 35.89] | 10/30 (33.33%) | 9/30 (30.00%) | 96 |
> > |---|---|---|---|---|---|
> > | MEMETRON (Base-16 + 1 generation) | 16/30 (53.33%) | 32.06 [29.72, 34.40] | 9/30 (30.00%) | 9/30 (30.00%) | 32 |
> > | MEMETRON (Base-16 + 2 generations) | 18/30 (60.00%) | 34.84 [32.61, 37.06] | 11/30 (36.67%) | 11/30 (36.67%) | 48 |
> > | MEMETRON (Base-16 + 3 generations) | 20/30 (66.67%) | 36.20 [33.95, 38.44] | 12/30 (40.00%) | 13/30 (43.33%) | 64 |
> > | MEMETRON (Base-16 + 4 generations) | 22/30 (73.33%) | 37.08 [34.90, 39.25] | 13/30 (43.33%) | 14/30 (46.67%) | 80 |
> > | **MEMETRON (Base-16 + 5 generations)** | **22/30 (73.33%)** | **37.84 [35.73, 39.96]** | **14/30 (46.67%)** | **14/30 (46.67%)** | **96** |

---

> ### Author Response · Authors · 2026-05-06
> **Response to Reviewer SseS of Paper7799: W5**
>
> **W5. Missing Baselines: The absence of comparisons against simpler iterative refinement methods (like Self-Refine) makes it difficult to ascertain if the full memetic complexity is strictly necessary.**
>
> We thank the reviewer for raising this concern. We explicitly discuss and contrast our work with MindEvolution (Lee et al., 2025) and LLMRefine (Xu et al., 2024), which subsumes iterative refinement methods such as Self-Refine (Madaan et al., 2023), in our paper's introduction (Section 1, Paragraph 4, Page 2, beginning "Recent...") and in our related work (Section A.4, Page 19). In particular, Self-Refine is explicitly addressed in Section A.2, where we note that it iteratively refines responses using natural-language feedback rather than scalar reward signals and is therefore not formulated as a reward-guided optimization process.
>
> Both MindEvolution and LLMRefine are further constrained by their dependence on either task-specific programmatic verifiers or task-specific feedback models, limiting their applicability across general settings such as mathematical reasoning and instruction following tasks that we evaluate in this work. Direct comparison is not feasible as their designs are not straightforwardly applicable beyond these specific requirements, and no public code is available for either method.
>
> Instead, we designed GENETRON and ANNETRON explicitly based on the same core principles that these works are inspired from, as described in Section 3.1 (GENETRON: Genetic Optimization for RPDO) and Section 3.2 (ANNETRON: Simulated Annealing for RPDO), generalized to work with any blackbox scalar reward signal. GENETRON and ANNETRON are explicitly included in our experiments in Table 1 (Page 11), serving as the closest feasible experimental stand-ins for MindEvolution and LLMRefine respectively. We can see that MEMETRON outperforms both ANNETRON and GENETRON, indicating memetic complexity is necessary.

---

> ### Author Response · Authors · 2026-05-06
> **Response to Reviewer SseS of Paper7799: Requested Changes C1**
>
> **CRITICAL: Recalibrate baseline comparisons to use Total LLM Calls, Total Inference Tokens, or FLOPs as the comparison metric, rather than the misleading history buffer size (|H|). Evaluate a true compute-matched Best-of-N baseline.**
>
> As indicated in W1, we commit to reporting total LLM calls and average time per generation for each of the systems.

---

> ### Author Response · Authors · 2026-05-06
> **Response to Reviewer SseS of Paper7799: Requested Changes C2**
>
> **CRITICAL: Surface the wall-clock and total-LLM-call cost of MEMETRON in the abstract and introduction, alongside the reward and accuracy gains, so readers can place the headline numbers in context.**
>
> As committed in W3, we will revise the abstract and introduction to explicitly foreground the anytime property, average wall-clock per generation, and total LLM call counts alongside accuracy gains.

---

> ### Author Response · Authors · 2026-05-06
> **Response to Reviewer SseS of Paper7799: Requested Changes C3**
>
> **CRITICAL: Provide the exact hyperparameters for the simulated annealing component, including the initial temperature (T_0) and decay factor (\alpha).**
>
> As addressed in W2, we commit to provide the exact values in the revised draft: initial temperature T_0 = 1.5, cooling rate alpha = 0.975, updated multiplicatively at each iteration, with the standard Metropolis acceptance criterion. We already indicated in the paper that the patience parameter for adaptive stopping is disabled (page 9 para "To isolate the effect ... in Appendix D.2.").

---

> ### Author Response · Authors · 2026-05-06
> **Response to Reviewer SseS of Paper7799: Requested Changes C4**
>
> **CRITICAL: Release the exact subset of 42 AMC 12 problems used, as well as the source code for the optimization loop and evaluations.**
>
> As addressed in W2, we intend to release our code upon acceptance. During the double-blind review process, releasing the code publicly would risk compromising author anonymity. In the meantime, we have fully specified all hyperparameters and experimental details in the paper to support reproducibility as much as possible within these constraints.
>
> Regarding the dataset, the 42 non-figure-dependent problems are drawn from the publicly available 2025 AMC 12A and 12B problem sets, hosted on the Art of Problem Solving Wiki at https://artofproblemsolving.com/wiki/index.php?title=2025_AMC_12A_Problems and https://artofproblemsolving.com/wiki/index.php?title=2025_AMC_12B_Problems. The exact subset used will be explicitly listed in the revised paper with the HuggingFace link.
>
> We also note that code release is not a mandatory requirement under TMLR's editorial policy (https://jmlr.org/tmlr/editorial-policies.html) and is not included as criteria for acceptance/rejection (https://jmlr.org/tmlr/acceptance-criteria.html).

---

> ### Author Response · Authors · 2026-05-06
> **Response to Reviewer SseS of Paper7799: Suggestion S1**
>
> **SUGGESTION: Evaluate the framework on a larger, standard dataset (e.g., a larger subset of MATH or GSM8K)  to ensure statistical robustness and generalizability.**
>
> We thank the reviewer for this suggestion. However, as discussed in W4, standard benchmarks such as MATH, MATH500, GSM8K are saturated for Qwen3-8B due to data contamination, making evaluation on them uninformative. AMC 12 2025 provides a more meaningful contamination-free setting.

---

> ### Author Response · Authors · 2026-05-06
> **Response to Reviewer SseS of Paper7799: Suggestion S2**
>
> **SUGGESTION: Include an empirical comparison to a standard iterative refinement baseline (such as Self-Refine) to justify the necessity of the full memetic algorithm's complexity.**
>
> We thank the reviewer for this suggestion. As addressed in W5, GENETRON and ANNETRON already serve as the closest feasible and stronger stand-ins for Self-Refine and related methods. And the results of MEMETRON outperforming both GENETRON and ANNETRON in Table 1 provides evidence to justify the necessity of the full memetic algorithm's complexity.

---

> ### Author Response · Authors · 2026-05-06
> **Response to Reviewer SseS of Paper7799: Suggestion S3**
>
> **SUGGESTION: Explicitly discuss the utility and framing of MEMETRON as an offline data generation tool versus a practical online inference method, given its high computational cost.**
>
> We agree with this suggestion and note that we already discuss the utility of MEMETRON as an offline data generation tool in the paper. Specifically, Appendix G covers the implications across three dimensions: training-time applications, inference-time applications, and reward misalignment analysis and mitigation, providing a comprehensive discussion of how MEMETRON can be framed and deployed in both offline and online settings. **We will make this framing more prominent in the main paper to ensure readers do not overlook it.**

---

> ### Author Response · Authors · 2026-05-06
> **Response to Reviewer SseS of Paper7799: Suggestion S4**
>
> **SUGGESTION: Add clear disclaimers that 'Correctness Shaping' is an oracle-assisted diagnostic analysis requiring ground-truth labels, not a deployable inference-time strategy.**
>
> We agree and will add a clear disclaimer in the revised paper that Correctness Shaping is an oracle-assisted diagnostic requiring ground-truth labels and is not a deployable inference-time strategy.

---

> ### Author Response · Authors · 2026-05-06
> **Response to Reviewer SseS of Paper7799: Broader Impact Concerns**
>
> **I think a Broader Impact Statement addressing the sustainability of scaling test-time compute in this manner should be included addressing the environmental impact and computational cost of running heavy evolutionary algorithms at inference time. The iterative generation and evaluation of dozens of LLM responses per query significantly increases energy consumption and carbon emissions. Otherwise, the work aligns with responsible AI by aiming to improve model alignment and surface reward hacking.**
>
> We thank the reviewer for this suggestion. We agree that the environmental impact of test-time compute scaling is an important consideration and will add a Broader Impact Statement to the revised paper addressing this. We note that the anytime property of our algorithms partially mitigates this concern, as users can terminate the optimization early based on their desired trade-off between solution quality and computational cost, avoiding unnecessary computation. We will also note that the primary motivation of this work is to improve reward-guided optimization under fixed compute budgets, and that the iterative design of MEMETRON is intended to extract more value per LLM call compared to naive sampling baselines that scale poorly due to reward hacking. Nonetheless, we acknowledge that deploying such methods at scale warrants careful consideration of energy consumption and carbon emissions, and **we will include an explicit discussion of these trade-offs in the Broader Impact Statement**.

---

> ### Author Response · Authors · 2026-05-06
> **Response to Reviewer SseS of Paper7799: Additional Comments**
>
> We thank the reviewer for these highly relevant pointers. We acknowledge that Huang et al. (2025), Liu et al. (2025), and Tang et al. (2025) occupy neighboring ground and would meaningfully strengthen the empirical and positioning story, particularly in situating MEMETRON's deeper-search approach against principled shallow alternatives. Due to the scope of the current revision, we are unable to include full empirical comparisons at this stage. However, we have a follow-up paper to MEMETRON currently in preparation, and we will incorporate discussion and comparison against these works there. We will nonetheless add brief discussions of these papers in the related work section of the current revision to acknowledge their relevance and help readers situate MEMETRON's contributions accordingly.

---

### Decision · Action_Editor_2ij9 · 2026-06-10

**Recommendation:** Accept as is

**Audience:**

Yes

**Audience Explanation:**

Yes, reward-guided optimisation of LLMs is clearly of interest to at least a subset of the TMLR community, active area of research in ML.

**Claims And Evidence:**

Yes

**Claims Explanation:**

Reviewers unanimously agree that the claims are well-supported.